# On the Convergence of Adam under Non-uniform Smoothness: Separability from SGDM and Beyond

## Abstract

This paper aims to clearly distinguish between Stochastic Gradient Descent with Momentum (SGDM) and Adam in terms of their convergence rates. We demonstrate that Adam achieves a faster convergence compared to SGDM under the condition of non-uniformly bounded smoothness. Our findings reveal that: (1) in deterministic environments, Adam can attain the known lower bound for the convergence rate of deterministic first-order optimizers, whereas the convergence rate of Gradient Descent with Momentum (GDM) has higher order dependence on the initial function value; (2) in stochastic setting, Adam's convergence rate upper bound matches the lower bounds of stochastic first-order optimizers, considering both the initial function value and the final error, whereas there are instances where SGDM fails to converge with any learning rate. These insights distinctly differentiate Adam and SGDM regarding their convergence rates. Additionally, by introducing a novel stopping-time based technique, we further prove that if we consider the minimum gradient norm during iterations, the corresponding convergence rate can match the lower bounds across all problem hyperparameters. The technique can also help proving that Adam with a specific hyperparameter scheduler is parameter-agnostic, which hence can be of independent interest.

## 1 Introduction

Among various optimization techniques, the Adam optimizer Kingma & Ba (2014); Loshchilov & Hutter (2019) stands out due to its empirical success in a wide range of deep learning applications, especially for pre-training large foundation models with enormous data Touvron et al. (2023); Brown et al. (2020); Zhang et al. (2022a); Rae et al. (2021); Chowdhery et al. (2022); Du et al. (2021). This popularity of Adam can be attributed to its adaptive learning rate mechanism, which smartly adjusts the step size for each parameter, allowing flexible and robust learning rate choices. Adam's versatility is further highlighted by its consistent performance in training various kinds of models, making it a preferred optimizer in both academic and industrial settings Schneider et al. (2022). Its empirical success extends beyond standard benchmarks to real-world challenges, where it often delivers state-of-the-art results. This track record solidifies Adam's position as a fundamental tool for deep learning practitioners.

Exploring the theoretical foundations of the Adam optimizer, particularly why it often outperforms traditional optimizers like Stochastic Gradient Descent with Momentum (SGDM), is an intriguing yet complex task. Understanding Adam's convergence behavior is challenging, especially in settings defined by standard convergence rate analysis. In these settings, assumptions include uniformly bounded smoothness and finite gradient noise variance. Current research indicates that under these conditions, SGDM can attain the lower bound of the convergence rate for all first-order optimizers Carmon et al. (2017). This finding implies that, theoretically, Adam's convergence rate should not exceed that of SGDM. This theoretical result contrasts with practical observations where Adam frequently excels, presenting a fascinating challenge for researchers. It highlights the need for more refined theoretical models that can bridge the gap between Adam's empirical success and its theoretical understanding.

Recent research by Zhang et al. (2019) has provided valuable insights into the complexity of neural network optimization, particularly challenging the assumption of uniform bounded smoothness. Their observations indicate that smoothness often varies, showing a positive correlation with the norm of the gradient and experiencing considerable fluctuations during the optimization process. Building on this, they introduce the $(L_0, L_1)$-smooth condition (detailed in our Assumption 1), which posits that local smoothness can be bounded in relation to the gradient norm. This concept presents an exciting opportunity to theoretically demonstrate that Adam could potentially converge faster than SGDM. However, even in the relatively simpler deterministic settings, no study has yet conclusively shown this to be the case.

To effectively compare the convergence rates of Adam and Stochastic Gradient Descent with Momentum (SGDM), it's essential to establish an upper bound on Adam's convergence rate and a lower bound for SGDM, and then prove Adam's superiority. This endeavor faces several challenges. First, the known lower bound for SGDM's convergence rate is only available in deterministic settings without momentum Zhang et al. (2019); Crawshaw et al. (2022). Moreover, this result is based on a scenario where the counter-example objective function is selected after fixing the learning rate. This procedure deviates from more common practices where the learning rate is adjusted after defining the objective function Drori & Shamir (2020); Carmon et al. (2017); Arjevani et al. (2022), casting doubts on the standard applicability of this lower bound. Secondly, for Adam, the current assumptions required to derive an upper bound for its convergence rate are quite strict. These include assumptions like bounded adaptive learning rates or deterministically bounded noise Wang et al. (2022); Li et al. (2023a). However, even under these constraints, the convergence rates obtained for Adam are weaker than those of algorithms like clipped SGDM Zhang et al. (2019).

These complexities hinder a straightforward comparison between the convergence rates of Adam and SGDM, highlighting a significant gap in the theoretical understanding that remains to be bridged.

**Our contributions.** In this paper, we aim to bridge the gap and summarize our contributions as follows.

- We separate the convergence rate of Adam and SGDM under $(L_0, L_1)$-smooth condition both in the deterministic setting and in the stochastic setting.

  - In the deterministic setting, for the first time, we prove that under the $(L_0, L_1)$-smooth condition, the convergence rate of the Adam optimizer can match the existing lower bound for first-order deterministic optimizers, up to numerical constants. Additionally, we establish a new lower bound for the convergence rate of GDM, where one is allowed to tune the learning rate and the momentum coefficient after the problem is fixed. The lower bound exhibits a higher order dependence on the initial function value gap compared to the upper bound of Adam. This distinction clearly separates Adam and GDM for the deterministic setting.

  - In the stochastic setting, for the first time, we prove that under the $(L_0, L_1)$-smooth condition, the convergence rate of Adam matches the existing lower bound for first-order stochastic optimizers regarding the initial function value $f(\boldsymbol{w}_1) - f^*$ and the final error $\varepsilon$. In contrast, counterexamples exist where SGDM fails to converge, irrespective of the learning rate and momentum coefficient. These findings distinctly separate the convergence properties of Adam and SGDM in stochastic settings.

- With the aid of a novel stopping time based technique, we further demonstrate that the convergence rate of minimum error point of Adam can match the lower bound across all problem hyperparameters. We demonstrate that such a technique can be of independent interest by proving that Adam with specific scheduler is parameter-agnostic based on the stopping time.

## 2 RELATED WORKS

**Convergence analysis under non-uniform smoothness.** Observations from empirical studies on deep neural network training indicate that local smoothness can vary significantly throughout the optimization process. In response to this, Zhang et al. (2019) introduced the $(L_0, L_1)$-smooth condition, which posits that local smoothness can be bounded by a linear function of the gradient norm. Subsequent works have extended this concept by generalizing the linear function to polynomials Chen et al. (2023); Li et al. (2023a), or to more general functions Mei et al. (2021). Under non-uniform

smoothness, convergence properties of various optimizers have been studied. For instance, upper bounds on the convergence rate have been established for optimizers such as Clipped SGDM Zhang et al. (2020), sign-based optimizers Jin et al. (2021); Hübler et al. (2023); Sun et al. (2023), AdaGrad Faw et al. (2023); Wang et al. (2023b), variance-reduction methods Reisizadeh et al. (2023); Chen et al. (2023), and trust-region methods Xie et al. (2023). However, research on lower bounds has been comparatively limited, with results primarily focusing on Gradient Descent.

**Convergence analysis of Adam.** The development of convergence analysis for Adam has been quite tortuous. While Adam was originally proposed with a convergence guarantee Kingma & Ba (2014), subsequent analysis by Reddi et al. (2018) pointed out flaws in this initial analysis and provided counterexamples claiming that Adam could fail to converge. Only recently, Shi et al. (2021) and Zhang et al. (2022b) have shown that the counterexamples in Reddi et al. (2018) only rule out the possibility that Adam can converge problem-agnostically, and it is still possible that Adam can converge with problem-dependent hyperparameters.

So far, several works have established the convergence of Adam under the $L$-smooth condition. Zaheer et al. (2018) proved that Adam without momentum can converge to the neighborhood of stationary points by additionally assuming that $\lambda$ is large. De et al. (2018) showed that Adam without momentum can converge to stationary points but under the strong assumption that the sign of gradients does not change during the optimization. Zou et al. (2019), Défossez et al. (2022), and Guo et al. (2021) derived the convergence of Adam by assuming the stochastic gradient is bounded. Shi et al. (2021) and Zhang et al. (2022b) characterized the convergence of random-reshuffling Adam but suffer from sub-optimal rates. He et al. (2023) studied the non-ergodic convergence of Adam under a bounded gradient assumption, while Hong & Lin (2023) provided high-probability guarantees for Adam under a deterministically bounded noise assumption. A concurrent work by Wang et al. (2023a) shows that Adam can achieve the lower bound of first-order optimizers with respect to the final error $\varepsilon$ under standard assumptions, but it is unknown whether Adam can match the lower bound with respect to other problem specifics.

On the other hand, closely related to our work, there are only two works studying the convergence of Adam under non-uniform smoothness Wang et al. (2022); Li et al. (2023a), both with restricted assumptions and results. We will provide a detailed discussion in Section 4.

## 3 PRELIMINARY

**Notations.** In this paper, we will use asymptotic notations $\mathcal{O}, \Omega, \Theta$ to respectively denote asymptotically smaller, larger , and equivalent. We also use $\tilde{\mathcal{O}}, \tilde{\Omega}, \tilde{\Theta}$ to indicate that there is logarithmic factor hidden. We denote $\mathcal{F}_t$ as the filter given by $\boldsymbol{w}_1, \cdots, \boldsymbol{w}_t$.

**Problem and Algorithm.** We study the unconstrained minimization problem $\min_{\boldsymbol{w}} f(\boldsymbol{w})$. We present the psedo-code of Adam as follows.

---
**Algorithm 1** Adam Optimizer
---
**Input:** Stochastic oracle $\boldsymbol{O}$, learning rate $\eta > 0$, initial point $\boldsymbol{w}_1 \in \mathbb{R}^d$, initial conditioner $\boldsymbol{\nu}_0 \in \mathbb{R}^+$, initial momentum $\boldsymbol{m}_0$, momentum parameter $\beta_1$, conditioner parameter $\beta_2$, number of epoch $T$
**for** $t = 1$ **to** $T$ **do**
    Generate a random $z_t$, and query stochastic oracle $\boldsymbol{g}_t = \boldsymbol{O}_f(\boldsymbol{w}_t, z_t)$
    Calculate $\boldsymbol{\nu}_t = \beta_2 \boldsymbol{\nu}_{t-1} + (1 - \beta_2)\boldsymbol{g}_t^{\odot 2}$
    Calculate $\boldsymbol{m}_t = \beta_1 \boldsymbol{m}_{t-1} + (1 - \beta_1)\boldsymbol{g}_t$
    Update $\boldsymbol{w}_{t+1} = \boldsymbol{w}_t - \eta \frac{1}{\lambda + \sqrt{\boldsymbol{\nu}_t}} \odot \boldsymbol{m}_t$
**end for**

---

We would like to highlight that all the analysis in this paper is for $\lambda = 0$. This is because $\lambda = 0$ means we do not require the adaptive learning rate to be upper bounded (a restrictive assumption in existing works Li et al. (2023a); Guo et al. (2021)) and is most challenging. The proof can be immediately extended to $\lambda > 0$ without any modification.

Meanwhile, we briefly state the SGDM optimizer as follows: with initial point $\boldsymbol{w}_1$ and initial momentum $\boldsymbol{m}_0$, the update of $t$-th iteration of SGDM is given by

$$\boldsymbol{m}_t = \beta \boldsymbol{m}_{t-1} + (1-\beta)\boldsymbol{g}_t, \boldsymbol{w}_{t+1} = \boldsymbol{w}_t - \eta \boldsymbol{m}_t.$$

**Assumptions.** In this paper, all the analyses are established under the following two standard assumptions.

**Assumption 1** (($L_0, L_1$)-smooth condition). *We assume $f$ is differentiable and lower bounded, and there exist non-negative constants $L_0, L_1 > 0$, such that $\forall \boldsymbol{w}_1, \boldsymbol{w}_2 \in \mathbb{R}^d$ satisfying $\|\boldsymbol{w}_1 - \boldsymbol{w}_2\| \leq \frac{1}{L_1}$,*

$$\|\nabla f(\boldsymbol{w}_1) - \nabla f(\boldsymbol{w}_2)\| \leq (L_0 + L_1 \|\nabla f(\boldsymbol{w}_1)\|)\|\boldsymbol{w}_1 - \boldsymbol{w}_2\|.$$

**Assumption 2** (Affine noise variance). *We assume that the stochastic noise $\boldsymbol{g}_t$ is unbiased, i.e., $\mathbb{E}^{|\mathcal{F}_t} \boldsymbol{g}_t = \boldsymbol{G}_t$. We further assume $\boldsymbol{g}_t$ has affine variance, i.e., there exists $\sigma_0 \geq 0, \sigma_1 \geq 1$, $\mathbb{E}^{|\mathcal{F}_t}[\|\boldsymbol{g}_t\|^2] \leq \sigma_0^2 + \sigma_1^2 \|\nabla f(\boldsymbol{w}_t)\|^2$.*

Assumption 1 is a more general form of ($L_0, L_1$)-smooth condition and is equivalent to the Hessian-bound form Zhang et al. (2019) when Hessian exists. Assumption 2 is one of the weakest assumptions on the noise in existing literature, and generalizes bounded variance assumption Li et al. (2023b), bounded gradient assumption Défossez et al. (2022), bounded noise assumption Li et al. (2023a).

## 4 SEPARATING THE CONVERGENCE RATES OF ADAM AND (S)GD

In this section, we elucidate the disparate convergence rates of Adam and (S)GD under Assumptions 1 and 2, examining both deterministic and stochastic settings. We commence with the deterministic scenario before delving into the stochastic complexities.

### 4.1 ANALYSIS FOR THE DETERMINISTIC SETTING

As discussed in the introduction section, to discern the differential convergence rates of deterministic Adam and GD, it is necessary to establish not only Adam's upper bound but also GD's lower bound, given a consistent set of assumptions. Crucially, these bounds must be sufficiently tight to ensure that Adam's upper bound is indeed the lesser. To date, only a couple of studies have addressed the convergence of deterministic Adam. The first, referenced in Wang et al. (2022), indicates a convergence rate of $\mathcal{O}(\frac{(f(\boldsymbol{w}_1)-f^*)^2}{\varepsilon^2})$, which is sub-optimal compared to the classical deterministic rate of $\mathcal{O}(\frac{f(\boldsymbol{w}_1)-f^*}{\varepsilon^2})$ Zhang et al. (2019; 2020) regarding the initial function value gap $(f(\boldsymbol{w}_1) - f^*)$. The second study, Li et al. (2023a), presents a convergence rate that depends polynomially on $\frac{1}{\lambda}$, where $\lambda$ is the small constant introduced to prevent the adaptive learning rate from becoming infinity. Therefore, their result is only non-vacuous when $\lambda$ is large, which deviates from practical settings. Additionally, their bound exhibits an exaggerated dependency on the initial function value gap, yielding $\min_{t \in [T]} \|\nabla f(\boldsymbol{w}_t)\| = \mathcal{O}(\frac{(f(\boldsymbol{w}_1)-f^*)^3}{\varepsilon^2})$. As we will see later, such dependencies create upper bounds that surpass the lower bounds of GD, making them unable to serve our purpose. To overcome these limitations and accurately assess the performance of deterministic Adam, we propose a new theorem that establishes an improved convergence rate for deterministic Adam.

**An upper bound for the convergence rate of deterministic Adam.**

**Theorem 1** (Informal). *Let Assumption 1 hold. Then, $\forall \beta_1, \beta_2 \geq 0$ satisfying $\beta_1^2 < \beta_2 < 1$, $\lambda = 0$, and $\varepsilon = \mathcal{O}(L_0/L_1)$, if $T \geq \Theta\left(\frac{L_0(f(\boldsymbol{w}_1)-f^*)}{\varepsilon^2}\right)$, then Algorithm 1 satisfies*

$$\frac{1}{T}\sum_{t=1}^{T} \|\nabla f(\boldsymbol{w}_t)\| \leq \varepsilon.$$

*Proof.* Please see Appendix B.1 for the formal statement of theorem and the proof. □

Our result offers a tighter bound than those presented in prior studies Wang et al. (2022); Li et al. (2023a). It is noteworthy that under the uniform smoothness constraint—where the objective function's smoothness is capped at $L$ (that is, when $L_0 = L$ and $L_1 = 0$ as per Assumption 1, referred to

as the $L$-smooth condition in existing literature Arjevani et al. (2022); Carmon et al. (2017); Faw et al. (2022))—Assumption 1 is met with $L_0 = L$ and any $L_1 \geq 0$. Consequently, the established lower bound for all first-order optimizers Carmon et al. (2017) pertaining to the $L$-smooth condition inherently provides a lower bound for the $(L_0, L_1)$-smooth condition, which is $\Omega\left(\frac{\sqrt{L_0(f(\mathbf{w}_1) - f^*)}}{\sqrt{T}}\right)$. This coincides with our upper bound up to numerical constants. Such correspondence suggests that our proposed bound is, in fact, optimal.

Our proof strategy utilizes a distinctive Lyapunov function, $f(\boldsymbol{w}_t) + \frac{\beta_1}{2(1-\beta_1)\sqrt[4]{\beta_2}}\eta\frac{||\boldsymbol{m}_{t-1}||^2}{\lambda + \sqrt{\boldsymbol{\nu}_{t-1}}}$, which draws inspiration from the current analysis of Gradient Descent with Momentum (GDM) under the $L$-smooth condition Sun et al. (2019). However, we have introduced significant modifications to accommodate the integration of an adaptive learning rate. This carefully crafted Lyapunov function enables us to effectively control the deviation between the momentum term and the current gradient, even under $(L_0, L_1)$-smooth condition. Through this approach, we successfully establish the final optimal bound.

**Remark 1** (On the comparison with AdaGrad). *Our result also suffices to separate Adam from AdaGrad. It is important to note that the convergence rate of AdaGrad under the $(L_0, L_1)$-smooth condition in a deterministic setting, as reported inWang et al. (2023b), is $\frac{(f(\boldsymbol{w}_1) - f^*)^2}{\varepsilon^2}$. This rate is outperformed by that of Adam[1]. In Appendix B.3, we show that the rate in Wang et al. (2023b) is tight by providing a counterexample. The comparatively slower convergence rate of AdaGrad can be attributed to that $(L_0, L_1)$-smooth condition demands the update norm to be bounded by $\mathcal{O}(1)$ to prevent the local smoothness from exponentially increasing. This, in turn, necessitates a learning rate of $\mathcal{O}(1)$. However, the adaptive conditioner in AdaGrad, which accumulates over time, causes the adaptive learning rate to become excessively small during later training stages, resulting in reduced convergence speed. Conversely, Adam utilizes an exponential moving average for its adaptive learning rate, which prevents the conditioner from accumulating excessively. Consequently, Adam does not suffer from the aforementioned issue.*

**A lower bound for the convergence rate of GDM**

With Adam's upper bound, we then move on to a lower bound for the convergence rate of GDM. In fact, there has already been such lower bounds for GD in the existing literature Zhang et al. (2019); Crawshaw et al. (2022), which we restate as follows:

**Proposition 1** (Theorem 2, Crawshaw et al. (2022)). *Fix $\varepsilon, L_0, L_1$, and $\Delta_1$, with learning rate $\eta$, there exists objective function $f$ satisfying $(L_0, L_1)$-smooth condition and $f(\boldsymbol{w}_1) - f^* = \Delta_1$, such that the minimum step $T$ of GD to achieve final error $\varepsilon$ (i.e., let $\{\boldsymbol{w}_t\}_{t=1}^{\infty}$ be the iterates of GD, and $T \triangleq \min\{t : \|\nabla f(\boldsymbol{w}_t)\| < \varepsilon\}$) satisfies*

$$T = \tilde{\Omega}\left(\frac{L_1^2\Delta_1^2 + L_0\Delta_1}{\varepsilon^2}\right).$$

However, the proposition presents a limitation: the counter-example is chosen after the learning rate has been determined. This approach is inconsistent with standard practices, where hyperparameters are usually adjusted based on the specific task, and deviates from conventional lower bounds Carmon et al. (2017); Arjevani et al. (2022) that offer assurances for optimally-tuned hyperparameters. This type of result does not eliminate the possibility that, if the learning rate were adjusted after selecting the objective function—as is common practice—Gradient Descent (GD) could potentially achieve a markedly faster convergence rate. This misalignment raises concerns about the appropriateness of the proposition's methodology. Moreover, this proposition does not take momentum into account, a technique that is commonly employed in conjunction with GD in practice.

To address these shortcomings, we introduce a new lower bound for GDM. This lower bound is applicable under the standard practice of adjusting hyperparameters after the objective function has been selected. Moreover, it encompasses scenarios where momentum is incorporated.

**Theorem 2** (Informal). *Fixing $\varepsilon, L_0, L_1$, and $\Delta_1$, there exists an objective function $f$ satisfying $(L_0, L_1)$-smooth condition and $f(\boldsymbol{w}_1) - f^* = \Delta_1$, such that for any learning rate $\eta > 0$ and*

---

[1]The state-of-art rate of AdaGrad under $(L_0, L_1)$-smooth condition and stochastic setting is $\frac{(f(\boldsymbol{w}_1) - f^*)^2}{\varepsilon^4}$, which is also worse than the rate of Adam established latter in Theorem 3.

$\beta \in [0, 1]$, the minimum step $T$ of GDM to achieve final error $\varepsilon$ satisfies

$$T = \tilde{\Omega}\left(\frac{L_1^2 \Delta_1^2 + L_0 \Delta_1}{\varepsilon^2}\right).$$

*Proof.* Please see Appendix B.2 for the formal statement of theorem and the proof. □

It should be noticed in the above theorem, the hyperparameters (i.e., the learning rate and the momentum coefficient) are chosen after the objective function is determined, which agrees with practice and the settings of common lower bounds, and overcomes the shortcoming of Proposition 1. Moreover, as shown in Zhang et al. (2019), it is easy to prove that the upper bound of GD's convergence rate is also $\mathcal{O}\left(\frac{L_1^2 \Delta_1^2 + L_0 \Delta_1}{\varepsilon^2}\right)$, which indicates such a lower bound is optimal.

The proof addresses two primary challenges outlined above. The first challenge involves handling momentum. To tackle this, we extend the counterexample provided in Proposition 1 for cases where the momentum coefficient $\beta$ is small. Additionally, we introduce a new counterexample for situations with a large $\beta$, demonstrating how large momentum can bias the optimization process and decelerate convergence. The second challenge is how to derive a universal counterexample such that every hyperparameter setting will lead to slow convergence. We overcome this by a simple but effective trick: we independently put counterexamples for different hyperparameters in Proposition 1 over different coordinates and make it a whole counterexample. Therefore, for different hyperparameters, there will be at least one coordinate converge slowly, which leads to the final result.

**Separating deterministic Adam and GDM.** Upon careful examination of Theorem 1 and Theorem 2, it becomes apparent that the convergence rate of GDM is inferior to that of Adam since $\frac{\sum_{t=1}^T \|\boldsymbol{G}_t\|}{T} \geq \min_{t \in [T]} \|\boldsymbol{G}_t\|$. Notably, GDM exhibits a more pronounced dependency on the initial function value gap in comparison to Adam. This implies that with a sufficiently poor initial point, the convergence of GDM can be significantly slower than that of Adam. The underlying reason for this disparity can be attributed to GDM's inability to adeptly manage varying degrees of sharpness within the optimization landscape. Consequently, GDM necessitates a learning rate selection that is conservative, tailored to the most adverse sharpness encountered—often present during the initial optimization stages.

### 4.2 ANALYSIS FOR THE STOCHASTIC SETTING

Transitioning to the more complex stochastic setting, we extend our analysis beyond the deterministic framework. As with our previous approach, we start by reviewing the literature to determine if the existing convergence rates for Adam under the $(L_0, L_1)$-smooth condition can delineate a clear distinction between the convergence behaviors of Adam and Stochastic Gradient Descent with Momentum (SGDM). In fact, the only two studies that delve into this problem are the ones we discussed in Section 4.1, i.e., Wang et al. (2022); Li et al. (2023a). However, these results pertaining to Adam are contingent upon rather stringent assumptions. Wang et al. (2022) postulates that stochastic gradients not only conform to the $(L_0, L_1)$-smooth condition but are also limited to a finite set of possibilities. These assumptions are more restrictive than merely assuming that the true gradients satisfy the $(L_0, L_1)$-smooth condition, and such strong prerequisites are seldom employed outside of the analysis of variance-reduction algorithms. Meanwhile, Li et al. (2023a) aligns its findings on stochastic Adam with those on deterministic Adam, leading to a polynomial dependency on $1/\lambda$, which deviates from practical scenarios as discussed in Section 4.1. Furthermore, it presumes an a.s. bounded difference between stochastic gradients and true gradients, an assumption that closely resembles the boundedness of stochastic gradients and is more limiting than the standard assumption of bounded variance for stochastic gradients.

These more restricted and non-standard assumptions cast challenges in establishing a lower bound for the convergence of SGDM in the relevant contexts, let alone attempting a comparison between SGDM and Adam. In addition to the fact that these upper bounds fail to facilitate a clear comparison between Adam and SGDM, there are also concerns regarding their convergence rates. Wang et al. (2022) reports a convergence rate of $\frac{(f(\boldsymbol{w}_1) - f^*)^2}{\varepsilon^8}$, which has a higher-order dependence on the initial function value gap and the final error than the $\frac{(f(\boldsymbol{w}_1) - f^*)}{\varepsilon^4}$ rate established for Clipped SGDM under

the $(L_0, L_1)$-smooth condition Zhang et al. (2020)[2]. Furthermore, Li et al. (2023a) indicates a convergence rate of $\mathcal{O}(\frac{(f(\boldsymbol{w}_1) - f^*)^4 \operatorname{poly}(1/\lambda)}{\varepsilon^4})$, which, aside from the previously mentioned dependency issues on $1/\lambda$, shows a significantly stronger dependence over the initial function value gap compared to the analysis of Clipped SGDM. This naturally leads to the question of whether such rates for Adam can be improved to match Clipped SGDM.

To tackle these obstacles, we present the following upper bound for Adam.

**An upper bound for the convergence rate of Adam.**

**Theorem 3** (Informal). *Let Assumptions 1 and 2 hold. Then, $\forall 1 > \beta_1 \geq 0$ and $\lambda = 0$, if $\varepsilon \leq \frac{1}{\operatorname{poly}(f(\boldsymbol{w}_1) - f^*, L_0, L_1, \sigma_0, \sigma_1)}$, with a proper choice of learning rate $\eta$ and momentum hyperparameter $\beta_2$, we have if $T \geq \Theta\left(\frac{(L_0 + L_1\sigma_0)\sigma_0^2\sigma_1^2(f(\boldsymbol{w}_1) - f^*)}{\varepsilon^4}\right)$,*

$$\frac{1}{T}\mathbb{E}\sum_{t=1}^{T} \|\nabla f(\boldsymbol{w}_t)\| \leq \varepsilon.$$

*Proof.* Please see Appendix C.1 for the formal statement of theorem and the proof. $\square$

Below we include several discussions regarding Theorem 3. To begin with, one can immediately observe that Theorem 3 only requires Assumptions 1 and 2, and the convergence rate with respect to the initial function value gap and the final error $\frac{f(\boldsymbol{w}_1) - f^*}{\varepsilon^4}$ matches that of Clipped SGDM Zhang et al. (2020) even with a weaker noise assumption. Therefore, our result successfully mitigate these barriers raised above. Indeed, to the best of our knowledge, it is for the first time that an algorithm is shown to converge with rate $\mathcal{O}\left(\frac{f(\boldsymbol{w}_1) - f^*}{\varepsilon^4}\right)$ only requiring Assumptions 1 and 2, showcasing the advantage of Adam.

We briefly sketch the proof here before moving on to the result of SGDM. Specifically, the proof is inspired by recent analysis of Adam under $L$-smooth condition Wang et al. (2023a), but several challenges arise during the proof:

- The first challenge lies in the additional error introduced by the $(L_0, L_1)$-smooth condition. We address this by demonstrating that the telescoping sum involving the auxiliary function $\frac{\|\boldsymbol{G}_t\|^2}{\sqrt{\boldsymbol{\nu}_{t-1}}}$, as employed in Wang et al. (2023a), can bound this additional error when the adaptive learning rate is upper bounded. Although the adaptive learning rate in the Adam algorithm is not inherently bounded, we establish that the deviation incurred by employing a bounded surrogate adaptive learning rate is manageable;

- The second challenge involves deriving the desired dependence on the initial function value gap. Wang et al. (2023a) introduces two distinct proof strategies for bounding the conditioner $\boldsymbol{\nu}_t$ and determining the final convergence rate. However, one strategy introduces an additional logarithmic dependence on $\varepsilon$, while the other exhibits sub-optimal dependence on the initial function value gap. We propose a novel two-stage divide-and-conquer approach to surmount this issue. In the first stage, we bound $\boldsymbol{\nu}_t$ effectively. Subsequently, we leverage this bound within the original descent lemma to achieve the optimal dependence on $f(\boldsymbol{w}_1) - f^*$.

**Remark 2** (On the limitations). *Although Theorem 3 addresses certain deficiencies identified in prior studies Wang et al. (2022); Li et al. (2023a), it is not without its limitations. As noted by Arjevani et al. (2022), the established lower bound for the convergence rate of first-order optimization algorithms under the $L_0$-smooth condition with bounded noise variance (specifically, $\sigma_0 = \sigma_0$ and $\sigma_1 = 1$ as stated in Assumption 2) is $\mathcal{O}(\frac{(f(\boldsymbol{w}_1) - f^*)L_0\sigma_0^2}{\varepsilon^4})$. This sets a benchmark for the performance under Assumptions 1 and 2. The upper bound of Adam's convergence rate as presented in Theorem 3 falls short when compared to this benchmark, exhibiting a weaker noise scale dependency ($\sigma_0^3$ as opposed to $\sigma_0^2$) and additional dependencies on $L_1$ and $\sigma_1$.*

*To address these issues, we demonstrate in the subsequent section that by focusing on the convergence of the minimum gradient norm, $\mathbb{E}\min_{t\in[T]} \|\nabla f(\boldsymbol{w}_t)\|$, we can attain an improved convergence rate*

---

[2]While Zhang et al. (2020) also assumes an a.s. bounded gap between stochastic gradients and true gradients.

of $\mathcal{O}(\frac{(f(\boldsymbol{w}_1)-f^*)L_0\sigma_0^2}{\varepsilon^4})$. *This rate aligns with the aforementioned lower bound across all the problem hyperparameters.*

We now establish the lower bound of SGDM. This is, however, more challenging than the deterministic case as to the best of our knowledge, there is no such a lower bound in existing literature (despite that the lower bounds of GD Zhang et al. (2019); Crawshaw et al. (2022) naturally offer a lower bound of SGD, which is considerably loose given the factor of $1/\varepsilon^2$). Intuitively, stochasticity can make the convergence of GDM even worse, as random fluctuations can inadvertently propel the iterations towards regions characterized by high smoothness even with a good initialization. We formulate this insight into the following theorem.

**A lower bound for the convergence rate of SGDM.**

**Theorem 4** (Informal). *Fix $L_0, L_1$, and $\Delta_1$, there exists objective function $f$ satisfying $(L_0, L_1)$-smooth condition and $f(\boldsymbol{w}_1) - f^* = \Delta_1$, and a gradient noise oracle satisfying Assumption 2, such that for any learning rate $\eta > 0$ and $\beta \in [0, 1]$, for all $T > 0$,*

$$\min_{t\in[T]} \mathbb{E}\|\nabla f(\boldsymbol{w}_t)\| = \|\nabla f(\boldsymbol{w}_1)\| \geq L_1\Delta_1.$$

*Proof.* Please see Appendix C.2 for the formal statement of theorem and the proof. □

Theorem 4 provides concrete evidence for the challenges inherent in the convergence of SGDM. It shows that there are instances that comply with Assumption 1 and Assumption 2 for which SGDM fails to converge, regardless of the chosen learning rate and momentum coefficient. This outcome confirms our earlier hypothesis: the stochastic elements within SGDM can indeed adversely affect its convergence properties under non-uniform smoothness.

Our proof is founded upon a pivotal observation: an objective function that escalates rapidly can effectively convert non-heavy-tailed noise into a "heavy-tailed" one. In particular, under the $(L_0, L_1)$-smooth condition, the magnitude of the gradient is capable of exponential growth. As a result, even if the density diminishes exponentially, the expected value of the gradient norm may still become unbounded. This situation mirrors what occurs under the $L$-smooth condition when faced with heavy-tailed noise. Such a dynamic can lead to the non-convergence of SGDM.

**Separating Adam and SGDM.** Considering that Adam can achieve convergence under Assumptions 1 and 2, while SGD cannot, the superiority of Adam over SGDM becomes evident. It is important to note, however, a recent study by Li et al. (2023b), which demonstrates that SGD can converge with high probability under the same assumptions, provided the noise variance is bounded. We would like to contextualize this finding in relation to our work as follows: First, this result does not conflict with our Theorem 4, since our theorem pertains to bounds in expectation rather than with high probability. Second, our comparison of Adam and SGDM within an in-expectation framework is reasonable and aligns with the convention of most existing lower bounds in the literature Carmon et al. (2017); Drori & Shamir (2020); Arjevani et al. (2022). Moreover, establishing high-probability lower bounds is technically challenging, and there are few references to such bounds in the existing literature. Lastly, while we have not derived a corresponding high-probability lower bound for SGD, the upper bound provided by Li et al. (2023b) is $\mathcal{O}(\frac{(f(\boldsymbol{w}_1)-f^*)^4}{\varepsilon^4})$, which indicates a less favorable dependency on the initial function value gap compared to the bound for Adam.

## 5 CAN ADAM REACH THE LOWER BOUND OF THE CONVERGENCE RATE UNDER $(L_0, L_1)$-SMOOTH CONDITION?

As we mentioned in Remark 2, although Theorem 3 matches the lower bound established by Arjevani et al. (2022) with respect to the initial function value gap $f(\boldsymbol{w}_1) - f^*$, the final error $\varepsilon$, and the smoothness coefficient $L_0$, it exhibits sub-optimal dependence on the noise scale $\sigma_0$ and additional dependence on $L_1$ and $\sigma_1$. One may wonder whether these dependencies are inherently unavoidable or if they stem from technical limitations in our analysis.

Upon revisiting the proof, we identified that the sub-optimal dependencies arise from our strategy of substituting the original adaptive learning rate with a bounded surrogate. For example, the correlation between stochastic gradient and adaptive learning rate will introduce an error term $\eta\frac{\sigma_0^2(1-\beta_2)\|\boldsymbol{g}_t\|^2}{\sqrt{\beta_2\boldsymbol{\nu}_{t-1}}\boldsymbol{\nu}_t}$,

detailed in Eq. (8). To bound this term, we add a constant $\lambda$ to $\beta_2 \boldsymbol{\nu}_{t-1}$, allowing us to upper bound $\frac{1}{\sqrt{\beta_2 \boldsymbol{\nu}_{t-1} + \lambda}}$. Consequently, the term $\eta \frac{\sigma_0^2 (1-\beta_2) \|\boldsymbol{g}_t\|^2}{\sqrt{\beta_2 \boldsymbol{\nu}_{t-1} + \lambda \boldsymbol{\nu}_t}}$ can be bounded by $\eta \frac{\sigma_0^2 (1-\beta_2) \|\boldsymbol{g}_t\|^2}{\sqrt{\lambda} \boldsymbol{\nu}_t}$, which has the same order as a second-order Taylor expansion. To control the error introduced by adding $\lambda$, we cannot choose a value for $\lambda$ that is too large. The optimal choice of $\lambda$ for balancing the new error against the original error is $(1 - \beta_2)\sigma_0^2$. This selection results in the original error term $\eta \frac{\sigma_0 \sqrt{1-\beta_2} \|\boldsymbol{g}_t\|^2}{\boldsymbol{\nu}_t}$, which induces an additional $\sigma_0$ factor, ultimately leading to the sub-optimal dependence on $\sigma_0$. Therefore, we need to explore alternative methods to handle the error term to eliminate the sub-optimal dependence on $\sigma_0$.

We begin our analysis by observing that the term $\frac{(1-\beta_2) \|\boldsymbol{g}_t\|^2}{\sqrt{\beta_2 \boldsymbol{\nu}_{t-1}} \boldsymbol{\nu}_t}$ can in fact be bounded by an "approximate telescoping" series of $\frac{1}{\sqrt{\boldsymbol{\nu}_t}}$ (noting an additional coefficient $\frac{1}{\sqrt{\beta_2}}$ in comparison to standard telescoping):

$$\frac{(1-\beta_2) \|\boldsymbol{g}_t\|^2}{\sqrt{\beta_2 \boldsymbol{\nu}_{t-1}} \boldsymbol{\nu}_t} \leq \mathcal{O}\left( \frac{1}{\sqrt{\beta_2 \boldsymbol{\nu}_{t-1}}} - \frac{1}{\sqrt{\boldsymbol{\nu}_t}} \right).$$

Accordingly, summing $\eta \frac{\sigma_0^2 (1-\beta_2) \|\boldsymbol{g}_t\|^2}{\sqrt{\beta_2 \boldsymbol{\nu}_{t-1}} \boldsymbol{\nu}_t}$ over $t$ yields a bound of $\mathcal{O}(\eta \sigma_0^2 \sum_t (1-\beta_2) \frac{1}{\sqrt{\boldsymbol{\nu}_t}})$. However, this term could potentially be unbounded since $\sqrt{\boldsymbol{\nu}_t}$ is not lower bounded. To circumvent this issue, we consider the first-order Taylor's expansion of the descent lemma, which, gives $-\sum_t \eta \frac{\|\nabla f(\boldsymbol{w}_t)\|^2}{\sqrt{\boldsymbol{\nu}_t}}$. Intuitively, if any $\|\nabla f(\boldsymbol{w}_t)\|^2$ is of the order $\mathcal{O}(\sigma_0^2 (1-\beta_2))$, our proof would be completed since we choose $1 - \beta_2 = \Theta(\varepsilon^4)$. In the other case, the term $\mathcal{O}(\eta \sigma_0^2 \sum_t (1-\beta_2) \frac{1}{\sqrt{\boldsymbol{\nu}_t}})$ can be offset by the negative term $-\sum_t \eta \frac{\|\nabla f(\boldsymbol{w}_t)\|^2}{\sqrt{\boldsymbol{\nu}_t}}$. However, formalizing this intuition into a proof is challenging in the context of stochastic analysis, where the randomness across iterations complicates the analysis. Specifically, if we condition on the event that "no gradient norm is as small as $\sigma_0^2 (1 - \beta_2)$," which is supported over the randomness of all iterations, it becomes difficult to express many expected values (such as those from the first-order Taylor expansion) in closed form.

We address this difficulty by introducing a stopping time $\tau \triangleq \min\{t : \|\nabla f(\boldsymbol{w}_{t+1})\|^2 \leq \mathcal{O}(\sigma_0^2 (1 - \beta_2))\}$. By applying the optimal stopping theorem Durrett (2019), we can maintain closed-form expressions for the expected values up to the stopping time, allowing the problematic error term to be absorbed within this interval. Building on this methodology, we formulate the following theorem.

**Theorem 5** (Informal). *Let Assumptions 1 and 2 hold. Then, $\forall 1 > \beta_1 \geq 0$, if $\varepsilon \leq \frac{1}{\text{Poly}(L_0, L_1, \sigma_0, \sigma_1, \frac{1}{1-\beta_1}, f(\boldsymbol{w}_1)-f^*)}$, with a proper choice of learning rate $\eta$ and momentum hyperparameter $\beta_2$, we have that if $T \geq \Theta(\frac{L_0 \sigma_0^2 (f(\boldsymbol{w}_1)-f^*)}{\varepsilon^4})$*

$$\mathbb{E} \min_{t \in [1,T]} \|\nabla f(\boldsymbol{w}_t)\| \leq \varepsilon.$$

*Proof.* Please see Appendix D.1 for the formal statement of theorem and the proof. $\square$

One can easily see that the convergence rate of Theorem 5 matches the lower bound in Arjevani et al. (2022) with respect to all problem hyperparameters up to numerical constants even under the weaker $(L_0, L_1)$-smooth condition. Therefore, such a rate is optimal and provides an affirmative answer to the question raised in the beginning of this section.

One may notice that in the construction of the stopping time, we set the threshold for the squared gradient norm to be $\mathcal{O}(1-\beta_2)$. As we set $1-\beta_2 = \Theta(\varepsilon^4)$, the threshold is actually much smaller than what we aim for, since our goal is to have $\|\nabla f(\mathbf{w}_t)\|^2 \leq \varepsilon^2$. Therefore, based on the stopping-time technique, we can actually show that Adam can converge with an optimal rate of $\mathcal{O}(\varepsilon^{-4})$ when $1 - \beta_2 = \varepsilon^2$, or $1/\sqrt{T}$ if expressed in terms of the iteration number $T$. To the best of our knowledge, this is the first time that Adam has been shown to converge with an optimal rate under the condition that $1 - \beta_2 = \Omega(1/T)$, which greatly enlarges the hyperparameter range. We show in Appendix D.2 that based on this technique, we can show Adam is hyperparameter agnostic even under the $(L_0, L_1)$-smooth condition.

## 6 CONCLUSION

In this paper, we have conducted a mathematical examination of the performance of the Adam optimizer and SGDM within the context of non-uniform smoothness. Our convergence analysis reveals that Adam exhibits a faster rate of convergence compared to SGDM under these conditions. Moreover, we introduce a novel stopping time technique that demonstrates Adam's capability to achieve the existing lower bounds for convergence rates. This finding underscores the robustness of Adam in complex optimization landscapes and contributes to a deeper understanding of its theoretical properties.

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

## A  AUXILIARY LEMMAS

In this section, we provide auxiliary results which will be used in subsequent results.

**Lemma 1.** *We have $\forall t \geq 1$, $\|\boldsymbol{w}_{t+1} - \boldsymbol{w}_t\| \leq \eta \frac{1-\beta_1}{\sqrt{1-\beta_2}\sqrt{1-\frac{\beta_1^2}{\beta_2}}}$.*

*Proof.* We have that

$$\|\boldsymbol{w}_{t+1} - \boldsymbol{w}_t\| = \eta \left| \frac{\boldsymbol{m}_t}{\sqrt{\boldsymbol{\nu}_t}} \right| \leq \eta \frac{\sum_{i=0}^{t-1}(1-\beta_1)\beta_1^i\|\boldsymbol{g}_{t-i}\|}{\sqrt{\sum_{i=0}^{t-1}(1-\beta_2)\beta_2^i\|\boldsymbol{g}_{t-i}\|^2 + \beta_2^t\boldsymbol{\nu}_0}}$$

$$\leq \eta \frac{1-\beta_1}{\sqrt{1-\beta_2}} \frac{\sqrt{\sum_{i=0}^{t-1}\beta_2^i\|\boldsymbol{g}_{t-i}\|^2}\sqrt{\sum_{i=0}^{t-1}\frac{\beta_1^{2i}}{\beta_2^i}}}{\sqrt{\sum_{i=0}^{t-1}\beta_2^i\|\boldsymbol{g}_{t-i}\|^2}} \leq \eta \frac{1-\beta_1}{\sqrt{1-\beta_2}\sqrt{1-\frac{\beta_1^2}{\beta_2}}}.$$

Here the second inequality is due to Cauchy's inequality. The proof is completed. □

The following lemma provides a novel descent lemma under $(L_0, L_1)$-smooth condition.

**Lemma 2.** *Let Assumption 1 hold. Then, for any three points $\boldsymbol{w}^1, \boldsymbol{w}^2, \boldsymbol{w}^3 \in \mathcal{X}$ satisfying $\|\boldsymbol{w}^1 - \boldsymbol{w}^2\| \leq \frac{1}{2L_1}$ and $\|\boldsymbol{w}^1 - \boldsymbol{w}^3\| \leq \frac{1}{2L_1}$, we have*

$$f(\boldsymbol{w}^2) \leq f(\boldsymbol{w}^3) + \langle \nabla f(\boldsymbol{w}^1), \boldsymbol{w}^2 - \boldsymbol{w}^3 \rangle + \frac{1}{2}(L_0 + L_1\|\nabla f(\boldsymbol{w}^1)\|)\|\boldsymbol{w}^2 - \boldsymbol{w}^3\|(\|\boldsymbol{w}^1 - \boldsymbol{w}^3\| + \|\boldsymbol{w}^1 - \boldsymbol{w}^2\|).$$

*Proof.* By the Fundamental Theorem of Calculus, we have

$$f(\boldsymbol{w}^2) = f(\boldsymbol{w}^3) + \int_0^1 \langle \nabla f(\boldsymbol{w}^3 + a(\boldsymbol{w}^2 - \boldsymbol{w}^3)), \boldsymbol{w}^2 - \boldsymbol{w}^3 \rangle \mathrm{d}a$$

$$= f(\boldsymbol{w}^3) + \langle \nabla f(\boldsymbol{w}^1), \boldsymbol{w}^2 - \boldsymbol{w}^3 \rangle + \int_0^1 \langle \nabla f(\boldsymbol{w}^3 + a(\boldsymbol{w}^2 - \boldsymbol{w}^3)) - \nabla f(\boldsymbol{w}^1), \boldsymbol{w}^2 - \boldsymbol{w}^3 \rangle \mathrm{d}a$$

$$\leq f(\boldsymbol{w}^3) + \langle \nabla f(\boldsymbol{w}^1), \boldsymbol{w}^2 - \boldsymbol{w}^3 \rangle + \int_0^1 \|\nabla f(\boldsymbol{w}^3 + a(\boldsymbol{w}^2 - \boldsymbol{w}^3)) - \nabla f(\boldsymbol{w}^1)\|\|\boldsymbol{w}^2 - \boldsymbol{w}^3\| \mathrm{d}a$$

$$\overset{(\star)}{\leq} f(\boldsymbol{w}^3) + \langle \nabla f(\boldsymbol{w}^1), \boldsymbol{w}^2 - \boldsymbol{w}^3 \rangle + \int_0^1 (L_0 + L_1\|\nabla f(\boldsymbol{w}^1)\|)\|a(\boldsymbol{w}^2 - \boldsymbol{w}^1) + (1-a)(\boldsymbol{w}^3 - \boldsymbol{w}^1)\|\|\boldsymbol{w}^2 - \boldsymbol{w}^3\| \mathrm{d}a$$

$$\leq f(\boldsymbol{w}^3) + \langle \nabla f(\boldsymbol{w}^1), \boldsymbol{w}^2 - \boldsymbol{w}^3 \rangle + \frac{1}{2}(L_0 + L_1\|\nabla f(\boldsymbol{w}^1)\|)\|\boldsymbol{w}^2 - \boldsymbol{w}^3\|(\|\boldsymbol{w}^1 - \boldsymbol{w}^3\| + \|\boldsymbol{w}^1 - \boldsymbol{w}^2\|),$$

where Inequality $(\star)$ is because due to

$$\|\boldsymbol{w}^3 + a(\boldsymbol{w}^2 - \boldsymbol{w}^3) - \boldsymbol{w}^1\| = \|a(\boldsymbol{w}^2 - \boldsymbol{w}^1) + (1-a)(\boldsymbol{w}^3 - \boldsymbol{w}^1)\| \leq \frac{1}{L_1},$$

the definition of $(L_0, L_1)$-smooth condition can be applied.

The proof is completed. □

The following lemma is helpful when bounding the second-order term.

**Lemma 3.** *Assume we have $0 < \beta_1^2 < \beta_2 < 1$ and a sequence of real numbers $(a_n)_{n=1}^{\infty}$. Let $b_0 > 0$, $b_n = \beta_2 b_{n-1} + (1-\beta_2)a_n^2$, $c_0 = 0$, and $c_n = \beta_1 c_{n-1} + (1-\beta_1)a_n$. Then, we have*

$$\sum_{n=1}^T \frac{|c_n|^2}{b_n} \leq \frac{(1-\beta_1)^2}{(1-\frac{\beta_1}{\sqrt{\beta_2}})^2(1-\beta_2)}\left(\ln\left(\frac{b_T}{b_0}\right) - T\ln\beta_2\right).$$

*Proof.* This is a lemma commonly adopted in the literature of the convergence of Adam Défossez et al. (2022); Wang et al. (2023a). We invite interesting readers to see (Lemma A.2, Défossez et al. (2022)) for the proof. □

**Lemma 4.** *If $\beta_2 \geq \beta_1$, then we have*

$$\frac{\|\boldsymbol{m}_t\|^2}{(\sqrt{\boldsymbol{\nu}_t})^3} \leq 4(1-\beta_1)\left(\sum_{s=1}^t \sqrt[4]{\beta_1^{t-s}}\frac{2}{1-\beta_2}\left(\frac{1}{\sqrt{\beta_2\boldsymbol{\nu}_{s-1}}} - \frac{1}{\sqrt{\boldsymbol{\nu}_s}}\right)\right).$$

*Proof.* To begin with, we have

$$\frac{\|\boldsymbol{m}_t\|}{\sqrt[4]{\boldsymbol{\nu}_t^3}} \leq (1-\beta_1)\sum_{s=1}^t \frac{\beta_2^{t-s}\|\boldsymbol{g}_s\|}{\sqrt[4]{\boldsymbol{\nu}_t^3}} \leq (1-\beta_1)\sum_{s=1}^t \frac{\beta_1^{t-s}\|\boldsymbol{g}_s\|}{\sqrt[4]{\beta_2^{3(t-s)}}\sqrt[4]{\boldsymbol{\nu}_s^3}}.$$

Here in the last inequality we use $\boldsymbol{\nu}_t \geq \beta_2^{t-s}\boldsymbol{\nu}_s$.

By further applying Cauchy-Schwartz inequality, we obtain

$$\frac{\|\boldsymbol{m}_t\|^2}{\sqrt{\boldsymbol{\nu}_t^3}} \leq (1-\beta_1)^2\left(\sum_{s=1}^t \frac{\beta_1^{t-s}\|\boldsymbol{g}_s\|^2}{\sqrt[4]{\beta_2^{3(t-s)}}\sqrt{\boldsymbol{\nu}_s^3}}\right)\left(\sum_{s=1}^t \frac{\beta_1^{t-s}}{\sqrt[4]{\beta_2^{3(t-s)}}}\right)$$

$$\leq \frac{(1-\beta_1)^2}{1-\frac{\beta_1}{\sqrt[4]{\beta_2^3}}}\left(\sum_{s=1}^t \frac{\beta_1^{t-s}\|\boldsymbol{g}_s\|^2}{\sqrt[4]{\beta_2^{3(t-s)}}\sqrt{\boldsymbol{\nu}_s^3}}\right)$$

$$\leq 4(1-\beta_1)\left(\sum_{s=1}^t \frac{\beta_1^{t-s}\|\boldsymbol{g}_s\|^2}{\sqrt[4]{\beta_2^{3(t-s)}}\sqrt{\boldsymbol{\nu}_s^3}}\right).$$

As $\frac{\|\boldsymbol{g}_s\|^2}{\sqrt{\boldsymbol{\nu}_s^3}} \leq \frac{2\|\boldsymbol{g}_s\|^2}{\sqrt{\boldsymbol{\nu}_s}\sqrt{\beta_2\boldsymbol{\nu}_{s-1}}(\sqrt{\boldsymbol{\nu}_s}+\sqrt{\beta_2\boldsymbol{\nu}_{s-1}})} = \frac{2}{1-\beta_2}\left(\frac{1}{\sqrt{\beta_2\boldsymbol{\nu}_{s-1}}} - \frac{1}{\sqrt{\boldsymbol{\nu}_s}}\right)$, the proof is completed. $\square$

**Lemma 5.** *Under the same set of assumptions in Theorem 11, if $\beta_2 \geq \beta_1$, then we have*

$$\frac{\|\boldsymbol{m}_t\|^2\|\boldsymbol{G}_t\|^2}{\boldsymbol{\nu}_t\sqrt{\beta_2\boldsymbol{\nu}_{t-1}}} \leq 4(1-\beta_1)\left(\sum_{s=1}^t \frac{\sqrt[8]{\beta_1^{t-s}}\|\boldsymbol{g}_s\|^2\|\boldsymbol{G}_s\|^2}{\boldsymbol{\nu}_s\sqrt{\beta_2\boldsymbol{\nu}_{s-1}}}\right) + 8\frac{1-\beta_1}{1-\beta_2}\frac{L_1^2}{L_0^2}\left(\sum_{s=1}^t \sqrt[8]{\beta_1^{t-s}}\left(\frac{1}{\sqrt{\beta_2\boldsymbol{\nu}_{s-1}}} - \frac{1}{\sqrt{\boldsymbol{\nu}_s}}\right)\right).$$

*Proof.* Similar to the proof of Lemma 4, we have

$$\frac{\|\boldsymbol{m}_t\|^2}{\sqrt{\beta_2\boldsymbol{\nu}_{t-1}}\boldsymbol{\nu}_t} \leq 4(1-\beta_1)\left(\sum_{s=1}^t \frac{\beta_1^{t-s}\|\boldsymbol{g}_s\|^2}{\sqrt[4]{\beta_2^{3(t-s)}}\sqrt{\beta_2\boldsymbol{\nu}_{s-1}}\boldsymbol{\nu}_s}\right). \tag{1}$$

Meanwhile, according to Assumption 1, we have

$$\|\boldsymbol{G}_t\|^2 \leq \|\boldsymbol{G}_{t-1}\|^2 + 2\|\boldsymbol{G}_{t-1}\|\|\boldsymbol{G}_t - \boldsymbol{G}_{t-1}\| + \|\boldsymbol{G}_t - \boldsymbol{G}_{t-1}\|^2$$

$$\leq \|\boldsymbol{G}_{t-1}\|^2 + 2\|\boldsymbol{G}_{t-1}\|(L_0 + L_1\|\boldsymbol{G}_{t-1}\|)\|\boldsymbol{w}_{t+1} - \boldsymbol{w}_t\| + 2(L_0^2 + L_1^2\|\boldsymbol{G}_{t-1}\|^2)\|\boldsymbol{w}_{t+1} - \boldsymbol{w}_t\|^2$$

$$\leq \|\boldsymbol{G}_{t-1}\|^2 + \frac{1-\sqrt[8]{\beta_1}}{3\sqrt[8]{\beta_1}}\|\boldsymbol{G}_{t-1}\|^2 + \frac{3\sqrt[8]{\beta_1}L_0^2}{1-\sqrt[8]{\beta_1}}\|\boldsymbol{w}_{t+1} - \boldsymbol{w}_t\|^2 + 2L_1\|\boldsymbol{G}_{t-1}\|^2\|\boldsymbol{w}_{t+1} - \boldsymbol{w}_t\|$$

$$\quad + 2(L_0^2 + L_1^2\|\boldsymbol{G}_{t-1}\|^2)\|\boldsymbol{w}_{t+1} - \boldsymbol{w}_t\|^2$$

$$\overset{(\star)}{\leq} \|\boldsymbol{G}_{t-1}\|^2 + \frac{1-\sqrt[8]{\beta_1}}{3\sqrt[8]{\beta_1}}\|\boldsymbol{G}_{t-1}\|^2 + \frac{1-\sqrt[8]{\beta_1}}{2}\frac{L_0^2}{L_1^2} + \frac{1-\sqrt[8]{\beta_1}}{3\sqrt[8]{\beta_1}}\|\boldsymbol{G}_{t-1}\|^2$$

$$\quad + \frac{1-\sqrt[8]{\beta_1}}{2}\frac{L_0^2}{L_1^2} + \frac{1-\sqrt[8]{\beta_1}}{3\sqrt[8]{\beta_1}}\|\boldsymbol{G}_{t-1}\|^2$$

$$\leq \frac{1}{\sqrt[8]{\beta_1}}\|\boldsymbol{G}_{t-1}\|^2 + (1-\sqrt[8]{\beta_1})\frac{L_1^2}{L_0^2}.$$

Here inequality $(\star)$ is because $\|\boldsymbol{w}_{t+1} - \boldsymbol{w}_t\| \leq \frac{1-\sqrt[8]{\beta_1}}{6L_1}$ (According to Lemma 1 and the choice of $\eta$ and $\beta_2$ in Theorem 11, we have $|w_{t+1} - w_t| \leq \frac{(1-\beta_1)\sqrt{1-\beta_1}}{256\sigma_1^2 L_1}$, and to prove the conclusion, we need to

show that $\frac{1-\sqrt[8]{\beta_1}}{6} \geq \frac{(1-\beta_1)\sqrt{1-\beta_1}}{256\sigma_1^2}$. Since $(1-\sqrt[8]{\beta_1})(1+\sqrt[8]{\beta_1})(1+\sqrt[4]{\beta_1})(1+\sqrt[2]{\beta_1}) = (1-\beta_1)$, and $(1+\sqrt[8]{\beta_1})(1+\sqrt[4]{\beta_1})(1+\sqrt[2]{\beta_1}) \leq 2 \times 2 \times 2 \leq 8$, it follows that $\frac{1-\sqrt[8]{\beta_1}}{6} \geq \frac{1-\beta_1}{48} \geq \frac{(1-\beta_1)\sqrt{1-\beta_1}}{256\sigma_1^2}$. Thus, the claim is proven).

Recursively applying the above inequality, we obtain that

$$\|\boldsymbol{G}_t\|^2 \leq \frac{1}{\sqrt[8]{\beta_1^{t-s}}}\|\boldsymbol{G}_s\|^2 + \left(\left(\frac{1}{\sqrt[8]{\beta_1}}\right)^{t-s} - 1\right)\frac{L_1^2}{L_0^2},$$

which by Eq. (1) further gives

$$\frac{\|\boldsymbol{m}_t\|^2\|\boldsymbol{G}_t\|^2}{\sqrt{\beta_2\boldsymbol{\nu}_{t-1}}\boldsymbol{\nu}_t} \leq 4(1-\beta_1)\left(\sum_{s=1}^{t}\frac{\beta_1^{t-s}\|\boldsymbol{g}_s\|^2\|\boldsymbol{G}_t\|^2}{\sqrt[4]{\beta_2^{3(t-s)}}\boldsymbol{\nu}_s\sqrt{\beta_2\boldsymbol{\nu}_{s-1}}}\right)$$

$$\leq 4(1-\beta_1)\left(\sum_{s=1}^{t}\frac{\sqrt[8]{\beta_1^{t-s}}\|\boldsymbol{g}_s\|^2\|\boldsymbol{G}_s\|^2}{\boldsymbol{\nu}_s\sqrt{\beta_2\boldsymbol{\nu}_{s-1}}} + \sum_{s=1}^{t}\frac{\sqrt[8]{\beta_1^{t-s}}\|\boldsymbol{g}_s\|^2}{\boldsymbol{\nu}_s\sqrt{\beta_2\boldsymbol{\nu}_{s-1}}}\frac{L_1^2}{L_0^2}\right)$$

$$\leq 4(1-\beta_1)\left(\sum_{s=1}^{t}\frac{\sqrt[8]{\beta_1^{t-s}}\|\boldsymbol{g}_s\|^2\|\boldsymbol{G}_s\|^2}{\boldsymbol{\nu}_s\sqrt{\beta_2\boldsymbol{\nu}_{s-1}}}\right) + 8\frac{1-\beta_1}{1-\beta_2}\frac{L_1^2}{L_0^2}\left(\sum_{s=1}^{t}\sqrt[8]{\beta_1^{t-s}}\left(\frac{1}{\sqrt{\beta_2\boldsymbol{\nu}_{s-1}}} - \frac{1}{\sqrt{\boldsymbol{\nu}_s}}\right)\right).$$

Here the last inequality is based on the similar reasoning of Lemma 5.

The proof is completed. $\qquad\square$

# B   PROOFS FOR DETERMINISTIC ALGORITHMS

## B.1   PROOF FOR DETERMINISTIC ADAM

We will first provide the formal statement of Theorem 1 [3], and then show the corresponding proof.

**Theorem 6** (Theorem 1, restated)**.** *Let Assumption 1 hold. Then,* $\forall\beta_1, \beta_2$ *satisfying* $0 \leq \beta_1^2 < \beta_2 < 1$,

*if* $T > 16L_1^2L_0(f(w_0) - f^*)/(1-\beta_2)$, *picking* $\eta = \frac{\sqrt{f(\boldsymbol{w}_1)-f^*}\sqrt{1-\frac{\beta_1^2}{\beta_2}}}{\sqrt{TL_0}(1-\beta_1)}$, *we have*

$$\frac{1}{T}\sum_{t=1}^{T}\|\nabla f(\boldsymbol{w}_t)\| \leq \frac{64}{(1-\beta_2)(1-\frac{\beta_1^2}{\beta_2})\left(1-\frac{\beta_1}{\sqrt[4]{\beta_2}}\right)^2}\left(\frac{\sqrt{L_0(f(\boldsymbol{w}_1)-f^*)}}{\sqrt{T}}\right).$$

*Proof.* To begin with, according to Lemma 1 and restriction on the value of $T$, we obtain that

$$\forall t \in \mathbb{N} \,\&t \geq 1, \|\boldsymbol{w}_{t+1} - \boldsymbol{w}_t\| \leq \frac{1}{4L_1}.$$

This is because by Lemma 1, $|w_{t+1} - w_t| \leq \eta(1-\beta_1)/(\sqrt{1-\beta_2}\sqrt{1-\beta_1^2/\beta_2})$, and by substituting the definition of $\eta$, we know $|w_{t+1} - w_t| \leq \sqrt{f(w_0) - f^*}/(\sqrt{1-\beta_2}\sqrt{TL_0})$. Finally, by substituting the requirement for $T$, we confirm the conclusion holds.

Therefore, the descent lemma can then be applied and thus $\forall t \in \mathbb{N}\&t \geq 1$,

$$f(\boldsymbol{w}_{t+1}) \leq f(\boldsymbol{w}_t) \underbrace{-\eta\left\langle\boldsymbol{G}_t, \frac{\boldsymbol{m}_t}{\lambda + \sqrt{\boldsymbol{\nu}_t}}\right\rangle}_{\text{First Order}} + \underbrace{\eta^2\frac{L_0 + L_1\|\boldsymbol{G}_t\|}{2}\frac{\|\boldsymbol{m}_t\|^2}{(\lambda + \sqrt{\boldsymbol{\nu}_t})^2}}_{\text{Second Order}}.$$

---

[3]In the theorem below and other theorems in this paper afterward, without loss of generality, we analyze the norm version of Adam, i.e., Adam with scalar adaptive learning rate, for a more readable proof. The extension to the coordinate-wise Adam can be easily done, as evidenced by literature such as Xing et al. (2021); Faw et al. (2022; 2023); Wang et al. (2023b)

To begin with, as for the "First Order" term, acording to $m_t = \beta_1 m_{t-1} + (1 - \beta_1) G_t$ we have that

$$-\eta \left\langle G_t, \frac{m_t}{\lambda + \sqrt{\nu_t}} \right\rangle = -\eta \frac{1}{1 - \beta_1} \left\langle m_t, \frac{m_t}{\lambda + \sqrt{\nu_t}} \right\rangle + \eta \frac{\beta_1}{1 - \beta_1} \left\langle m_{t-1}, \frac{m_t}{\lambda + \sqrt{\nu_t}} \right\rangle$$

$$\overset{(\star)}{\leq} -\eta \frac{1}{1 - \beta_1} \frac{\|m_t\|^2}{\lambda + \sqrt{\nu_t}} + \eta \frac{\beta_1}{(1 - \beta_1) \sqrt[4]{\beta_2}} \left\langle m_{t-1}, \frac{m_t}{\sqrt{\lambda + \sqrt{\nu_t}} \sqrt{\lambda + \sqrt{\nu_{t-1}}}} \right\rangle$$

$$\overset{(*)}{\leq} -\eta \frac{1}{1 - \beta_1} \frac{\|m_t\|^2}{\lambda + \sqrt{\nu_t}} + \frac{\beta_1}{2(1 - \beta_1) \sqrt[4]{\beta_2}} \eta \frac{\|m_t\|^2}{\lambda + \sqrt{\nu_t}} + \frac{\beta_1}{2(1 - \beta_1) \sqrt[4]{\beta_2}} \eta \frac{\|m_{t-1}\|^2}{\lambda + \sqrt{\nu_{t-1}}}$$

$$= -\eta \frac{1 - \frac{\beta_1}{\sqrt[4]{\beta_2}}}{1 - \beta_1} \frac{\|m_t\|^2}{\lambda + \sqrt{\nu_t}} - \frac{\beta_1}{2(1 - \beta_1) \sqrt[4]{\beta_2}} \eta \frac{\|m_t\|^2}{\lambda + \sqrt{\nu_t}} + \frac{\beta_1}{2(1 - \beta_1) \sqrt[4]{\beta_2}} \eta \frac{\|m_{t-1}\|^2}{\lambda + \sqrt{\nu_{t-1}}}.$$

where inequality $(\star)$ is due to that $\sqrt{\nu_t} \geq \sqrt{\beta_2 \nu_{t-1}}$ and inequality $(*)$ is due to Young's inequality.

Meanwhile, as for the "Second Order" term, we have

$$\eta^2 \frac{L_0 + L_1 \|G_t\|}{2} \frac{\|m_t\|^2}{(\lambda + \sqrt{\nu_t})^2} \overset{(\bullet)}{\leq} L_0 \eta^2 \frac{(1 - \beta_1)^2}{(1 - \beta_2)(1 - \frac{\beta_1^2}{\beta_2})} + \frac{L_1 \eta^2}{\sqrt{1 - \beta_2}} \frac{\|m_t\|^2}{\lambda + \sqrt{\nu_t}}$$

$$\overset{(\circ)}{\leq} L_0 \eta^2 \frac{(1 - \beta_1)^2}{(1 - \beta_2)(1 - \frac{\beta_1^2}{\beta_2})} + \frac{\eta}{2} \frac{1 - \frac{\beta_1}{\sqrt[4]{\beta_2}}}{1 - \beta_1} \frac{\|m_t\|^2}{\lambda + \sqrt{\nu_t}}.$$

Here inequality $(\bullet)$ is due to Lemma 1 and

$$\nu_t \geq (1 - \beta_2) \|G_t\|^2,$$

and inequality $(\circ)$ is due to the requirement over $T$.

Applying the estimations of both the "First Order" and the "Second Order" terms, we obtain that

$$f(w_{t+1}) - f(w_t) \leq -\frac{\eta}{2} \frac{1 - \frac{\beta_1}{\sqrt[4]{\beta_2}}}{1 - \beta_1} \frac{\|m_t\|^2}{\lambda + \sqrt{\nu_t}} - \frac{\beta_1}{2(1 - \beta_1) \sqrt[4]{\beta_2}} \eta \frac{\|m_t\|^2}{\lambda + \sqrt{\nu_t}} + \frac{\beta_1}{2(1 - \beta_1) \sqrt[4]{\beta_2}} \eta \frac{\|m_{t-1}\|^2}{\lambda + \sqrt{\nu_{t-1}}}$$

$$+ L_0 \eta^2 \frac{(1 - \beta_1)^2}{(1 - \beta_2)(1 - \frac{\beta_1^2}{\beta_2})}.$$

Summing the above inequality over $t \in \{1, \cdots, T\}$ then gives

$$\sum_{t=1}^{T} \frac{\eta}{2} \frac{1 - \frac{\beta_1}{\sqrt[4]{\beta_2}}}{1 - \beta_1} \frac{\|m_t\|^2}{\lambda + \sqrt{\nu_t}}$$

$$\leq f(w_1) - f(w_{T+1}) - \frac{\beta_1}{2(1 - \beta_1) \sqrt[4]{\beta_2}} \eta \frac{\|m_T\|^2}{\lambda + \sqrt{\nu_T}} + T L_0 \eta^2 \frac{(1 - \beta_1)^2}{(1 - \beta_2)(1 - \frac{\beta_1^2}{\beta_2})} \qquad (2)$$

$$\leq f(w_1) - f(w_{T+1}) + T L_0 \eta^2 \frac{(1 - \beta_1)^2}{(1 - \beta_2)(1 - \frac{\beta_1^2}{\beta_2})}.$$

Furthermore, as $(1 - \beta_1) G_t = m_t - \beta_1 m_{t-1}$, we have that

$$\|G_t\|^2 \leq \frac{1}{(1 - \beta_1)^2} \|m_t\|^2 + \frac{1}{(1 - \beta_1)^2} \|m_{t-1}\|^2.$$

Applying the above inequality and $\lambda = 0$ to Eq. (2), we obtain that

$$\sum_{t=1}^{T} \frac{\eta}{4} \left(1 - \frac{\beta_1}{\sqrt[4]{\beta_2}}\right) (1 - \beta_1) \frac{\|G_t\|^2}{\sqrt{\nu_t}} \leq f(w_1) - f(w_{T+1}) + T L_0 \eta^2 \frac{(1 - \beta_1)^2}{(1 - \beta_2)(1 - \frac{\beta_1^2}{\beta_2})}.$$

Meanwhile, we have

$$\sqrt{\nu_t} - \sqrt{\beta_2 \nu_{t-1}} = \frac{(1 - \beta_2) \|G_t\|^2}{\sqrt{\nu_t} + \sqrt{\beta_2 \nu_{t-1}}} \leq (1 - \beta_2) \frac{\|G_t\|^2}{\sqrt{\nu_t}}.$$

Therefore, applying the above inequality and dividing both sides by $\eta$, we have

$$\frac{1}{4}\left(1 - \frac{\beta_1}{\sqrt[4]{\beta_2}}\right)(1 - \beta_1)\sum_{t=1}^{T}(\sqrt{\boldsymbol{\nu}_t} - \sqrt{\beta_2\boldsymbol{\nu}_{t-1}}) \leq \frac{f(\boldsymbol{w}_1) - f(\boldsymbol{w}_{T+1})}{\eta} + TL_0\eta\frac{(1 - \beta_1)^2}{(1 - \beta_2)(1 - \frac{\beta_1^2}{\beta_2})},$$

which by telescoping further leads to

$$\frac{1}{4}\left(1 - \frac{\beta_1}{\sqrt[4]{\beta_2}}\right)(1 - \beta_1)\sum_{t=1}^{T}(1 - \beta_2)\sqrt{\boldsymbol{\nu}_t} \leq \frac{f(\boldsymbol{w}_1) - f(\boldsymbol{w}_{T+1})}{\eta} + TL_0\eta\frac{(1 - \beta_1)^2}{(1 - \beta_2)(1 - \frac{\beta_1^2}{\beta_2})}.$$

According to Cauchy-Schwartz's inequality, we then obtain

$$\left(\sum_{t=1}^{T}\|\boldsymbol{G}_t\|\right)^2 \leq \left(\sum_{t=1}^{T}\sqrt{\boldsymbol{\nu}_t}\right)\left(\sum_{t=1}^{T}\frac{\|\boldsymbol{G}_t\|^2}{\sqrt{\boldsymbol{\nu}_t}}\right)$$

$$\leq \frac{1}{1 - \beta_2}\left(\frac{4(f(\boldsymbol{w}_1) - f(\boldsymbol{w}_{T+1}))}{\eta\left(1 - \frac{\beta_1}{\sqrt[4]{\beta_2}}\right)(1 - \beta_1)} + TL_0\eta\frac{(1 - \beta_1)}{(1 - \beta_2)\left(1 - \frac{\beta_1}{\sqrt[4]{\beta_2}}\right)(1 - \frac{\beta_1^2}{\beta_2})}\right)^2$$

$$= \frac{1}{1 - \beta_2}\left(\frac{4(f(\boldsymbol{w}_1) - f(\boldsymbol{w}_{T+1}))}{\eta\left(1 - \frac{\beta_1}{\sqrt[4]{\beta_2}}\right)(1 - \beta_1)} + 4TL_0\eta\frac{(1 - \beta_1)}{(1 - \beta_2)\left(1 - \frac{\beta_1}{\sqrt[4]{\beta_2}}\right)(1 - \frac{\beta_1^2}{\beta_2})}\right)^2.$$

The proof is completed by applying the value of $\eta$. $\qquad\square$

### B.2 PROOF FOR GDM

This section collects the proof of Theorem 2. To begin with, given problem hyperparameters $\Delta_1$, $\varepsilon$, $L_0$, and $L_1$. We first construct three 1D functions as follows:

$$f_1(x) = \begin{cases} \dfrac{L_0 e^{L_1 x - 1}}{L_1^2} & , x \in \left[\dfrac{1}{L_1}, \infty\right), \\[2mm] \dfrac{L_0 x^2}{2} + \dfrac{L_0}{2L_1^2} & , x \in [-\dfrac{1}{L_1}, \dfrac{1}{L_1}], \\[2mm] \dfrac{L_0 e^{-L_1 x - 1}}{L_1^2} & , x \in \left(-\infty, -\dfrac{1}{L_1}\right]. \end{cases} \tag{3}$$

$$f_2(y) = \begin{cases} \varepsilon(y - 1) + \dfrac{\varepsilon}{2} & , y \in [1, \infty), \\[2mm] \dfrac{\varepsilon}{2} y^2 & , y \in [-1, 1], \\[2mm] -\varepsilon(y + 1) + \dfrac{\varepsilon}{2} & , y \in (-\infty, -1]. \end{cases} \tag{4}$$

$$f_3(z) = \begin{cases} \varepsilon(z - 1) + \dfrac{\varepsilon}{2L_1} + \dfrac{L_0}{2L_1^2} & , z \in [\dfrac{1}{L_1}, \infty), \\[2mm] \dfrac{\varepsilon L_1}{2} z^2 + \dfrac{L_0}{2L_1^2} & , z \in [0, \dfrac{1}{L_1}], \\[2mm] \dfrac{L_0 z^2}{2} + \dfrac{L_0}{2L_1^2} & , z \in [-\dfrac{1}{L_1}, 0], \\[2mm] \dfrac{L_0 e^{-L_1 z - 1}}{L_1^2} & , z \in \left(-\infty, -\dfrac{1}{L_1}\right]. \end{cases} \tag{5}$$

It is easy to verify that these functions satisfy $(L_0, L_1)$-smooth condition as long as $\varepsilon \leq L_0$. We then respectively the convergence of GDM over these three examples with different learning rate and momentum coefficient.

**Lemma 6** (Convergence over $f_1$). *Assume* $\Delta_1 \geq \frac{L_0}{L_1^2}(e - \frac{1}{2})$, $\varepsilon \leq 1$ ,*and let* $x_1 = \frac{1 + \log(\frac{1}{2} + \frac{L_1^2}{L_0}\Delta_1)}{L_1}$.

*Then, we have* $f_1(x_1) - f_1^* = \Delta_1$, *and if* $\eta \geq \frac{(5 + 8\log\frac{1}{\varepsilon})(1 + \log(\frac{1}{2} + \frac{L_1^2}{L_0}\Delta_1))}{L_1^2(\Delta_1 + \frac{L_0}{2L_1^2})}$ *and* $\beta \leq 1 - 2\left(\frac{L_1^2}{L_0}e\right)^{-4\log\frac{1}{\varepsilon} - 2}(\Delta_1 + \frac{L_0}{2L_1^2})^{-4\log\frac{1}{\varepsilon} - 2}$, *we have that GDM satisfies that* $\forall t \in [1, \infty)$, $|f_1'(x_t)| \geq L_1\Delta_1$.

*Proof.* We prove this lemma by proving that $\forall k \geq 1$, $|x_{k+1}| \geq (4 + 8\log\frac{1}{\varepsilon})|x_k|$ and $\text{Sign}(x_{k+1}) = (-1)^{k+1}$ by induction. When $k = 1$, according to the update rule of GDM, we have

$$x_2 = x_1 - \eta f_1'(x_1).$$

As $\eta \geq \frac{(5 + 8\log\frac{1}{\varepsilon})(1 + \log(\frac{1}{2} + \frac{L_1^2}{L_0}\Delta_1))}{\Delta_1 + \frac{L_0}{2L_1^2}} = -\frac{(5 + 8\log\frac{1}{\varepsilon})x_1}{f_1'(x_1)}$, we have

$$x_2 \leq -(4 + 8\log\frac{1}{\varepsilon})x_1,$$

which leads to the claim.

Now assuming that the claim has been proved for $k \leq t - 1$ ($t \geq 2$). Then, for $k = t$, with induction hypothesis we have

$$x_{t+1} = x_t - \eta m_t = x_t - \eta\left(\beta^t f_1'(x_1) + (1 - \beta)\sum_{s=1}^{t-1}\beta^{t-s}f_1'(x_s) + (1 - \beta)f_1'(x_t)\right).$$

Without the loss of generality, we assume $t$ is even. By the induction hypothesis, we obtain that $f_1'(x_t) < 0$ and $f_1'(x_{t-1}) < 0$, and

$$|f_1'(x_1)| \leq |f_1'(x_2)| \leq \cdots \leq |f_1'(x_{t-1})|.$$

Therefore, we have

$$\begin{aligned}
x_{t+1} &\geq x_t - \eta\left(\beta f_1'(x_{t-1}) + (1 - \beta)f_1'(x_t)\right) \\
&= x_t - \frac{L_0}{L_1}\eta\left(\beta e^{L_1 x_{t-1} - 1} - (1 - \beta)e^{-L_1 x_t - 1}\right) \\
&\geq x_t - \frac{L_0}{L_1}\eta\left(\beta e^{-\frac{L_1 x_t}{8\log\frac{1}{\varepsilon} + 4} - 1} - (1 - \beta)e^{-L_1 x_t - 1}\right).
\end{aligned}$$

Furthermore, according to the definition of $x_1$, we have

$$1 - \beta \geq 2e^{-L_1(4\log\frac{1}{\varepsilon} + 2)x_1} \geq 2e^{\frac{L_1 x_t}{2}},$$

which leads to

$$x_{t+1} \geq x_t + \frac{L_0}{L_1}\eta e^{-\frac{L_1 x_t}{2} - 1} \geq x_t + \frac{(5 + 8\log\frac{1}{\varepsilon})x_1}{e^{L_1 x_1}}e^{-\frac{L_1 x_t}{2}} \geq x_t + \frac{(5 + 8\log\frac{1}{\varepsilon})x_1}{e^{L_1 x_1}}e^{L_1 x_t(2 + 4\log\frac{1}{\varepsilon})}.$$

Then, as $\frac{e^{\frac{L_1 x}{2}}}{x}$ is monotonously increasing for $x \in [\frac{2}{L_1}, \infty)$, and $x_1 \geq \frac{2}{L_1}$, we have

$$x_{t+1} \geq x_t + \frac{(5 + 8\log\frac{1}{\varepsilon})x_1}{e^{L_1 x_1}}e^{L_1 x_t(1 + 2\log\frac{1}{\varepsilon})} \geq x_t - (5 + 8\log\frac{1}{\varepsilon})x_t \geq -(4 + 8\log\frac{1}{\varepsilon})x_t.$$

The proof is completed. $\qquad\square$

**Lemma 7** (Convergence over $f_2$). *Assume that* $\Delta_1 \geq \frac{\varepsilon}{2} + \frac{L_1}{L_0}$, *and let* $y_1 \triangleq \frac{\Delta_1}{\varepsilon} + \frac{1}{2}$. *Then, if* $\eta \leq \frac{(5 + 8\log\frac{1}{\varepsilon})(1 + \log(\frac{1}{2} + \frac{L_1^2}{L_0}\Delta_1))}{L_1^2(\Delta_1 + \frac{L_0}{2L_1^2})}$, *we have that GDM satisfies that* $\|\nabla f_2(y_t)\| \geq \varepsilon$ *if* $T \leq \tilde{\Theta}(\frac{L_1^2\Delta_1^2 + L_0\Delta_1}{\varepsilon^2})$.

*Proof.* We have that $m_t = \varepsilon$ before $y_t$ enters the region $(-\infty, 1]$. As the movement of each step before $y_t$ enters the region $(-\infty, 1]$ is $\eta\varepsilon$ and the total length to enter $(-\infty, 1]$ is $y_1 - 1$, the proof is completed. □

**Lemma 8** (Convergence over $f_3$). *Assume* $\Delta_1 \geq \frac{L_0}{L_1^2}e + 4e + \frac{L_0^2}{e^2 L_1^2}$, $L_1 \geq 1$, $\varepsilon \leq \frac{1}{2}$, *and let*

$$z_1 = -\frac{1 + \log(\frac{1}{2} + \frac{L_1^2}{L_0}\Delta_1)}{L_1}. \text{ Then, we have } f_3(z_1) - f_3^* = \Delta_1, \text{ and if } \eta \geq \frac{(5 + 8\log\frac{1}{\varepsilon})(1 + \log(\frac{1}{2} + \frac{L_1^2}{L_0}\Delta_1))}{L_1^2(\Delta_1 + \frac{L_0}{2L_1^2})}$$

*and* $\beta \geq 1 - 2\left(\frac{L_1^2}{L_0}e\right)^{-4\log\frac{1}{\varepsilon} - 2}(\Delta_1 + \frac{L_0}{2L_1^2})^{-4\log\frac{1}{\varepsilon} - 2}$, *we have that GDM satisfies that* $\forall t \in [1, \Theta(\frac{L_1^2 \Delta_1^2}{\varepsilon^3})), |f_3'(x_t)| \geq \varepsilon$.

*Proof.* To begin with, according to the definition of $z_1$, we have $\eta \geq -\frac{(5 + 8\log\frac{1}{\varepsilon})z_1}{f_3'(x_1)}$ and $1 - \beta \geq 2e^{L_1(4\log\frac{1}{\varepsilon} + 2)z_1} \geq \frac{1}{2}$. Also. as $\Delta_1 \geq \frac{L_0}{L_1^2}(e - \frac{1}{2})$, we have $z_1 \leq -\frac{2}{L_1}$, and thus

$$f_3'(z_1) = -\frac{L_0}{L_1}e^{-L_1 z_1 - 1} \leq -L_1\left(\Delta_1 + \frac{L_0}{2L_1^2}\right) \leq -4.$$

We will first prove the following claim by induction: for $k \in [2, \lfloor\frac{1}{1-\beta}\rfloor]$, we have $z_k \geq \frac{1}{L_1}$, and $m_k \leq \frac{\beta^{k-1}f_3'(z_1)}{2}$.

As for $k = 2$, we have

$$z_2 = z_1 - \eta f_3'(z_1) \geq -\left(4 + 8\log\frac{1}{\varepsilon}\right)z_1.$$

According to $\Delta_1 \geq \frac{L_0}{L_1^2}(e - \frac{1}{2})$, we have $z_1 \leq -\frac{2}{L_1}$, and thus $z_2 \geq \frac{1}{L_1}$. Since $m_2 = \beta f'(z_1) + (1 - \beta)\varepsilon < \frac{f_3'(z_1)}{2}$, the claim is proved for $k = 2$.

Now assuming that we have prove the claim for $k \leq t - 1$. According to the induction hypothesis, we have

$$f_3'(z_2) = \cdots = f_3'(z_{t-1}) = \varepsilon,$$

and thus

$$m_t = \beta^{t-1}f_3'(z_1) + (1 - \beta^{t-1})\varepsilon \overset{(\star)}{\leq} \beta^{t-1}f_3'(z_1) - \frac{\beta^{t-1}f_3'(z_1)}{2} \leq \frac{\beta^{t-1}f_3'(z_1)}{2}.$$

Here inequality $(\star)$ is due to $\beta^{\lfloor\frac{1}{1-\beta}\rfloor} \geq \frac{1}{4}$ as $\beta \geq \frac{1}{2}$. Therefore, as $z_t = z_{t-1} - \eta m_t \geq z_{t-1} \geq \frac{1}{L_1}$, we prove the claim.

It should be noticed that $\forall t \in [1, \lfloor\frac{1}{1-\beta}\rfloor], \|f_3'(z_t)\| \geq \varepsilon$. Furthermore, according to the claim, $z_{\lfloor\frac{1}{1-\beta}\rfloor + 1}$ can now be bounded as

$$z_{\lfloor\frac{1}{1-\beta}\rfloor + 1} = z_1 - \eta\sum_{k=1}^{\lfloor\frac{1}{1-\beta}\rfloor} m_t \geq \frac{\eta}{5 + 8\log\frac{1}{\varepsilon}}f_3'(z_1) - \eta\sum_{k=1}^{\lfloor\frac{1}{1-\beta}\rfloor}\frac{\beta^{k-1}f_3'(z_1)}{2} \geq \frac{\eta}{5 + 8\log\frac{1}{\varepsilon}}f_3'(z_1) - \eta\frac{1 - \frac{1}{e}}{(1-\beta)}\frac{f_3'(z_1)}{2}$$

$$\geq \frac{1}{L_1} - \eta\frac{1 - \frac{1}{e}}{(1-\beta)}\frac{f_3'(z_1)}{4} \geq \frac{1}{L_1} - \eta\left(1 - \frac{1}{e}\right)\frac{f_3'(z_1)}{8}\left(\frac{L_1^2}{L_0}e\right)^{4\log\frac{1}{\varepsilon} + 2}\left(\Delta_1 + \frac{L_0}{2L_1^2}\right)^{4\log\frac{1}{\varepsilon} + 2}$$

$$\geq \frac{1}{L_1} + \frac{\eta}{16}\frac{L_1^2\Delta_1^2 + L_0\Delta_1}{\varepsilon^2}.$$

As $f_3'(z) = \varepsilon$ for all $z \geq \frac{1}{L_1}$, the iterates needs additional $\frac{\frac{\eta}{16}\frac{L_1^2\Delta_1^2}{\varepsilon^2}}{\eta\varepsilon} = \frac{1}{16}\frac{L_1^2\Delta_1^2}{\varepsilon^3}$ steps to make $f_3'(z_t) < \varepsilon$. The proof is completed. □

**Lemma 9.** *Let* $f_1, f_2, f_3 : \mathcal{R} \to \mathcal{R}$ *satisfies* $(L_0, L_1)$-*smooth condition. Then* $f_1(x) + f_2(y) + f_3(z)$ *satisfies* $(L_0, L_1)$-*smooth condition.*

*Proof.* If we consider a point $(x_1, y_1, z_1)$ within a ball centered at $(x_2, y_2, z_2)$ with radius $1/L_1$, it follows that $x_1$ is within a ball centered at $x_2$ with the same radius. Thus, we have:

$$|\nabla f_1(x_1) - \nabla f_1(x_2)| \leq (L_0 + L_1 |\nabla f_1(x_1)|)|x_1 - x_2| \leq (L_0 + L_1 |\nabla f(x_1, y_1, z_1)|)|x_1 - x_2|.$$

The last inequality holds because $\nabla f_1(x_1)$ is one coordinate of $\nabla f(x_1, y_1, z_1)$. Similarly, we can derive:

$$|\nabla f_2(y_1) - \nabla f_2(y_2)| \leq (L_0 + L_1 |\nabla f(x_1, y_1, z_1)|)|y_1 - y_2|,$$

$$|\nabla f_3(z_1) - \nabla f_3(z_2)| \leq (L_0 + L_1 |\nabla f(x_1, y_1, z_1)|)|z_1 - z_2|.$$

Taking the squared sum of these inequalities confirms that $f$ is indeed $(L_0, L_1)$-smooth. $\square$

**Theorem 7** (Theorem 2, restated). *Assume that $\Delta_1 \geq 4\frac{L_0}{L_1}e + 16e + 4\frac{L_0^2}{e^2 L_1^2}$, $L_1 \geq 1$ and $\varepsilon \leq 1$, then there exists objective function $f$ satisfying $(L_0, L_1)$-smooth condition and $f(\boldsymbol{w}_1) - f^* = \Delta_1$, such that **for any learning rate** $\eta > 0$ **and** $\beta \in [0, 1]$, the minimum step $T$ of GDM to achieve final error $\varepsilon$ satisfies*

$$T = \tilde{\Omega}\left(\frac{L_1^2 \Delta_1^2 + L_0 \Delta_1}{\varepsilon^2}\right).$$

*Proof.* Construct the objective function as $f(x, y, z, u) = f_1(x) + f_2(y) + f_3(z)$. Then, let $x_1$, $y_1$, $z_1$ be chosen as $f_1(x_1) - f_1^* = f_2(y_1) - f_2^* = f_3(z_1) - f_3^* = \frac{\Delta_1}{3}$ and $z_1 \leq 0$. According to Lemma 9, $f$ satisfies $(L_0, L_1)$-smooth condition. Then, for each learning rate and momentum coefficient, they will always be covered by one of the above lemmas, and applying the corresponding lemma gives the desired result.

The proof is completed. $\square$

## B.3 PROOF FOR DETERMINISTIC ADAGRAD

To begin with, we recall the following result from Wang et al. (2023b):

**Proposition 2.** *For every learning rate $\eta \geq \Theta(\frac{1}{L_1})$ and $\Delta_1$, there exist a lower-bounded objective function $g_1$ obeying Assumption 1 and a corresponding initialization point $\boldsymbol{w}_0$ with $g_1(\boldsymbol{w}_1) - g_1^* = \Delta_1$, such that AdaGrad with learning rate $\eta$ and initialized at $\boldsymbol{w}_0$ diverges over $g_1$.*

We then define $g_2$ as the $f_2$ in the proof of Theorem 2, i.e.,

$$g_2(y) = \begin{cases} \varepsilon(y-1) + \frac{\varepsilon}{2} & , y \in [1, \infty), \\ \frac{\varepsilon}{2}y^2 & , y \in [-1, 1], \\ -\varepsilon(y+1) + \frac{\varepsilon}{2} & , y \in (-\infty, -1]. \end{cases} \tag{6}$$

We then have the following lemma characterizing the convergence of AdaGrad over $g_2$.

**Lemma 10** (Convergence over $g_2$). *Assume that $\Delta_1 \geq \frac{\varepsilon}{2} + \frac{L_1}{L_0}$, and let $y_1 \triangleq \frac{\Delta_1}{\varepsilon} + \frac{1}{2}$. Then, if $\eta \leq \Theta(\frac{1}{L_1})$, we have that AdaGrad satisfies that $\|\nabla g_2(y_t)\| \geq \varepsilon$ if $T \leq \tilde{\Theta}(\frac{L_1^2 \Delta_1^2}{\varepsilon^2})$.*

*Proof.* We have that $\boldsymbol{g}_t = \varepsilon$ before $y_t$ enters the region $(-\infty, 1]$. Therefore, the sum of movement of each step before $y_t$ enters the region $(-\infty, 1]$ is

$$\eta \sum_{s=1}^{t} \frac{\varepsilon}{\sqrt{s\varepsilon}} = \eta\Theta(\sqrt{t}).$$

Solving $\eta\Theta(\sqrt{t}) = \frac{\Delta_1}{\varepsilon} + \frac{1}{2} - 1$ gives $t = \frac{L_1^2 \Delta_1^2}{\varepsilon^2}$, and the proof is completed. $\square$

We then have the following lower bound for deterministic AdaGrad.

**Theorem 8.** *Assume that $\Delta_1 \geq \frac{\varepsilon}{2} + \frac{L_1}{L_0}$. Then, there exists objective function $f$ satisfying $(L_0, L_1)$-smooth condition and $f(\boldsymbol{w}_1) - f^* = \Delta_1$, such that **for any learning rate** $\eta > 0$ **and** $\beta \in [0, 1]$, the minimum step $T$ of AdaGrad to achieve final error $\varepsilon$ satisfies*

$$T = \Omega\left(\frac{L_1^2 \Delta_1^2}{\varepsilon^2}\right).$$

*Proof.* The proof is completed by letting $f(x, y) = g_1(x) + g_2(y)$ following the same routine as Theorem 7. $\square$

## C  PROOF FOR STOCHASTIC ALGORITHMS

### C.1  PROOF FOR ADAM

To begin with, we restate the theorem as follows:

**Theorem 9** (Theorem 3, restated). *Let Assumptions 1 and 2 hold. Then, $\forall \beta_1 \geq 0$ and $\lambda = 0$, if $\varepsilon \leq \frac{1}{\mathrm{poly}(f(\boldsymbol{w}_1) - f^*, L_0, L_1, \sigma_0, \sigma_1)}$, with $\eta = \frac{\sqrt{f(\boldsymbol{w}_1) - f^*}}{\sqrt{L_0 + L_1}\sqrt{T\sigma_0 \sigma_1^2}}$ and momentum hyperparameter $\beta_2 = 1 - \eta^2 \left(\frac{1024\sigma_1^2(L_1 + L_0)(1 - \beta_1)}{\sqrt{1 - \frac{\beta_1^2}{\beta_2}(1 - \frac{\beta_1}{\sqrt{\beta_2}})}}\right)^2$, we have if $T \geq \Theta\left(\frac{(L_0 + L_1)\sigma_0^3 \sigma_1^2 (f(\boldsymbol{w}_1) - f^*)}{\varepsilon^4}\right)$, then Algorithm 1 satisfies*

$$\frac{1}{T}\mathbb{E}\sum_{t=1}^{T} \|\nabla f(\boldsymbol{w}_t)\| \leq \varepsilon.$$

*Proof.* Let the approximate iterative sequence be defined as $\boldsymbol{u}_t \triangleq \frac{\boldsymbol{w}_t - \frac{\beta_1}{\sqrt{\beta_2}}\boldsymbol{w}_{t-1}}{1 - \frac{\beta_1}{\sqrt{\beta_2}}}$ and the surrogate second-order momentum be defined as $\widetilde{\boldsymbol{\nu}}_t \triangleq \beta_2 \boldsymbol{\nu}_{t-1} + (1 - \beta_2)\sigma_0^2$. Then, as $\frac{\eta}{\sqrt{1 - \beta_2}} = \frac{\sqrt{1 - \frac{\beta_1^2}{\beta_2}(1 - \frac{\beta_1}{\sqrt{\beta_2}})}}{1024\sigma_1^2(L_1 + L_0)(1 - \beta_1)}$, we have

$$\|\boldsymbol{u}_t - \boldsymbol{w}_t\| = \frac{\frac{\beta_1}{\sqrt{\beta_2}}}{1 - \frac{\beta_1}{\sqrt{\beta_2}}}\|\boldsymbol{w}_t - \boldsymbol{w}_{t-1}\| \overset{(*)}{\leq} \eta \frac{\frac{\beta_1}{\sqrt{\beta_2}}}{1 - \frac{\beta_1}{\sqrt{\beta_2}}}\frac{1 - \beta_1}{\sqrt{1 - \beta_2}}\sqrt{1 - \frac{\beta_1^2}{\beta_2}} \leq \frac{1}{4L_1},$$

and

$$\|\boldsymbol{u}_{t+1} - \boldsymbol{w}_t\| = \frac{1}{1 - \frac{\beta_1}{\sqrt{\beta_2}}}\|\boldsymbol{w}_{t+1} - \boldsymbol{w}_t\| \overset{(*)}{\leq} \eta \frac{1}{1 - \frac{\beta_1}{\sqrt{\beta_2}}}\frac{1 - \beta_1}{\sqrt{1 - \beta_2}}\sqrt{1 - \frac{\beta_1^2}{\beta_2}} \leq \frac{1}{4L_1}.$$

Therefore, if choosing $\boldsymbol{w}^1 = \boldsymbol{w}_t$, $\boldsymbol{w}^2 = \boldsymbol{u}_{t+1}$, and $\boldsymbol{w}^3 = \boldsymbol{u}_t$ in Lemma 2, we see the conditions of Lemma 2 is satisfied, which after taking expectation gives

$$\mathbb{E}^{|\mathcal{F}_t} f(\boldsymbol{u}_{t+1}) \leq f(\boldsymbol{u}_t) + \mathbb{E}^{|\mathcal{F}_t}\langle \nabla f(\boldsymbol{w}_t), \boldsymbol{u}_{t+1} - \boldsymbol{u}_t\rangle + \frac{1}{2}(L_0 + L_1\|\nabla f(\boldsymbol{w}_t)\|)\mathbb{E}^{|\mathcal{F}_t}(\|\boldsymbol{u}_{t+1} - \boldsymbol{w}_t\| + \|\boldsymbol{u}_t - \boldsymbol{w}_t\|)\|\boldsymbol{u}_{t+1} - \boldsymbol{u}_t\|.$$

We call $\langle \nabla f(\boldsymbol{w}_t), \boldsymbol{u}_{t+1} - \boldsymbol{u}_t\rangle$ the first-order term and $\frac{1}{2}(L_0 + L_1\|\nabla f(\boldsymbol{w}_t)\|)(\|\boldsymbol{u}_{t+1} - \boldsymbol{w}_t\| + \|\boldsymbol{u}_t - \boldsymbol{w}_t\|)\|\boldsymbol{u}_{t+1} - \boldsymbol{u}_t\|$ the second-order term, as they respectively correspond to the first-order and second-order Taylor's expansion. We then respectively bound these two terms as follows.

**Analysis for the first-order term.** Before we start, denote $\widetilde{\boldsymbol{\nu}}_t \triangleq \beta_2 \boldsymbol{\nu}_{t-1} + (1-\beta_2)\sigma_0^2$

$$
\begin{aligned}
\boldsymbol{u}_{t+1} - \boldsymbol{u}_t =& \frac{\boldsymbol{w}_{t+1} - \boldsymbol{w}_t}{1 - \frac{\beta_1}{\sqrt{\beta_2}}} - \frac{\beta_1}{\sqrt{\beta_2}} \frac{\boldsymbol{w}_t - \boldsymbol{w}_{t-1}}{1 - \frac{\beta_1}{\sqrt{\beta_2}}} \\
=& - \frac{\eta}{1 - \frac{\beta_1}{\sqrt{\beta_2}}} \frac{1}{\sqrt{\boldsymbol{\nu}_t}} \boldsymbol{m}_t + \beta_1 \frac{\eta}{1 - \frac{\beta_1}{\sqrt{\beta_2}}} \frac{1}{\sqrt{\beta_2 \boldsymbol{\nu}_{t-1}}} \boldsymbol{m}_{t-1} \\
=& - \frac{\eta}{1 - \frac{\beta_1}{\sqrt{\beta_2}}} \frac{1}{\sqrt{\widetilde{\boldsymbol{\nu}}_t}} \boldsymbol{m}_t + \beta_1 \frac{\eta}{1 - \frac{\beta_1}{\sqrt{\beta_2}}} \frac{1}{\sqrt{\widetilde{\boldsymbol{\nu}}_t}} \boldsymbol{m}_{t-1} - \frac{\eta}{1 - \frac{\beta_1}{\sqrt{\beta_2}}} \left( \frac{1}{\sqrt{\boldsymbol{\nu}_t}} - \frac{1}{\sqrt{\widetilde{\boldsymbol{\nu}}_t}} \right) \boldsymbol{m}_t \\
& + \beta_1 \frac{\eta}{1 - \frac{\beta_1}{\sqrt{\beta_2}}} \left( \frac{1}{\sqrt{\beta_2 \boldsymbol{\nu}_{t-1}}} - \frac{1}{\sqrt{\widetilde{\boldsymbol{\nu}}_t}} \right) \boldsymbol{m}_{t-1} \\
=& - \eta \frac{1 - \beta_1}{1 - \frac{\beta_1}{\sqrt{\beta_2}}} \frac{1}{\sqrt{\widetilde{\boldsymbol{\nu}}_t}} \boldsymbol{g}_t - \frac{\eta}{1 - \frac{\beta_1}{\sqrt{\beta_2}}} \left( \frac{1}{\sqrt{\boldsymbol{\nu}_t}} - \frac{1}{\sqrt{\widetilde{\boldsymbol{\nu}}_t}} \right) \boldsymbol{m}_t + \beta_1 \frac{\eta}{1 - \frac{\beta_1}{\sqrt{\beta_2}}} \left( \frac{1}{\sqrt{\beta_2 \boldsymbol{\nu}_{t-1}}} - \frac{1}{\sqrt{\widetilde{\boldsymbol{\nu}}_t}} \right) \boldsymbol{m}_{t-1}.
\end{aligned}
$$

According to the above decomposition, we have the first-order term can also be decomposed into

$$
\begin{aligned}
& \mathbb{E}^{|\mathcal{F}_t} \left[ \langle \nabla f(\boldsymbol{w}_t), \boldsymbol{u}_{t+1} - \boldsymbol{u}_t \rangle \right] \\
=& \frac{1 - \beta_1}{1 - \frac{\beta_1}{\sqrt{\beta_2}}} \mathbb{E}^{|\mathcal{F}_t} \left[ \left\langle \boldsymbol{G}_t, -\eta \frac{1}{\sqrt{\widetilde{\boldsymbol{\nu}}_t}} \boldsymbol{g}_t \right\rangle \right] + \mathbb{E}^{|\mathcal{F}_t} \left[ \left\langle \boldsymbol{G}_t, -\frac{\eta}{1 - \frac{\beta_1}{\sqrt{\beta_2}}} \left( \frac{1}{\sqrt{\boldsymbol{\nu}_t}} - \frac{1}{\sqrt{\widetilde{\boldsymbol{\nu}}_t}} \right) \boldsymbol{m}_t \right\rangle \right] \\
& + \mathbb{E}^{|\mathcal{F}_t} \left[ \left\langle \boldsymbol{G}_t, \beta_1 \frac{\eta}{1 - \frac{\beta_1}{\sqrt{\beta_2}}} \left( \frac{1}{\sqrt{\beta_2 \boldsymbol{\nu}_{t-1}}} - \frac{1}{\sqrt{\widetilde{\boldsymbol{\nu}}_t}} \right) \boldsymbol{m}_{t-1} \right\rangle \right].
\end{aligned} \tag{7}
$$

As $\mathbb{E}^{|\mathcal{F}_t} \left[ \left\langle \boldsymbol{G}_t, -\eta \frac{1}{\sqrt{\widetilde{\boldsymbol{\nu}}_t}} \boldsymbol{g}_t \right\rangle \right] = -\eta \frac{\|\boldsymbol{G}_t\|^2}{\sqrt{\widetilde{\boldsymbol{\nu}}_t}}$, we have

$$
\frac{1 - \beta_1}{1 - \frac{\beta_1}{\sqrt{\beta_2}}} \mathbb{E}^{|\mathcal{F}_t} \left[ \left\langle \boldsymbol{G}_t, -\eta \frac{1}{\sqrt{\widetilde{\boldsymbol{\nu}}_t}} \boldsymbol{g}_t \right\rangle \right] \leq -\frac{\|\boldsymbol{G}_t\|^2}{\sqrt{\widetilde{\boldsymbol{\nu}}_t}}.
$$

We then respectively bound the rest of the two terms in Eq. (7). To begin with,

$$
\begin{aligned}
& \mathbb{E}^{|\mathcal{F}_t} \left[ \left\langle \boldsymbol{G}_t, -\frac{\eta}{1 - \frac{\beta_1}{\sqrt{\beta_2}}} \left( \frac{1}{\sqrt{\boldsymbol{\nu}_t}} - \frac{1}{\sqrt{\widetilde{\boldsymbol{\nu}}_t}} \right) \boldsymbol{m}_t \right\rangle \right] \\
=& \mathbb{E}^{|\mathcal{F}_t} \left[ \left\langle \boldsymbol{G}_t, -\frac{\eta}{1 - \frac{\beta_1}{\sqrt{\beta_2}}} \left( \frac{(1-\beta_2)(\sigma_0^2 - \|\boldsymbol{g}_t\|^2)}{\sqrt{\boldsymbol{\nu}_t} \sqrt{\widetilde{\boldsymbol{\nu}}_t} (\sqrt{\boldsymbol{\nu}_t} + \sqrt{\widetilde{\boldsymbol{\nu}}_t})} \right) \boldsymbol{m}_t \right\rangle \right] \\
\leq& \frac{\eta}{1 - \frac{\beta_1}{\sqrt{\beta_2}}} \mathbb{E}^{|\mathcal{F}_t} \left[ \|\boldsymbol{G}_t\| \left( \frac{(1-\beta_2)(\sigma_0^2 + \|\boldsymbol{g}_t\|^2)}{\sqrt{\boldsymbol{\nu}_t} \sqrt{\widetilde{\boldsymbol{\nu}}_t} (\sqrt{\boldsymbol{\nu}_t} + \sqrt{\widetilde{\boldsymbol{\nu}}_t})} \right) \|\boldsymbol{m}_t\| \right] \\
=& \frac{\eta}{1 - \frac{\beta_1}{\sqrt{\beta_2}}} \mathbb{E}^{|\mathcal{F}_t} \left[ \|\boldsymbol{G}_t\| \left( \frac{(1-\beta_2)\|\boldsymbol{g}_t\|^2}{\sqrt{\boldsymbol{\nu}_t} \sqrt{\widetilde{\boldsymbol{\nu}}_t} (\sqrt{\boldsymbol{\nu}_t} + \sqrt{\widetilde{\boldsymbol{\nu}}_t})} \right) \|\boldsymbol{m}_t\| \right] + \frac{\eta}{1 - \frac{\beta_1}{\sqrt{\beta_2}}} \mathbb{E}^{|\mathcal{F}_t} \left[ \|\boldsymbol{G}_t\| \left( \frac{(1-\beta_2)\sigma_0^2}{\sqrt{\boldsymbol{\nu}_t} \sqrt{\widetilde{\boldsymbol{\nu}}_t} (\sqrt{\boldsymbol{\nu}_t} + \sqrt{\widetilde{\boldsymbol{\nu}}_t})} \right) \|\boldsymbol{m}_t\| \right].
\end{aligned} \tag{8}
$$

The first term in the right-hand-side of Eq. (8) can be bounded as

$$
\begin{aligned}
& \frac{\eta}{1 - \frac{\beta_1}{\sqrt{\beta_2}}} \mathbb{E}^{|\mathcal{F}_t} \left[ \|\boldsymbol{G}_t\| \left( \frac{(1-\beta_2)\|\boldsymbol{g}_t\|^2}{\sqrt{\boldsymbol{\nu}_t} \sqrt{\widetilde{\boldsymbol{\nu}}_t} (\sqrt{\boldsymbol{\nu}_t} + \sqrt{\widetilde{\boldsymbol{\nu}}_t})} \right) \|\boldsymbol{m}_t\| \right] \overset{(*)}{\leq} \frac{\eta(1-\beta_1)}{\left( \sqrt{1 - \frac{\beta_1}{\sqrt{\beta_2}}} \right)^3} \mathbb{E}^{|\mathcal{F}_t} \left[ \|\boldsymbol{G}_t\| \left( \frac{\sqrt{1-\beta_2}\|\boldsymbol{g}_t\|^2}{\sqrt{\widetilde{\boldsymbol{\nu}}_t} (\sqrt{\boldsymbol{\nu}_t} + \sqrt{\widetilde{\boldsymbol{\nu}}_t})} \right) \right] \\
& \overset{(\circ)}{\leq} \frac{\eta(1-\beta_1)}{\left( \sqrt{1 - \frac{\beta_1}{\sqrt{\beta_2}}} \right)^3} \frac{\|\boldsymbol{G}_t\|}{\sqrt{\widetilde{\boldsymbol{\nu}}_t}} \sqrt{\mathbb{E}^{|\mathcal{F}_t} \|\boldsymbol{g}_t\|^2} \sqrt{\mathbb{E}^{|\mathcal{F}_t} \frac{\|\boldsymbol{g}_t\|^2}{(\sqrt{\boldsymbol{\nu}_t} + \sqrt{\widetilde{\boldsymbol{\nu}}_t})^2}} \overset{(\bullet)}{\leq} \frac{\eta(1-\beta_1)\sqrt{1-\beta_2}}{\left( \sqrt{1 - \frac{\beta_1}{\sqrt{\beta_2}}} \right)^3} \frac{\|\boldsymbol{G}_t\|}{\sqrt{\widetilde{\boldsymbol{\nu}}_t}} \sqrt{\sigma_0^2 + \sigma_1^2 \|\boldsymbol{G}_t\|^2} \sqrt{\mathbb{E}^{|\mathcal{F}_t} \frac{\|\boldsymbol{g}_t\|^2}{(\sqrt{\boldsymbol{\nu}_t} + \sqrt{\widetilde{\boldsymbol{\nu}}_t})^2}} \\
& \leq \frac{\eta(1-\beta_1)\sqrt{1-\beta_2}}{\left( \sqrt{1 - \frac{\beta_1}{\sqrt{\beta_2}}} \right)^3} \frac{\|\boldsymbol{G}_t\|}{\sqrt{\widetilde{\boldsymbol{\nu}}_t}} (\sigma_0 + \sigma_1 \|\boldsymbol{G}_t\|) \sqrt{\mathbb{E}^{|\mathcal{F}_t} \frac{\|\boldsymbol{g}_t\|^2}{(\sqrt{\boldsymbol{\nu}_t} + \sqrt{\widetilde{\boldsymbol{\nu}}_t})^2}},
\end{aligned}
$$

where inequality $(*)$ uses Lemma 1, inequality $(\circ)$ is due to Holder's inequality, and inequality $(\bullet)$ is due to Assumption 2. Applying mean-value inequality respectively to $\frac{\eta(1-\beta_1)\sqrt{1-\beta_2}}{\left(\sqrt{1-\frac{\beta_1}{\sqrt{\beta_2}}}\right)^3}\mathbb{E}^{|\mathcal{F}_t}\frac{\|\boldsymbol{G}_t\|}{\sqrt{\widetilde{\boldsymbol{\nu}}_t}}\sigma_0\sqrt{\mathbb{E}^{|\mathcal{F}_t}\frac{\|\boldsymbol{g}_t\|^2}{(\sqrt{\boldsymbol{\nu}_t}+\sqrt{\widetilde{\boldsymbol{\nu}}_t})^2}}$ and

$\frac{\eta(1-\beta_1)\sqrt{1-\beta_2}}{\left(\sqrt{1-\frac{\beta_1}{\sqrt{\beta_2}}}\right)^3}\mathbb{E}^{|\mathcal{F}_t}\frac{\|\boldsymbol{G}_t\|}{\sqrt{\widetilde{\boldsymbol{\nu}}_t}}\sigma_1\|\boldsymbol{G}_t\|\sqrt{\mathbb{E}^{|\mathcal{F}_t}\frac{\|\boldsymbol{g}_t\|^2}{(\sqrt{\boldsymbol{\nu}_t}+\sqrt{\widetilde{\boldsymbol{\nu}}_t})^2}}$ and due to $\beta_1 \leq \beta_2$, we obtain that the

right-hand-side of the above inequality can be bounded by

$$\frac{1}{16}\eta\frac{1-\beta_1}{1-\frac{\beta_1}{\sqrt{\beta_2}}}\sqrt{1-\beta_2}\sigma_0\frac{\|\boldsymbol{G}_t\|^2}{\widetilde{\boldsymbol{\nu}}_t} + \frac{4\eta\sqrt{1-\beta_2}\sigma_0}{\left(1-\frac{\beta_1}{\sqrt{\beta_2}}\right)^2}\mathbb{E}^{|\mathcal{F}_t}\frac{\|\boldsymbol{g}_t\|^2}{(\sqrt{\boldsymbol{\nu}_t}+\sqrt{\widetilde{\boldsymbol{\nu}}_t})^2}$$

$$+ \frac{1}{16}\eta\frac{1-\beta_1}{1-\frac{\beta_1}{\sqrt{\beta_2}}}\frac{\|\boldsymbol{G}_t\|^2}{\sqrt{\widetilde{\boldsymbol{\nu}}_t}} + 4\eta\frac{(1-\beta_2)(1-\beta_1)}{(1-\frac{\beta_1}{\sqrt{\beta_2}})^2}\sigma_1^2\frac{\|\boldsymbol{G}_t\|^2}{\sqrt{\widetilde{\boldsymbol{\nu}}_t}}\mathbb{E}^{|\mathcal{F}_t}\frac{\|\boldsymbol{g}_t\|^2}{(\sqrt{\boldsymbol{\nu}_t}+\sqrt{\widetilde{\boldsymbol{\nu}}_t})^2}$$

$$\leq\frac{1}{8}\eta\frac{\|\boldsymbol{G}_t\|^2}{\sqrt{\widetilde{\boldsymbol{\nu}}_t}} + \frac{4\eta\sqrt{1-\beta_2}\sigma_0}{\left(1-\frac{\beta_1}{\sqrt{\beta_2}}\right)^2}\mathbb{E}^{|\mathcal{F}_t}\frac{\|\boldsymbol{g}_t\|^2}{\boldsymbol{\nu}_t} + \frac{1}{8}\eta\frac{\|\boldsymbol{G}_t\|^2}{\sqrt{\widetilde{\boldsymbol{\nu}}_t}} + 16\eta\frac{(1-\beta_2)}{(1-\beta_1)}\sigma_1^2\frac{\|\boldsymbol{G}_t\|^2}{\sqrt{\widetilde{\boldsymbol{\nu}}_t}}\mathbb{E}^{|\mathcal{F}_t}\frac{\|\boldsymbol{g}_t\|^2}{(\sqrt{\boldsymbol{\nu}_t}+\sqrt{\widetilde{\boldsymbol{\nu}}_t})^2}.$$

$$(9)$$

Here the inequality is due to $\widetilde{\boldsymbol{\nu}}_t = (1-\beta_2)\sigma_0^2 + \beta_2\boldsymbol{\nu}_{t-1} \geq (1-\beta_2)\sigma_0^2$. Meanwhile, we have

$$\left(\frac{1}{\sqrt{\beta_2\widetilde{\boldsymbol{\nu}}_t}} - \frac{1}{\sqrt{\widetilde{\boldsymbol{\nu}}_{t+1}}}\right)\|\boldsymbol{G}_t\|^2$$

$$= \frac{\|\boldsymbol{G}_t\|^2((1-\beta_2)^2\sigma_0^2 + \beta_2(1-\beta_2)\|\boldsymbol{g}_t\|^2)}{\sqrt{\beta_2\widetilde{\boldsymbol{\nu}}_t}\sqrt{\widetilde{\boldsymbol{\nu}}_{t+1}}(\sqrt{\beta_2\widetilde{\boldsymbol{\nu}}_t}+\sqrt{\widetilde{\boldsymbol{\nu}}_{t+1}})} \geq \frac{\|\boldsymbol{G}_t\|^2\beta_2(1-\beta_2)\|\boldsymbol{g}_t\|^2}{\sqrt{\beta_2\widetilde{\boldsymbol{\nu}}_t}\sqrt{\widetilde{\boldsymbol{\nu}}_{t+1}}(\sqrt{\beta_2\widetilde{\boldsymbol{\nu}}_t}+\sqrt{\widetilde{\boldsymbol{\nu}}_{t+1}})}$$

$$\geq \frac{1}{4}\frac{\|\boldsymbol{G}_t\|^2(1-\beta_2)\|\boldsymbol{g}_t\|^2}{\sqrt{\widetilde{\boldsymbol{\nu}}_t}(\sqrt{\boldsymbol{\nu}_t}+\sqrt{\widetilde{\boldsymbol{\nu}}_t})^2},$$

where in the last inequality, we use $\sqrt{\beta_2} \geq \frac{1}{2}$. Applying the above inequality back to Eq. (9), we obtain that

$$\frac{\eta}{1-\beta_1}\mathbb{E}^{|\mathcal{F}_t}\left[\|\boldsymbol{G}_t\|\left(\frac{(1-\beta_2)\boldsymbol{g}_t^2}{\sqrt{\boldsymbol{\nu}_t}\sqrt{\widetilde{\boldsymbol{\nu}}_t}(\sqrt{\boldsymbol{\nu}_t}+\sqrt{\widetilde{\boldsymbol{\nu}}_t})}\right)\|\boldsymbol{m}_t\|\right]$$

$$\leq\frac{1}{4}\eta\frac{\|\boldsymbol{G}_t\|^2}{\sqrt{\widetilde{\boldsymbol{\nu}}_t}} + \frac{4\eta\sqrt{1-\beta_2}\sigma_0}{\left(1-\frac{\beta_1^2}{\beta_2}\right)^2}\mathbb{E}^{|\mathcal{F}_t}\frac{\|\boldsymbol{g}_t\|^2}{\boldsymbol{\nu}_t} + \eta\frac{64}{(1-\beta_1)}\sigma_1^2\mathbb{E}^{|\mathcal{F}_t}\left(\frac{1}{\sqrt{\beta_2\widetilde{\boldsymbol{\nu}}_t}} - \frac{1}{\sqrt{\widetilde{\boldsymbol{\nu}}_{t+1}}}\right)\|\boldsymbol{G}_t\|^2.$$

$$(10)$$

Furthermore, due to Assumption 1, we have (we define $G_0 \triangleq G_1$)

$$\|\boldsymbol{G}_{t+1}\|^2 \leq\|\boldsymbol{G}_t\|^2 + 2\|\boldsymbol{G}_t\|\|\boldsymbol{G}_{t+1}-\boldsymbol{G}_t\| + \|\boldsymbol{G}_{t+1}-\boldsymbol{G}_t\|^2$$

$$\leq\|\boldsymbol{G}_t\|^2 + 2(L_0+L_1\|\boldsymbol{G}_t\|)\|\boldsymbol{G}_t\|\|\boldsymbol{w}_{t+1}-\boldsymbol{w}_t\| + 2(L_0^2+L_1^2\|\boldsymbol{G}_t\|^2)\|\boldsymbol{w}_{t+1}-\boldsymbol{w}_t\|^2,$$

which by $\frac{\eta}{\sqrt{1-\beta_2}} = \frac{\sqrt{1-\frac{\beta_1^2}{\beta_2}}(1-\frac{\beta_1}{\sqrt{\beta_2}})^2}{1024\sigma_1^2(L_1+L_0)(1-\beta_1)}$ further leads to

$$\frac{1}{\sqrt{\beta_2\widetilde{\boldsymbol{\nu}}_{t+1}}}\|\boldsymbol{G}_t\|^2$$

$$\geq\frac{1}{\sqrt{\beta_2\widetilde{\boldsymbol{\nu}}_{t+1}}}\left(\|\boldsymbol{G}_{t+1}\|^2 - 2(L_0+L_1\|\boldsymbol{G}_t\|)\|\boldsymbol{G}_t\|\|\boldsymbol{w}_{t+1}-\boldsymbol{w}_t\| - 2(L_0^2+L_1^2\|\boldsymbol{G}_t\|^2)\|\boldsymbol{w}_{t+1}-\boldsymbol{w}_t\|^2\right)$$

$$\geq\left(\frac{1}{\sqrt{\beta_2\widetilde{\boldsymbol{\nu}}_{t+1}}}\|\boldsymbol{G}_{t+1}\|^2 - \frac{2L_0}{\sigma_0}\frac{(1-\beta_1)}{64\sigma_1^2}\|\boldsymbol{w}_{t+1}-\boldsymbol{w}_t\|^2 - \frac{3}{8}\frac{(1-\beta_1)}{64\sigma_1^2}\frac{\|\boldsymbol{G}_t\|^2}{\sqrt{\widetilde{\boldsymbol{\nu}}_t}}\right).$$

Applying the above inequality back to Eq. (10) leads to that

$$\frac{\eta}{1-\frac{\beta_1}{\sqrt{\beta_2}}}\mathbb{E}^{|\mathcal{F}_t}\left[\|\boldsymbol{G}_t\|\left(\frac{(1-\beta_2)\boldsymbol{g}_t^2}{\sqrt{\boldsymbol{\nu}_t}\sqrt{\widetilde{\boldsymbol{\nu}}_t}(\sqrt{\boldsymbol{\nu}_t}+\sqrt{\widetilde{\boldsymbol{\nu}}_t})}\right)\|\boldsymbol{m}_t\|\right]$$

$$\leq \frac{5}{8}\eta\frac{\|\boldsymbol{G}_t\|^2}{\sqrt{\widetilde{\boldsymbol{\nu}}_t}}+\frac{4\eta\sqrt{1-\beta_2}\sigma_0}{(1-\beta_1)^2}\mathbb{E}^{|\mathcal{F}_t}\frac{\|\boldsymbol{g}_t\|^2}{\boldsymbol{\nu}_t}+\eta\frac{64}{(1-\beta_1)}\sigma_1^2\mathbb{E}^{|\mathcal{F}_t}\left(\frac{\|\boldsymbol{G}_t\|^2}{\sqrt{\beta_2}\widetilde{\boldsymbol{\nu}}_t}-\frac{\|\boldsymbol{G}_{t+1}\|^2}{\sqrt{\widetilde{\boldsymbol{\nu}}_{t+1}}}\right)$$

$$+2\frac{L_0}{\sigma_0}\mathbb{E}^{|\mathcal{F}_t}\|\boldsymbol{w}_{t+1}-\boldsymbol{w}_t\|^2. \tag{11}$$

As for the second term in the right-hand-side of Eq. (8), we have

$$\frac{\eta}{1-\frac{\beta_1}{\sqrt{\beta_2}}}\mathbb{E}^{|\mathcal{F}_t}\left[\|\boldsymbol{G}_t\|\left(\frac{(1-\beta_2)\sigma_0^2}{\sqrt{\boldsymbol{\nu}_t}\sqrt{\widetilde{\boldsymbol{\nu}}_t}(\sqrt{\boldsymbol{\nu}_t}+\sqrt{\widetilde{\boldsymbol{\nu}}_t})}\right)\|\boldsymbol{m}_t\|\right]$$

$$\leq \frac{\eta}{1-\frac{\beta_1}{\sqrt{\beta_2}}}\mathbb{E}^{|\mathcal{F}_t}\left[\|\boldsymbol{G}_t\|\left(\frac{\sqrt[4]{1-\beta_2}\sqrt{\sigma_0}}{\sqrt[4]{\widetilde{\boldsymbol{\nu}}_t}\sqrt{\boldsymbol{\nu}_t}}\right)\|\boldsymbol{m}_t\|\right]$$

$$\leq \frac{1}{8}\eta\frac{\|\boldsymbol{G}_t\|^2}{\sqrt{\widetilde{\boldsymbol{\nu}}_t}}+\frac{8\eta\sqrt{1-\beta_2}\sigma_0}{(1-\beta_1)^2}\mathbb{E}^{|\mathcal{F}_t}\left[\left(\frac{\|\boldsymbol{m}_t\|^2}{\boldsymbol{\nu}_t}\right)\right]. \tag{12}$$

In the last inequality we use again $\beta_2 \geq \beta_1$. With Inequalities (11) and (12), we conclude that the first-order term can be bounded by

$$\mathbb{E}^{|\mathcal{F}_t}\left[\langle\nabla f(\boldsymbol{w}_t),\boldsymbol{u}_{t+1}-\boldsymbol{u}_t\rangle\right]\leq-\frac{1}{4}\eta\mathbb{E}\frac{\|\boldsymbol{G}_t\|^2}{\sqrt{\widetilde{\boldsymbol{\nu}}_t}}+\frac{4\eta\sqrt{1-\beta_2}\sigma_0}{(1-\beta_1)^2}\mathbb{E}^{|\mathcal{F}_t}\frac{\|\boldsymbol{g}_t\|^2}{\boldsymbol{\nu}_t}+\eta\frac{64}{(1-\beta_1)}\sigma_1^2\mathbb{E}^{|\mathcal{F}_t}\left(\frac{\|\boldsymbol{G}_t\|^2}{\sqrt{\beta_2}\widetilde{\boldsymbol{\nu}}_t}-\frac{\|\boldsymbol{G}_{t+1}\|^2}{\sqrt{\widetilde{\boldsymbol{\nu}}_{t+1}}}\right)$$

$$+2\frac{L_0}{\sigma_0}\mathbb{E}^{|\mathcal{F}_t}\|\boldsymbol{w}_{t+1}-\boldsymbol{w}_t\|^2+\frac{8\eta\sqrt{1-\beta_2}\sigma_0}{(1-\beta_1)^2}\mathbb{E}^{|\mathcal{F}_t}\left[\left(\frac{\|\boldsymbol{m}_t\|^2}{\boldsymbol{\nu}_t}\right)\right]. \tag{13}$$

**Analysis for the second-order term.** To recall, the second order term is $\frac{1}{2}(L_0 + L_1\|\nabla f(\boldsymbol{w}_t)\|)(\|\boldsymbol{u}_{t+1}-\boldsymbol{w}_t\|+\|\boldsymbol{u}_t-\boldsymbol{w}_t\|)\|\boldsymbol{u}_{t+1}-\boldsymbol{u}_t\|$. Before we start, we have the following expansion for $\boldsymbol{u}_{t+1}-\boldsymbol{u}_t$: (here the operations are all coordinate-wisely)

$$\boldsymbol{u}_{t+1}-\boldsymbol{u}_t=\frac{\boldsymbol{w}_{t+1}-\boldsymbol{w}_t-\frac{\beta_1}{\sqrt{\beta_2}}(\boldsymbol{w}_t-\boldsymbol{w}_{t-1})}{1-\frac{\beta_1}{\sqrt{\beta_2}}}$$

$$=\frac{-\eta\frac{\boldsymbol{m}_t}{\sqrt{\boldsymbol{\nu}_t}}+\eta\frac{\beta_1}{\sqrt{\beta_2}}\frac{\boldsymbol{m}_{t-1}}{\sqrt{\boldsymbol{\nu}_{t-1}}}}{1-\frac{\beta_1}{\sqrt{\beta_2}}}=\frac{-\eta\frac{\boldsymbol{m}_t}{\sqrt{\boldsymbol{\nu}_t}}+\eta\beta_1\frac{\boldsymbol{m}_{t-1}}{\sqrt{\boldsymbol{\nu}_t}}-\eta\beta_1\frac{\boldsymbol{m}_{t-1}}{\sqrt{\boldsymbol{\nu}_t}}+\eta\frac{\beta_1}{\sqrt{\beta_2}}\frac{\boldsymbol{m}_{t-1}}{\sqrt{\boldsymbol{\nu}_{t-1}}}}{1-\frac{\beta_1}{\sqrt{\beta_2}}}$$

$$=\frac{-\eta\frac{(1-\beta_1)\boldsymbol{g}_t}{\sqrt{\boldsymbol{\nu}_t}}+\eta\frac{\beta_1(1-\beta_2)\|\boldsymbol{g}_t\|^2}{\sqrt{\beta_2}}\frac{\boldsymbol{m}_{t-1}}{\sqrt{\boldsymbol{\nu}_{t-1}}\sqrt{\boldsymbol{\nu}_t}(\sqrt{\boldsymbol{\nu}_t}+\sqrt{\beta_2\boldsymbol{\nu}_{t-1}})}}{1-\frac{\beta_1}{\sqrt{\beta_2}}} \tag{14}$$

Then firstly, we have

$$\frac{1}{2}L_0(\|\boldsymbol{u}_{t+1}-\boldsymbol{w}_t\|+\|\boldsymbol{u}_t-\boldsymbol{w}_t\|)\|\boldsymbol{u}_{t+1}-\boldsymbol{u}_t\|$$

$$\leq \frac{1}{2}L_0\left(\|\boldsymbol{u}_{t+1}-\boldsymbol{u}_t\|^2+\frac{1}{2}\|\boldsymbol{u}_{t+1}-\boldsymbol{w}_t\|^2+\frac{1}{2}\|\boldsymbol{u}_t-\boldsymbol{w}_t\|^2\right)$$

$$=\frac{1}{2}L_0\left(\left\|\frac{-\eta\frac{(1-\beta_1)\boldsymbol{g}_t}{\sqrt{\boldsymbol{\nu}_t}}+\eta\frac{\beta_1(1-\beta_2)\|\boldsymbol{g}_t\|^2}{\sqrt{\beta_2}}\frac{\boldsymbol{m}_{t-1}}{\sqrt{\boldsymbol{\nu}_{t-1}}\sqrt{\boldsymbol{\nu}_t}(\sqrt{\boldsymbol{\nu}_t}+\sqrt{\beta_2\boldsymbol{\nu}_{t-1}})}}{1-\frac{\beta_1}{\sqrt{\beta_2}}}\right\|^2+\frac{1}{2}\left\|\frac{\frac{\beta_1}{\sqrt{\beta_2}}}{1-\frac{\beta_1}{\sqrt{\beta_2}}}(\boldsymbol{w}_t-\boldsymbol{w}_{t-1})\right\|^2+\frac{1}{2}\left\|\frac{1}{1-\frac{\beta_1}{\sqrt{\beta_2}}}(\boldsymbol{w}_{t+1}-\boldsymbol{w}_t)\right\|^2\right)$$

$$\leq \frac{L_0\eta^2}{2}\left(\left(\frac{1-\beta_1}{1-\frac{\beta_1}{\sqrt{\beta_2}}}+\frac{\beta_1(1-\beta_1)}{(\sqrt{\beta_2}-\beta_1)\sqrt{1-\frac{\beta_1^2}{\beta_2}}}\right)^2\left\|\frac{\boldsymbol{g}_t}{\sqrt{\boldsymbol{\nu}_t}}\right\|^2+\frac{1}{2}\left(\frac{\frac{\beta_1}{\sqrt{\beta_2}}}{1-\frac{\beta_1}{\sqrt{\beta_2}}}\right)^2\left\|\frac{\boldsymbol{m}_{t-1}}{\sqrt{\boldsymbol{\nu}_{t-1}}}\right\|^2+\frac{1}{2}\left(\frac{1}{1-\frac{\beta_1}{\sqrt{\beta_2}}}\right)^2\left\|\frac{\boldsymbol{m}_t}{\sqrt{\boldsymbol{\nu}_t}}\right\|^2\right)$$

$$\overset{(\bullet)}{\leq}\frac{L_0\eta^2}{2}\left(2\left(\frac{1-\beta_1}{1-\frac{\beta_1}{\sqrt{\beta_2}}}+\frac{\beta_1(1-\beta_1)}{(\sqrt{\beta_2}-\beta_1)\sqrt{1-\frac{\beta_1^2}{\beta_2}}}\right)^2\left\|\frac{\boldsymbol{g}_t}{\sqrt{\boldsymbol{\nu}_t}}\right\|^2+\left(\frac{\frac{\beta_1}{\sqrt{\beta_2}}}{1-\frac{\beta_1}{\sqrt{\beta_2}}}\right)^2\left\|\frac{\boldsymbol{m}_{t-1}}{\sqrt{\boldsymbol{\nu}_{t-1}}}\right\|^2\right).$$

Secondly, we have

$$\frac{1}{2}L_1\|\nabla f(\boldsymbol{w}_t)\|(\|\boldsymbol{u}_{t+1}-\boldsymbol{w}_t\|+\|\boldsymbol{u}_t-\boldsymbol{w}_t\|)\|\boldsymbol{u}_{t+1}-\boldsymbol{u}_t\|$$

$$\leq\frac{1}{2}L_1\|\nabla f(\boldsymbol{w}_t)\|(2\|\boldsymbol{u}_{t+1}-\boldsymbol{w}_t\|+\|\boldsymbol{u}_{t+1}-\boldsymbol{u}_t\|)\left(\frac{\left\|\eta\frac{(1-\beta_1)\boldsymbol{g}_t}{\sqrt{\boldsymbol{\nu}_t}}\right\|}{1-\frac{\beta_1}{\sqrt{\beta_2}}}+\frac{\eta\frac{\beta_1(1-\beta_2)\|\boldsymbol{g}_t\|^2}{\sqrt{\beta_2}}\frac{\|\boldsymbol{m}_{t-1}\|}{\sqrt{\boldsymbol{\nu}_{t-1}}\sqrt{\boldsymbol{\nu}_t}(\sqrt{\boldsymbol{\nu}_t}+\sqrt{\beta_2\boldsymbol{\nu}_{t-1}})}}{1-\frac{\beta_1}{\sqrt{\beta_2}}}\right)$$

$$\overset{(*)}{\leq}\frac{1}{2}L_1\|\nabla f(\boldsymbol{w}_t)\|(2\|\boldsymbol{u}_{t+1}-\boldsymbol{w}_t\|+\|\boldsymbol{u}_{t+1}-\boldsymbol{u}_t\|)\left(\frac{\left\|\eta\frac{(1-\beta_1)\boldsymbol{g}_t}{\sqrt{\boldsymbol{\nu}_t}}\right\|}{1-\frac{\beta_1}{\sqrt{\beta_2}}}+\frac{\eta\frac{\beta_1(1-\beta_1)}{\sqrt{\beta_2}}\frac{\|\boldsymbol{g}_t\|}{\sqrt{\boldsymbol{\nu}_t}}}{(1-\frac{\beta_1}{\sqrt{\beta_2}})\sqrt{1-\frac{\beta_1^2}{\beta_2}}}\right)$$

$$=\frac{L_1}{2}\eta\left(\frac{1-\beta_1}{1-\frac{\beta_1}{\sqrt{\beta_2}}}+\frac{\beta_1(1-\beta_1)}{(\sqrt{\beta_2}-\beta_1)\sqrt{1-\frac{\beta_1^2}{\beta_2}}}\right)\|\nabla f(\boldsymbol{w}_t)\|(2\|\boldsymbol{u}_{t+1}-\boldsymbol{w}_t\|+\|\boldsymbol{u}_t-\boldsymbol{u}_{t+1}\|)\frac{\|\boldsymbol{g}_t\|}{\sqrt{\boldsymbol{\nu}_t}}$$

$$\overset{(\circ)}{=}\frac{L_1}{2}\eta\left(\frac{1-\beta_1}{1-\frac{\beta_1}{\sqrt{\beta_2}}}+\frac{\beta_1(1-\beta_1)}{(\sqrt{\beta_2}-\beta_1)\sqrt{1-\frac{\beta_1^2}{\beta_2}}}\right)\|\boldsymbol{G}_t\|\left(\|\boldsymbol{u}_{t+1}-\boldsymbol{u}_t\|+2\frac{1}{1-\frac{\beta_1}{\sqrt{\beta_2}}}\eta\left\|\frac{\boldsymbol{m}_t}{\sqrt{\boldsymbol{\nu}_t}}\right\|\right)\frac{\|\boldsymbol{g}_t\|}{\sqrt{\boldsymbol{\nu}_t}}.$$

where inequality $(*)$ is due to that $\frac{\|\boldsymbol{m}_{t-1}\|}{\sqrt{\boldsymbol{\nu}_{t-1}}}\leq\frac{1-\beta_1}{\sqrt{1-\beta_2}\sqrt{1-\frac{\beta_1^2}{\beta_2}}}$, $\frac{\|\boldsymbol{g}_t\|}{\sqrt{\boldsymbol{\nu}_t}}\leq\frac{1}{\sqrt{1-\beta_2}}$, and equation $(\circ)$ is

due to $\boldsymbol{u}_t-\boldsymbol{w}_t=\frac{\frac{\beta_1}{\sqrt{\beta_2}}}{1-\frac{\beta_1}{\sqrt{\beta_2}}}(\boldsymbol{w}_t-\boldsymbol{w}_{t-1})$ and $\boldsymbol{u}_{t+1}-\boldsymbol{w}_t=\frac{1}{1-\frac{\beta_1}{\sqrt{\beta_2}}}(\boldsymbol{w}_{t+1}-\boldsymbol{w}_t)$. As for the term

$\|\boldsymbol{G}_t\|\frac{\|\boldsymbol{m}_t\|}{\sqrt{\boldsymbol{\nu}_t}}\frac{\|\boldsymbol{g}_t\|}{\sqrt{\boldsymbol{\nu}_t}}$, we first add additional denominator for it. Specifically, we have

$$\begin{aligned}\|\boldsymbol{G}_t\|\frac{\|\boldsymbol{m}_t\|}{\sqrt{\boldsymbol{\nu}_t}}\frac{\|\boldsymbol{g}_t\|}{\sqrt{\boldsymbol{\nu}_t}}&=\frac{\|\boldsymbol{G}_t\|\|\boldsymbol{m}_t\|\|\boldsymbol{g}_t\|}{\boldsymbol{\nu}_t+(1-\beta_2)\sigma_0^2}+\frac{\|\boldsymbol{G}_t\|\|\boldsymbol{m}_t\|\|\boldsymbol{g}_t\|(1-\beta_2)\sigma_0^2}{(\boldsymbol{\nu}_t+(1-\beta_2)\sigma_0^2)\boldsymbol{\nu}_t}\\&\leq\frac{\|\boldsymbol{G}_t\|\|\boldsymbol{m}_t\|\|\boldsymbol{g}_t\|}{\boldsymbol{\nu}_t+(1-\beta_2)\sigma_0^2}+\frac{\|\boldsymbol{G}_t\|\|\boldsymbol{m}_t\|\sigma_0}{\sqrt{\boldsymbol{\nu}_t+(1-\beta_2)\sigma_0^2}\sqrt{\boldsymbol{\nu}_t}}\\&\leq\frac{\|\boldsymbol{G}_t\|\|\boldsymbol{m}_t\|\|\boldsymbol{g}_t\|}{\boldsymbol{\nu}_t+(1-\beta_2)\sigma_0^2}+\frac{1}{2}\frac{\|\boldsymbol{G}_t\|^2\sigma_0}{\boldsymbol{\nu}_t+(1-\beta_2)\sigma_0^2}+\frac{1}{2}\sigma_0\frac{\|\boldsymbol{m}_t\|^2}{\boldsymbol{\nu}_t}\\&\leq\frac{\|\boldsymbol{G}_t\|\|\boldsymbol{m}_t\|\|\boldsymbol{g}_t\|}{\boldsymbol{\nu}_t+(1-\beta_2)\sigma_0^2}+\frac{1}{2\sqrt{1-\beta_2}}\frac{\|\boldsymbol{G}_t\|^2}{\sqrt{\boldsymbol{\nu}_t+(1-\beta_2)\sigma_0^2}}+\frac{1}{2}\sigma_0\frac{\|\boldsymbol{m}_t\|^2}{\boldsymbol{\nu}_t}.\end{aligned}$$

We analyze the first term in the right-hand-side of above inequality more carefully. Specifically, this term with expectation can be bounded as

$$\mathbb{E}^{|\mathcal{F}_t}\frac{\|\boldsymbol{G}_t\|\|\boldsymbol{m}_t\|\|\boldsymbol{g}_t\|}{\boldsymbol{\nu}_t+(1-\beta_2)\sigma_0^2}$$

$$\leq\mathbb{E}^{|\mathcal{F}_t}\frac{\|\boldsymbol{G}_t\|\|\boldsymbol{m}_t\|\|\boldsymbol{g}_t\|}{\sqrt{\boldsymbol{\nu}_t+(1-\beta_2)\sigma_0^2}\sqrt{\beta_2\boldsymbol{\nu}_{t-1}+(1-\beta_2)\sigma_0^2}}$$

$$\leq\frac{\|\boldsymbol{G}_t\|}{\sqrt{\beta_2\boldsymbol{\nu}_{t-1}+(1-\beta_2)\sigma_0^2}}\sqrt{\|\boldsymbol{g}_t\|^2}\sqrt{\mathbb{E}^{|\mathcal{F}_t}\frac{\|\boldsymbol{m}_t\|^2}{\boldsymbol{\nu}_t+(1-\beta_2)\sigma_0^2}}$$

$$\overset{(\star)}{\leq}\frac{\|\boldsymbol{G}_t\|}{\sqrt{\beta_2\boldsymbol{\nu}_{t-1}+(1-\beta_2)\sigma_0^2}}\sqrt{\sigma_1^2\|\boldsymbol{G}_t\|^2+\sigma_0^2}\sqrt{\mathbb{E}^{|\mathcal{F}_t}\frac{\|\boldsymbol{m}_t\|^2}{\boldsymbol{\nu}_t+(1-\beta_2)\sigma_0^2}}$$

$$\leq\frac{\|\boldsymbol{G}_t\|}{\sqrt{\beta_2\boldsymbol{\nu}_{t-1}+(1-\beta_2)\sigma_0^2}}(\sigma_1\|\boldsymbol{G}_t\|+\sigma_0)\sqrt{\mathbb{E}^{|\mathcal{F}_t}\frac{\|\boldsymbol{m}_t\|^2}{\boldsymbol{\nu}_t+(1-\beta_2)\sigma_0^2}}$$

$$\leq\frac{1-\beta_1}{\sqrt{1-\beta_2}\sqrt{1-\frac{\beta_1^2}{\beta_2}}}\sigma_1\frac{\|\boldsymbol{G}_t\|^2}{\sqrt{\beta_2\boldsymbol{\nu}_{t-1}+(1-\beta_2)\sigma_0^2}}+\frac{1}{2\sqrt{1-\beta_2}}\frac{\|\boldsymbol{G}_t\|^2}{\sqrt{\beta_2\boldsymbol{\nu}_{t-1}+(1-\beta_2)\sigma_0^2}}+\frac{\sigma_0}{2}\mathbb{E}^{|\mathcal{F}_t}\frac{\|\boldsymbol{m}_t\|^2}{\boldsymbol{\nu}_t+(1-\beta_2)\sigma_0^2},$$

where Eq. $(\star)$ is due to Holder's inequality.

Meanwhile, due to Eq. (14), we have that the term $|\boldsymbol{G}_t|\|\boldsymbol{u}_{t+1} - \boldsymbol{u}_t\|\frac{\|\boldsymbol{g}_t\|}{\sqrt{\boldsymbol{\nu}_t}}$ can be be bounded as

$$|\boldsymbol{G}_t|\|\boldsymbol{u}_{t+1} - \boldsymbol{u}_t\|\frac{\|\boldsymbol{g}_t\|}{\sqrt{\boldsymbol{\nu}_t}} \leq \eta \left( \frac{1 - \beta_1}{1 - \frac{\beta_1}{\sqrt{\beta_2}}} + \frac{\beta_1(1 - \beta_1)}{(\sqrt{\beta_2} - \beta_1)\sqrt{1 - \frac{\beta_1^2}{\beta_2}}} \right) \|\boldsymbol{G}_t\|\frac{\|\boldsymbol{g}_t\|}{\sqrt{\boldsymbol{\nu}_t}}\frac{\|\boldsymbol{g}_t\|}{\sqrt{\boldsymbol{\nu}_t}}.$$

Then, following the similar reasoning above, we have $|\boldsymbol{G}_t|\|\boldsymbol{u}_{t+1} - \boldsymbol{u}_t\|\frac{\|\boldsymbol{g}_t\|}{\sqrt{\boldsymbol{\nu}_t}}$ can be bounded as

$$\mathbb{E}^{|\mathcal{F}_t}\|\boldsymbol{G}_t\|\frac{\|\boldsymbol{g}_t\|}{\sqrt{\boldsymbol{\nu}_t}}\frac{\|\boldsymbol{g}_t\|}{\sqrt{\boldsymbol{\nu}_t}}$$

$$\leq \frac{1}{\sqrt{1 - \beta_2}}\sigma_1\frac{\|\boldsymbol{G}_t\|^2}{\sqrt{\beta_2\boldsymbol{\nu}_{t-1} + (1 - \beta_2)\sigma_0^2}} + \frac{1}{2\sqrt{1 - \beta_2}}\frac{\|\boldsymbol{G}_t\|^2}{\sqrt{\beta_2\boldsymbol{\nu}_{t-1} + (1 - \beta_2)\sigma_0^2}} + \sigma_0\mathbb{E}^{|\mathcal{F}_t}\frac{\|\boldsymbol{g}_t\|^2}{\boldsymbol{\nu}_t + (1 - \beta_2)\sigma_0^2}$$

$$+ \frac{1}{2\sqrt{1 - \beta_2}}\frac{\|\boldsymbol{G}_t\|^2}{\sqrt{\boldsymbol{\nu}_t + (1 - \beta_2)\sigma_0^2}} + \frac{1}{2}\sigma_0\mathbb{E}^{|\mathcal{F}_t}\frac{\|\boldsymbol{g}_t\|^2}{\boldsymbol{\nu}_t}.$$

Putting all the estimations together, we have that the second-order term can be bounded by

$$\mathbb{E}^{|\mathcal{F}_t}\frac{1}{2}(L_0 + L_1\|\nabla f(\boldsymbol{w}_t)\|)(\|\boldsymbol{u}_{t+1} - \boldsymbol{w}_t\| + \|\boldsymbol{u}_t - \boldsymbol{w}_t\|)\|\boldsymbol{u}_{t+1} - \boldsymbol{u}_t\|$$

$$\leq \frac{L_1\eta^2}{1 - \frac{\beta_1}{\sqrt{\beta_2}}} \left( \frac{1 - \beta_1}{1 - \frac{\beta_1}{\sqrt{\beta_2}}} + \frac{\beta_1(1 - \beta_1)}{(\sqrt{\beta_2} - \beta_1)\sqrt{1 - \frac{\beta_1^2}{\beta_2}}} \right) \left( \frac{2}{\sqrt{1 - \beta_2}}\frac{\|\boldsymbol{G}_t\|^2}{\sqrt{\beta_2\boldsymbol{\nu}_{t-1} + (1 - \beta_2)\sigma_0^2}} + \frac{\sigma_0}{2}\mathbb{E}^{|\mathcal{F}_t}\frac{\|\boldsymbol{g}_t\|^2}{\boldsymbol{\nu}_t} \right)$$

$$+ \frac{L_0\eta^2}{2} \left( 2 \left( \frac{1 - \beta_1}{1 - \frac{\beta_1}{\sqrt{\beta_2}}} + \frac{\beta_1(1 - \beta_1)}{(\sqrt{\beta_2} - \beta_1)\sqrt{1 - \frac{\beta_1^2}{\beta_2}}} \right)^2 \mathbb{E}^{|\mathcal{F}_t}\left\|\frac{\boldsymbol{g}_t}{\sqrt{\boldsymbol{\nu}_t}}\right\|^2 + \left( \frac{\frac{\beta_1}{\sqrt{\beta_2}}}{1 - \frac{\beta_1}{\sqrt{\beta_2}}} \right)^2 \left\|\frac{\boldsymbol{m}_{t-1}}{\sqrt{\boldsymbol{\nu}_{t-1}}}\right\|^2 \right)$$

$$\leq 4\frac{L_1\eta^2}{1 - \beta_1} \left( 1 + \frac{1}{\sqrt{1 - \beta_1}} \right) \left( \frac{2}{\sqrt{1 - \beta_2}}\frac{\|\boldsymbol{G}_t\|^2}{\sqrt{\beta_2\boldsymbol{\nu}_{t-1} + (1 - \beta_2)\sigma_0^2}} + \frac{\sigma_0}{2}\mathbb{E}^{|\mathcal{F}_t}\frac{\|\boldsymbol{g}_t\|^2}{\boldsymbol{\nu}_t} \right)$$

$$+ 2L_0\eta^2 \left( 2 \left( 1 + \frac{1}{\sqrt{1 - \beta_1}} \right)^2 \mathbb{E}^{|\mathcal{F}_t}\left\|\frac{\boldsymbol{g}_t}{\sqrt{\boldsymbol{\nu}_t}}\right\|^2 + \left( \frac{1}{1 - \beta_1} \right)^2 \left\|\frac{\boldsymbol{m}_{t-1}}{\sqrt{\boldsymbol{\nu}_{t-1}}}\right\|^2 \right)$$

$$\leq \frac{1}{8}\eta\frac{\|\boldsymbol{G}_t\|^2}{\sqrt{\widetilde{\boldsymbol{\nu}}_t}} + 4\frac{L_1\eta^2\sigma_0}{(1 - \beta_1)^{\frac{3}{2}}}\mathbb{E}^{|\mathcal{F}_t}\frac{\|\boldsymbol{g}_t\|^2}{\boldsymbol{\nu}_t} + 2L_0\eta^2 \left( 8\frac{1}{1 - \beta_1}\mathbb{E}^{|\mathcal{F}_t}\left\|\frac{\boldsymbol{g}_t}{\sqrt{\boldsymbol{\nu}_t}}\right\|^2 + \left( \frac{1}{1 - \beta_1} \right)^2 \left\|\frac{\boldsymbol{m}_{t-1}}{\sqrt{\boldsymbol{\nu}_{t-1}}}\right\|^2 \right).$$

$$(15)$$

Here in the second inequality we use $\beta_2 \geq \beta_1$, and in the last inequality we use $\frac{\eta}{\sqrt{1 - \beta_2}} = \frac{\sqrt{1 - \frac{\beta_1^2}{\beta_2}}(1 - \frac{\beta_1}{\sqrt{\beta_2}})^2}{1024\sigma_1^2(L_1 + L_0)(1 - \beta_1)}$.

Applying the estimations of the first-order term (Eq. (13)) and the second-order term (Eq. (15)) back into the descent lemma, we derive that

$$\mathbb{E}^{|\mathcal{F}_t} f(\boldsymbol{u}_{t+1}) \leq f(\boldsymbol{u}_t) - \frac{1}{8}\eta\frac{\|\boldsymbol{G}_t\|^2}{\sqrt{\widetilde{\boldsymbol{\nu}}_t}} + \frac{4\eta\sqrt{1 - \beta_2}\sigma_0}{(1 - \beta_1)^2}\mathbb{E}^{|\mathcal{F}_t}\frac{\|\boldsymbol{g}_t\|^2}{\boldsymbol{\nu}_t} + \eta\frac{64}{(1 - \beta_1)}\sigma_1^2\mathbb{E}^{|\mathcal{F}_t} \left( \frac{\|\boldsymbol{G}_t\|^2}{\sqrt{\beta_2\widetilde{\boldsymbol{\nu}}_t}} - \frac{\|\boldsymbol{G}_{t+1}\|^2}{\sqrt{\widetilde{\boldsymbol{\nu}}_{t+1}}} \right)$$

$$+ 2\frac{L_0}{\sigma_0}\mathbb{E}^{|\mathcal{F}_t}\|\boldsymbol{w}_{t+1} - \boldsymbol{w}_t\|^2 + \frac{8\eta\sqrt{1 - \beta_2}\sigma_0}{(1 - \beta_1)^2}\mathbb{E}^{|\mathcal{F}_t} \left[ \left( \frac{\|\boldsymbol{m}_t\|^2}{\boldsymbol{\nu}_t} \right) \right]$$

$$+ 4\frac{L_1\eta^2\sigma_0}{(1 - \beta_1)^{\frac{3}{2}}}\mathbb{E}^{|\mathcal{F}_t}\frac{\|\boldsymbol{g}_t\|^2}{\boldsymbol{\nu}_t} + 2L_0\eta^2 \left( 8\frac{1}{1 - \beta_1}\mathbb{E}^{|\mathcal{F}_t}\left\|\frac{\boldsymbol{g}_t}{\sqrt{\boldsymbol{\nu}_t}}\right\|^2 + \left( \frac{1}{1 - \beta_1} \right)^2 \left\|\frac{\boldsymbol{m}_{t-1}}{\sqrt{\boldsymbol{\nu}_{t-1}}}\right\|^2 \right).$$

Taking expectation to the above inequality and summing it over $t \in [1, T]$ then gives

$$
\begin{aligned}
\frac{1}{8} \eta \sum_{t=1}^{T} \mathbb{E} \frac{\|\boldsymbol{G}_t\|^2}{\sqrt{\widetilde{\boldsymbol{\nu}}_t}} \leq & f(\boldsymbol{u}_1) - f^* + \eta \frac{64}{(1-\beta_1)} \sigma_1^2 \frac{\|\boldsymbol{G}_1\|^2}{\sqrt{\beta_2 \widetilde{\boldsymbol{\nu}}_1}} + \eta \frac{64}{(1-\beta_1)} \sigma_1^2 \left( \frac{1}{\sqrt{\beta_2}} - 1 \right) \sum_{t=1}^{T} \mathbb{E} \frac{\|\boldsymbol{G}_t\|^2}{\sqrt{\widetilde{\boldsymbol{\nu}}_t}} \\
& + \left( \frac{4\eta\sqrt{1-\beta_2}\sigma_0}{(1-\beta_1)^2} + 4 \frac{L_1\eta^2\sigma_0}{(1-\beta_1)^{\frac{3}{2}}} + \frac{16L_0\eta^2}{1-\beta_1} \right) \sum_{t=1}^{T} \mathbb{E} \frac{\|\boldsymbol{g}_t\|^2}{\boldsymbol{\nu}_t} \\
& + \left( 2\frac{L_0}{\sigma_0}\eta^2 + \frac{8\eta\sqrt{1-\beta_2}\sigma_0}{(1-\beta_1)^2} + \frac{2L_0\eta^2}{(1-\beta_1)^2} \right) \sum_{t=1}^{T} \mathbb{E} \left\| \frac{\boldsymbol{m}_t}{\sqrt{\boldsymbol{\nu}_t}} \right\|^2 .
\end{aligned}
$$

Since $\beta_2 \geq \frac{1}{2}$ and $1 - \beta_2 \leq \frac{1-\beta_1}{1024\sigma_1^2}$, we have

$$
\eta \frac{64}{(1-\beta_1)} \sigma_1^2 \left( \frac{1}{\sqrt{\beta_2}} - 1 \right) \sum_{t=1}^{T} \mathbb{E} \frac{\|\boldsymbol{G}_t\|^2}{\sqrt{\widetilde{\boldsymbol{\nu}}_t}} \leq \frac{1}{16} \eta \sum_{t=1}^{T} \mathbb{E} \frac{\|\boldsymbol{G}_t\|^2}{\sqrt{\widetilde{\boldsymbol{\nu}}_t}} .
$$

By further applying Lemma 3 and $\beta_2 \geq \beta_1$, we obtain

$$
\begin{aligned}
& \frac{1}{16} \eta \sum_{t=1}^{T} \mathbb{E} \frac{\|\boldsymbol{G}_t\|^2}{\sqrt{\widetilde{\boldsymbol{\nu}}_t}} \\
\leq & f(\boldsymbol{u}_1) - f^* + \eta \frac{64}{(1-\beta_1)} \sigma_1^2 \frac{\|\boldsymbol{G}_1\|^2}{\sqrt{\beta_2 \widetilde{\boldsymbol{\nu}}_1}} \\
& + \frac{1}{1-\beta_2} \left( \frac{36\eta\sqrt{1-\beta_2}\sigma_0}{(1-\beta_1)^2} + 4\frac{L_1\eta^2\sigma_0}{(1-\beta_1)^{\frac{3}{2}}} + \frac{24L_0\eta^2}{1-\beta_1} + 8\frac{L_0}{\sigma_0}\eta^2 \right) (\mathbb{E}\ln\boldsymbol{\nu}_T - T\ln\beta_2) \\
\leq & f(\boldsymbol{w}_1) - f^* + \eta \frac{64}{(1-\beta_1)} \sigma_1^2 \frac{\|\boldsymbol{G}_1\|^2}{\sqrt{\beta_2 \widetilde{\boldsymbol{\nu}}_1}} \\
& + \frac{1}{1-\beta_2} \left( \frac{147456\eta^2(L_0+L_1)\sigma_1^2\sigma_0}{(1-\beta_1)^{\frac{5}{2}}} + 4\frac{L_1\eta^2\sigma_0}{(1-\beta_1)^{\frac{3}{2}}} + \frac{24L_0\eta^2}{1-\beta_1} + 8\frac{L_0}{\sigma_0}\eta^2 \right) (\mathbb{E}\ln\boldsymbol{\nu}_T - T\ln\beta_2) .
\end{aligned}
$$

Here last inequality we apply $\frac{\eta}{\sqrt{1-\beta_2}} = \frac{\sqrt{1-\frac{\beta_1^2}{\beta_2}}(1-\frac{\beta_1}{\sqrt{\beta_2}})^2}{1024\sigma_1^2(L_1+L_0)(1-\beta_1)}$.

Below we transfer the above bound to the bound of $\sum_{t=1}^{T}\|\boldsymbol{G}_t\|$ by two rounds of divide-and-conquer. In the first round, we will bound $\mathbb{E}\ln\boldsymbol{\nu}_T$. To start with, we have that

$$
\begin{aligned}
\frac{\|\boldsymbol{G}_t\|^2}{\sqrt{\widetilde{\boldsymbol{\nu}}_t}} \mathbb{1}_{\|G_t\| \geq \frac{\sigma_0}{\sigma_1}} & \geq \frac{\frac{1}{2\sigma_1^2} \mathbb{E}^{|\mathcal{F}_t} \|\boldsymbol{g}_t\|^2}{\sqrt{\widetilde{\boldsymbol{\nu}}_t}} \mathbb{1}_{\|G_t\| \geq \frac{\sigma_0}{\sigma_1}} \\
& = \frac{\frac{1}{2\sigma_1^2} \mathbb{E}^{|\mathcal{F}_t} \|\boldsymbol{g}_t\|^2}{\sqrt{\beta_2 \boldsymbol{\nu}_{t-1} + (1-\beta_2)\sigma_0^2}} \mathbb{1}_{\|G_t\| \geq \frac{\sigma_0}{\sigma_1}} \\
& \geq \frac{1}{2\sigma_1^2} \mathbb{E}^{|\mathcal{F}_t} \frac{\beta_2^{T-t} \|\boldsymbol{g}_t\|^2}{\sqrt{\boldsymbol{\nu}_T + (1-\beta_2)\sigma_0^2}} \mathbb{1}_{\|G_t\| \geq \frac{\sigma_0}{\sigma_1}} ,
\end{aligned}
$$

where the last inequality is due to that

$$
\beta_2\boldsymbol{\nu}_{t-1} + (1-\beta_2)\sigma_0^2 \leq \beta_2^{t-T}\boldsymbol{\nu}_T + (1-\beta_2)\sigma_0^2 \leq (\boldsymbol{\nu}_T + (1-\beta_2)\sigma_0^2)\beta_2^{2(t-T)} . \tag{18}
$$

Furthermore, we have

$$\frac{\sigma_0^2 + \frac{\beta_2^T \boldsymbol{\nu}_0}{1-\beta_2}}{\sqrt{\boldsymbol{\nu}_T + (1-\beta_2)\sigma_0^2}} + \sum_{t=1}^{T} \mathbb{E} \frac{\beta_2^{T-t}\|\boldsymbol{g}_t\|^2}{\sqrt{\boldsymbol{\nu}_T + (1-\beta_2)\sigma_0^2}} \mathbb{1}_{\|\boldsymbol{G}_t\|<\frac{\sigma_0}{\sigma_1}}$$

$$\leq \frac{\sigma_0^2 + \frac{\beta_2^T \boldsymbol{\nu}_0}{1-\beta_2}}{\sqrt{\boldsymbol{\nu}_0\beta_2^T + \sum_{s=1}^{T}\beta_2^{T-s}\|g_s\|^2 \mathbb{1}_{\|\boldsymbol{G}_s\|<\frac{\sigma_0}{\sigma_1}} + (1-\beta_2)\sigma_0^2}} + \sum_{t=1}^{T} \mathbb{E} \frac{\beta_2^{T-s}\|\boldsymbol{g}_t\|^2}{\sqrt{\boldsymbol{\nu}_0\beta_2^T + \sum_{s=1}^{T}\beta_2^{T-s}\|g_s\|^2 \mathbb{1}_{\|\boldsymbol{G}_s\|<\frac{\sigma_0}{\sigma_1}} + (1-\beta_2)\sigma_0^2}} \mathbb{1}_{\|\boldsymbol{G}_t\|<\frac{\sigma_0}{\sigma_1}}$$

$$= \frac{1}{1-\beta_2}\mathbb{E}\sqrt{\boldsymbol{\nu}_0\beta_2^T + \sum_{s=1}^{T}\beta_2^{T-s}\|g_s\|^2 \mathbb{1}_{\|\boldsymbol{G}_s\|<\frac{\sigma_0}{\sigma_1}} + (1-\beta_2)\sigma_0^2} \leq \frac{1}{1-\beta_2}\sqrt{\beta_2^T \boldsymbol{\nu}_0 + 2\sigma_0^2}. \tag{19}$$

Conclusively, we obtain

$$\mathbb{E}\sqrt{\boldsymbol{\nu}_T + (1-\beta_2)\sigma_0^2}$$

$$= (1-\beta_2)\left( \frac{\sigma_0^2 + \frac{\beta_2^T \boldsymbol{\nu}_0}{1-\beta_2}}{\sqrt{\boldsymbol{\nu}_T + (1-\beta_2)\sigma_0^2}} + \sum_{t=1}^{T}\mathbb{E}\frac{\beta_2^{T-t}\|\boldsymbol{g}_t\|^2}{\sqrt{\boldsymbol{\nu}_T + (1-\beta_2)\sigma_0^2}}\mathbb{1}_{\|\boldsymbol{G}_t\|<\frac{\sigma_0}{\sigma_1}} \right.$$

$$\left. + \sum_{t=1}^{T}\mathbb{E}\frac{\beta_2^{T-t}\|\boldsymbol{g}_t\|^2}{\sqrt{\boldsymbol{\nu}_T + (1-\beta_2)\sigma_0^2}}\mathbb{1}_{\|\boldsymbol{G}_t\|\geq\frac{\sigma_0}{\sigma_1}} \right)$$

$$\leq \sqrt{\beta_2^T \boldsymbol{\nu}_0 + 2\sigma_0^2} + 2(1-\beta_2)\sigma_1^2\mathbb{E}\sum_{t=1}^{T}\frac{\|\boldsymbol{G}_t\|^2}{\sqrt{\widetilde{\boldsymbol{\nu}}_t}}\mathbb{1}_{\|\boldsymbol{G}_t\|\geq\frac{\sigma_0}{\sigma_1}}$$

$$\leq \sqrt{\beta_2^T \boldsymbol{\nu}_0 + 2\sigma_0^2} + 2(1-\beta_2)\sigma_1^2\mathbb{E}\sum_{t=1}^{T}\frac{\|\boldsymbol{G}_t\|^2}{\sqrt{\widetilde{\boldsymbol{\nu}}_t}}.$$

Substituting $\mathbb{E}\sum_{t=1}^{T}\frac{\|\boldsymbol{G}_t\|^2}{\sqrt{\widetilde{\boldsymbol{\nu}}_t}}$ according to Eq. (17), we obtain that

$$\mathbb{E}\sqrt{\boldsymbol{\nu}_T + (1-\beta_2)\sigma_0^2}$$

$$\leq \sqrt{\beta_2^T \boldsymbol{\nu}_0 + 2\sigma_0^2} + \frac{2(1-\beta_2)\sigma_1^2}{\eta}\eta\mathbb{E}\sum_{t=1}^{T}\frac{\|\boldsymbol{G}_t\|^2}{\sqrt{\widetilde{\boldsymbol{\nu}}_t}}$$

$$\leq \sqrt{\beta_2^T \boldsymbol{\nu}_0 + 2\sigma_0^2} + \frac{2(1-\beta_2)\sigma_1^2}{\eta}\left( f(\boldsymbol{w}_1) - f^* + \eta\frac{64}{(1-\beta_1)}\sigma_1^2\frac{\|\boldsymbol{G}_1\|^2}{\sqrt{\beta_2\widetilde{\boldsymbol{\nu}}_1}} \right.$$

$$\left. + \frac{1}{1-\beta_2}\left( \frac{147456\eta^2(L_0+L_1)\sigma_1^2\sigma_0}{(1-\beta_1)^{\frac{5}{2}}} + 4\frac{L_1\eta^2\sigma_0}{(1-\beta_1)^{\frac{3}{2}}} + \frac{24L_0\eta^2}{1-\beta_1} + 8\frac{L_0}{\sigma_0}\eta^2 \right)(\mathbb{E}\ln\boldsymbol{\nu}_T - T\ln\beta_2) \right)$$

$$\leq \sqrt{\beta_2^T \boldsymbol{\nu}_0 + 2\sigma_0^2} + \sigma_0 + \frac{1}{4}\mathbb{E}\ln\boldsymbol{\nu}_T$$

$$\leq \sqrt{\beta_2^T \boldsymbol{\nu}_0 + 2\sigma_0^2} + \sigma_0 + \frac{1}{2}\mathbb{E}\sqrt{\boldsymbol{\nu}_T + (1-\beta_2)\sigma_0^2}.$$

where the third inequality is due to

$$T \geq \frac{36 * 2048^4(L_0+L_1)^3\sigma_1^{12}(f(\boldsymbol{w}_1)-f^*)}{(1-\beta_1)^6\sigma_0^2} + \frac{768 * 2048^2(f(\boldsymbol{w}_1)-f^*)\sigma_1^8(8L_1^2(f(\boldsymbol{w}_1)-f^*)^2 + 4L_0(f(\boldsymbol{w}_1)-f^*))}{(1-\beta_1)^4\sigma_0^2}$$

$$+ \frac{24^2 * 147456(L_0+L_1)\sigma_1^8(f(\boldsymbol{w}_1)-f^*)\sigma_0^2}{(1-\beta_2)^5} + \frac{128^2(L_0+L_1)(f(\boldsymbol{w}_1)-f^*)\sigma_1^4}{\sigma_0^2}$$

$$+ \frac{24^2 * 147456 * 2048^2(L_0+L_1)^3\sigma_1^{16}(f(\boldsymbol{w}_1)-f^*)^3}{(1-\beta_2)^{11}} + \frac{128^2 * 2048^2(L_0+L_1)^3(f(\boldsymbol{w}_1)-f^*)^3\sigma^{12}}{\sigma_0^4(1-\beta_1)^6},$$

and the last inequality is due to $\ln x \leq x$. Solving the above inequality with respect to $\mathbb{E}\sqrt{\boldsymbol{\nu}_T + (1-\beta_2)\sigma_0^2}$ and applying $\boldsymbol{\nu}_0 = \sigma_0^2$ then gives

$$\mathbb{E}\sqrt{\boldsymbol{\nu}_T} \leq \mathbb{E}\sqrt{\boldsymbol{\nu}_T + (1-\beta_2)\sigma_0^2} \leq 6\sigma_0. \tag{20}$$

Therefore, Eq. (17) can be rewritten as

$$\frac{1}{16}\eta \sum_{t=1}^{T} \mathbb{E}\frac{\|\boldsymbol{G}_t\|^2}{\sqrt{\widetilde{\boldsymbol{\nu}}_t}}$$

$$\leq f(\boldsymbol{w}_1) - f^* + \eta\frac{64}{(1-\beta_1)}\sigma_1^2\frac{\|\boldsymbol{G}_1\|^2}{\sqrt{\beta_2\widetilde{\boldsymbol{\nu}}_1}}$$

$$+ \frac{1}{1-\beta_2}\left(\frac{147456\eta^2(L_0+L_1)\sigma_1^2\sigma_0}{(1-\beta_1)^{\frac{5}{2}}} + 4\frac{L_1\eta^2\sigma_0}{(1-\beta_1)^{\frac{3}{2}}} + \frac{24L_0\eta^2}{1-\beta_1} + 8\frac{L_0}{\sigma_0}\eta^2\right)(2\ln 6\sigma_0 - T\ln\beta_2).$$

$$(21)$$

We then execute the second round of divide-and-conquer. To begin with, we have that

$$\sum_{t=1}^{T}\mathbb{E}\left[\frac{\|\boldsymbol{G}_t\|^2}{\sqrt{\widetilde{\boldsymbol{\nu}}_t}}\mathbb{1}_{\|G_t\|\geq\frac{\sigma_0}{\sigma_1}}\right] \leq \sum_{t=1}^{T}\mathbb{E}\left[\frac{\|\boldsymbol{G}_t\|^2}{\sqrt{\widetilde{\boldsymbol{\nu}}_t}}\right]. \tag{22}$$

On the other hand, we have that

$$\frac{\|\boldsymbol{G}_t\|^2}{\sqrt{\widetilde{\boldsymbol{\nu}}_t}}\mathbb{1}_{\|G_t\|\geq\frac{\sigma_0}{\sigma_1}} \geq \frac{\frac{2}{3}\|\boldsymbol{G}_t\|^2 + \frac{1}{3}\frac{\sigma_0^2}{\sigma_1^2}}{\sqrt{\widetilde{\boldsymbol{\nu}}_t}}\mathbb{1}_{\|G_t\|\geq\frac{\sigma_0}{\sigma_1}} \geq \frac{\frac{\beta_2}{3\sigma_1^2}\mathbb{E}^{|\mathcal{F}_t}\|\boldsymbol{g}_t\|^2 + \frac{1-\beta_2}{3}\frac{\sigma_0^2}{\sigma_1^2}}{\sqrt{\widetilde{\boldsymbol{\nu}}_t}}\mathbb{1}_{\|G_t\|\geq\frac{\sigma_0}{\sigma_1}}$$

$$= \mathbb{E}^{|\mathcal{F}_t}\frac{\frac{\beta_2}{3\sigma_1^2}\|\boldsymbol{g}_t\|^2 + \frac{1-\beta_2}{3\sigma_1^2}\sigma_0^2}{\sqrt{\widetilde{\boldsymbol{\nu}}_t}}\mathbb{1}_{\|G_t\|\geq\frac{\sigma_0}{\sigma_1}} \geq \frac{1}{2}\mathbb{E}^{|\mathcal{F}_t}\frac{\frac{\beta_2}{3\sigma_1^2}\|\boldsymbol{g}_t\|^2 + \frac{1-\beta_2}{3\sigma_1^2}\sigma_0^2}{\sqrt{\widetilde{\boldsymbol{\nu}}_{t+1}} + \sqrt{\beta_2\widetilde{\boldsymbol{\nu}}_t}}\mathbb{1}_{\|G_t\|\geq\frac{\sigma_0}{\sigma_1}}.$$

As a conclusion,

$$\sum_{t=1}^{T}\mathbb{E}\left[\frac{\|\boldsymbol{G}_t\|^2}{\sqrt{\widetilde{\boldsymbol{\nu}}_t}}\mathbb{1}_{\|G_t\|\geq\frac{\sigma_0}{\sigma_1}}\right] \geq \frac{1}{2}\sum_{t=1}^{T}\mathbb{E}\left[\frac{\frac{\beta_2}{3\sigma_1^2}\|\boldsymbol{g}_t\|^2 + \frac{1-\beta_2}{3\sigma_1^2}\sigma_0^2}{\sqrt{\widetilde{\boldsymbol{\nu}}_{t+1}} + \sqrt{\beta_2\widetilde{\boldsymbol{\nu}}_t}}\mathbb{1}_{\|G_t\|\geq\frac{\sigma_0}{\sigma_1}}\right]$$

$$\geq \frac{1}{6(1-\beta_2)\sigma_1^2}\sum_{t=1}^{T}\mathbb{E}\left[\left(\sqrt{\widetilde{\boldsymbol{\nu}}_{t+1}} - \sqrt{\beta_2\widetilde{\boldsymbol{\nu}}_t}\right)\mathbb{1}_{\|G_t\|\geq\frac{\sigma_0}{\sigma_1}}\right].$$

Meanwhile, for convenience, we define $\{\bar{\boldsymbol{\nu}}_t\}_{t=0}^{\infty}$ as $\bar{\boldsymbol{\nu}}_0 = \boldsymbol{\nu}_0$, $\bar{\boldsymbol{\nu}}_t = \beta_2\bar{\boldsymbol{\nu}}_{t-1} + (1-\beta_2)|g_t|^2\mathbb{1}_{\|\boldsymbol{G}_t\|<\frac{\sigma_0^2}{\sigma_1^2}}$.
One can easily observe that $\bar{\boldsymbol{\nu}}_t \leq \boldsymbol{\nu}_t$, and thus

$$\sum_{t=1}^{T}\mathbb{E}\left[\left(\sqrt{\widetilde{\boldsymbol{\nu}}_{t+1}} - \sqrt{\beta_2\widetilde{\boldsymbol{\nu}}_t}\right)\mathbb{1}_{\|\boldsymbol{G}_t\|<\frac{\sigma_0^2}{\sigma_1^2}}\right]$$

$$= \sum_{t=1}^{T}\mathbb{E}\left(\sqrt{\beta_2^2\boldsymbol{\nu}_{t-1} + \beta_2(1-\beta_2)\|g_t\|^2 + (1-\beta_2)\sigma_0^2} - \sqrt{\beta_2(\beta_2\boldsymbol{\nu}_{t-1} + (1-\beta_2)\sigma_0^2)}\right)\mathbb{1}_{\|\boldsymbol{G}_t\|<\frac{\sigma_0^2}{\sigma_1^2}}$$

$$\leq \sum_{t=1}^{T}\mathbb{E}\left(\sqrt{\beta_2^2\bar{\boldsymbol{\nu}}_{t-1} + \beta_2(1-\beta_2)\|g_t\|^2 + (1-\beta_2)\sigma_0^2} - \sqrt{\beta_2(\beta_2\bar{\boldsymbol{\nu}}_{t-1} + (1-\beta_2)\sigma_0^2)}\right)\mathbb{1}_{\|\boldsymbol{G}_t\|<\frac{\sigma_0^2}{\sigma_1^2}}$$

$$\leq \sum_{t=1}^{T}\mathbb{E}\left(\sqrt{\beta_2^2\bar{\boldsymbol{\nu}}_{t-1} + \beta_2(1-\beta_2)\|g_t\|^2\mathbb{1}_{\|\boldsymbol{G}_t\|<\frac{\sigma_0^2}{\sigma_1^2}} + (1-\beta_2)\sigma_0^2} - \sqrt{\beta_2(\beta_2\bar{\boldsymbol{\nu}}_{t-1} + (1-\beta_2)\sigma_0^2)}\right)$$

$$= \sum_{t=1}^{T}\mathbb{E}\left(\sqrt{\beta_2\bar{\boldsymbol{\nu}}_t + (1-\beta_2)\sigma_0^2} - \sqrt{\beta_2(\beta_2\bar{\boldsymbol{\nu}}_{t-1} + (1-\beta_2)\sigma_0^2)}\right)$$

$$= \mathbb{E}\sqrt{\beta_2\bar{\boldsymbol{\nu}}_t + (1-\beta_2)\sigma_0^2} + (1-\sqrt{\beta_2})\sum_{t=1}^{T-1}\mathbb{E}\sqrt{\beta_2\bar{\boldsymbol{\nu}}_t + (1-\beta_2)\sigma_0^2} - \mathbb{E}\sqrt{\beta_2(\beta_2\bar{\boldsymbol{\nu}}_0 + (1-\beta_2)\sigma_0^2)}.$$

All in all, summing the above two inequalities together, we obtain that

$$\mathbb{E}\sqrt{\widetilde{\boldsymbol{\nu}}_{t+1}} + (1 - \sqrt{\beta_2})\sum_{t=2}^{T}\mathbb{E}\sqrt{\widetilde{\boldsymbol{\nu}}_t} - \sqrt{\beta_2\widetilde{\boldsymbol{\nu}}_1}$$

$$=\sum_{t=1}^{T}\mathbb{E}\left(\sqrt{\widetilde{\boldsymbol{\nu}}_t} - \sqrt{\beta_2\widetilde{\boldsymbol{\nu}}_{t-1}}\right)$$

$$\leq\sum_{t=1}^{T}\mathbb{E}\left(\sqrt{\widetilde{\boldsymbol{\nu}}_t} - \sqrt{\beta_2\widetilde{\boldsymbol{\nu}}_{t-1}}\right)\mathbb{1}_{\|\boldsymbol{G}_t\|\geq\frac{\sigma_0}{\sigma_1}} + \sum_{t=1}^{T}\mathbb{E}\left(\sqrt{\widetilde{\boldsymbol{\nu}}_t} - \sqrt{\beta_2\widetilde{\boldsymbol{\nu}}_{t-1}}\right)\mathbb{1}_{\|\boldsymbol{G}_t\|<\frac{\sigma_0^2}{\sigma_1^2}}$$

$$\leq\frac{3(1-\beta_2)\sigma_1^2}{\sqrt{\beta_2}}\sum_{t=1}^{T}\mathbb{E}\left[\frac{\|\boldsymbol{G}_t\|^2}{\sqrt{\widetilde{\boldsymbol{\nu}}_t}}\right] + \mathbb{E}\sqrt{\beta_2\bar{\boldsymbol{\nu}}_t + (1-\beta_2)\sigma_0^2} + (1 - \sqrt{\beta_2})\sum_{t=1}^{T-1}\mathbb{E}\sqrt{\beta_2\bar{\boldsymbol{\nu}}_t + (1-\beta_2)\sigma_0^2} - \sqrt{\beta_2(\beta_2\bar{\boldsymbol{\nu}}_0 + (1-\beta_2)\sigma_0^2)}.$$

Since $\forall t \geq 1$,

$$\mathbb{E}\sqrt{\beta_2\bar{\boldsymbol{\nu}}_t + (1-\beta_2)\sigma_0^2} \leq \sqrt{\beta_2\mathbb{E}\bar{\boldsymbol{\nu}}_t + (1-\beta_2)\sigma_0^2} \leq \sqrt{\sigma_0^2 + \boldsymbol{\nu}_0} \leq \sqrt{2}\sigma_0,$$

combining with $\sqrt{\beta_2\widetilde{\boldsymbol{\nu}}_1} = \sqrt{\beta_2(\beta_2\bar{\boldsymbol{\nu}}_0 + (1-\beta_2)\sigma_0^2)}$ and $\mathbb{E}\sqrt{\widetilde{\boldsymbol{\nu}}_{t+1}} = \mathbb{E}\sqrt{\beta_2\boldsymbol{\nu}_t + (1-\beta_2)\sigma_0^2} \geq \mathbb{E}\sqrt{\beta_2\bar{\boldsymbol{\nu}}_t + (1-\beta_2)\sigma_0^2}$, we obtain

$$(1 - \sqrt{\beta_2})\sum_{t=1}^{T}\mathbb{E}\sqrt{\widetilde{\boldsymbol{\nu}}_t} \leq \frac{3(1-\beta_2)\sigma_1^2}{\sqrt{\beta_2}}\sum_{t=2}^{T}\mathbb{E}\left[\frac{\|\boldsymbol{G}_t\|^2}{\sqrt{\widetilde{\boldsymbol{\nu}}_t}}\right] + +(1 - \sqrt{\beta_2})\sum_{t=1}^{T}\mathbb{E}\sqrt{\beta_2\bar{\boldsymbol{\nu}}_t + (1-\beta_2)\sigma_0^2}$$

$$\leq \frac{3(1-\beta_2)\sigma_1^2}{\sqrt{\beta_2}}\sum_{t=1}^{T}\mathbb{E}\left[\frac{\|\boldsymbol{G}_t\|^2}{\sqrt{\widetilde{\boldsymbol{\nu}}_t}}\right] + \sqrt{2}(1 - \sqrt{\beta_2})T\sigma_0..$$

Dividing both sides of the above equation by $1 - \sqrt{\beta_2}$ then gives

$$\sum_{t=1}^{T}\mathbb{E}\sqrt{\widetilde{\boldsymbol{\nu}}_t} \leq \frac{3(1-\beta_2)\sigma_1^2}{\sqrt{\beta_2}}\sum_{t=2}^{T}\mathbb{E}\left[\frac{\|\boldsymbol{G}_t\|^2}{\sqrt{\widetilde{\boldsymbol{\nu}}_t}}\right] + (1 - \sqrt{\beta_2})\sum_{t=1}^{T}\mathbb{E}\sqrt{\beta_2\bar{\boldsymbol{\nu}}_t + (1-\beta_2)\sigma_0^2}$$

$$\leq 12\sigma_1^2\sum_{t=1}^{T}\mathbb{E}\left[\frac{\|\boldsymbol{G}_t\|^2}{\sqrt{\widetilde{\boldsymbol{\nu}}_t}}\right] + \sqrt{2}T\sigma_0. \tag{23}$$

By applying Eq. (21) and the constraint of $T$, we obtain that

$$\sum_{t=1}^{T}\mathbb{E}\sqrt{\widetilde{\boldsymbol{\nu}}_t} \leq \sqrt{2}T\sigma_0 + 12\frac{\sigma_1^2}{\eta}\left(f(\boldsymbol{w}_1) - f^* + \eta\frac{64}{(1-\beta_1)}\sigma_1^2\frac{\|\boldsymbol{G}_1\|^2}{\sqrt{\beta_2\widetilde{\boldsymbol{\nu}}_1}}\right.$$

$$\left. + \frac{1}{1-\beta_2}\left(\frac{147456\eta^2(L_0 + L_1)\sigma_1^2\sigma_0}{(1-\beta_1)^{\frac{5}{2}}} + 4\frac{L_1\eta^2\sigma_0}{(1-\beta_1)^{\frac{3}{2}}} + \frac{24L_0\eta^2}{1-\beta_1} + 8\frac{L_0}{\sigma_0}\eta^2\right)(2\ln 6\sigma_0 - T\ln\beta_2)\right)$$

$$\leq 4T\sigma_0.$$

Combining the above inequality and Eq. (21) and applying Cauchy's inequality, we obtain that

$$\left(\mathbb{E}\sum_{t=1}^{T}\|\nabla f(\boldsymbol{w}_t)\|\right)^2 \leq \left(\sum_{t=1}^{T}\mathbb{E}\sqrt{\widetilde{\boldsymbol{\nu}}_t}\right)\left(\sum_{t=1}^{T}\mathbb{E}\left[\frac{\|\boldsymbol{G}_t\|^2}{\sqrt{\widetilde{\boldsymbol{\nu}}_t}}\right]\right)$$

$$\leq 4T\sigma_0 \times \frac{1}{\eta}\left(f(\boldsymbol{w}_1) - f^* + \eta\frac{64}{(1-\beta_1)}\sigma_1^2\frac{\|\boldsymbol{G}_1\|^2}{\sqrt{\beta_2\widetilde{\boldsymbol{\nu}}_1}}\right.$$

$$\left. + \frac{1}{1-\beta_2}\left(\frac{147456\eta^2(L_0 + L_1)\sigma_1^2\sigma_0}{(1-\beta_1)^{\frac{5}{2}}} + 4\frac{L_1\eta^2\sigma_0}{(1-\beta_1)^{\frac{3}{2}}} + \frac{24L_0\eta^2}{1-\beta_1} + 8\frac{L_0}{\sigma_0}\eta^2\right)(2\ln 6\sigma_0 - T\ln\beta_2)\right)$$

By $\eta = \frac{\sqrt{f(\boldsymbol{w}_1) - f^*}}{\sqrt{L_0 + L_1}\sqrt{T\sigma_0\sigma_1^2}}$ and the constraint of $T$, the proof is completed. $\qquad\square$

## C.2 PROOF FOR SGDM

**Theorem 10** (Informal). *Fix $L_0 \geq 0, L_1 > 0$, and $\Delta_1 \geq 0$, there exists objective function $f$ satisfying $(L_0, L_1)$-smooth condition and $f(\boldsymbol{w}_1) - f^* = \Delta_1$, and a noise oracle $\mathcal{O}(\boldsymbol{w}, z)$ generating stochastic gradient by $\boldsymbol{g}_t = \nabla f(\boldsymbol{w}_t) + \mathcal{O}(\boldsymbol{w}_t, z_t)$ and satisfying Assumption 2 ($z_t$ is i.i.d. sampled from some underlying distribution), such that **for any learning rate** $\eta > 0$ **and** $\beta \in [0, 1]$, for all $T > 0$,*

$$\min_{t \in [T]} \mathbb{E}\|\nabla f(\boldsymbol{w}_t)\| = \|\nabla f(\boldsymbol{w}_1)\| \geq L_1 \Delta_1.$$

*Proof.* Define the objective function $f$ as $f_1$ used in the the proof of Theorem 2 as

$$f_1(x) = \begin{cases} \dfrac{L_0 e^{L_1 x - 1}}{L_1^2} & , x \in \left[\dfrac{1}{L_1}, \infty\right), \\[2mm] \dfrac{L_0 x^2}{2} + \dfrac{L_0}{2L_1^2} & , x \in [-\dfrac{1}{L_1}, \dfrac{1}{L_1}], \\[2mm] \dfrac{L_0 e^{-L_1 x - 1}}{L_1^2} & , x \in \left(-\infty, -\dfrac{1}{L_1}\right]. \end{cases} \tag{24}$$

It is easy to verify that $f_1$ obeys Assumption 1. Then, we set $w_1$ as the solution of $f_1(x) - \frac{L_0}{2L_1^2} = \Delta_1$, thus $f(\boldsymbol{w}_1) - f^* = \Delta_1$ is satisfied. We then construct the noise oracle as $O_f(\boldsymbol{w}, z) = z$, where $z \sim e^{-\frac{\sqrt{|z|}}{\sqrt[6]{\sigma_0^2/960}}}$. One can easily verify that $\mathrm{Var}(z) = \sigma_0^2$ and Assumption 2 is meet.

Now, we prove the following claim: starting any point $w_t$ and with any previous momentum $\boldsymbol{m}_{t-1}$, one step of SGDM

$$\mathbb{E}[\|\nabla f(\boldsymbol{w}_{t+1})\| | \boldsymbol{w}_t] = \infty.$$

Specifically, we have one step of SGDM gives

$$\boldsymbol{w}_{t+1} = \boldsymbol{w}_t - \eta(1-\beta)\nabla f(\boldsymbol{w}_t) - \eta\beta\boldsymbol{m}_{t-1} - \eta(1-\beta)z_t.$$

Therefore, we have

$$\mathbb{E}[\|\nabla f(w_{t+1})\| | \boldsymbol{w}_t] \geq \mathbb{E}\left[|\nabla f(w_{t+1})| \mathbb{1}_{w_{t+1} \geq \max\{\frac{1}{L_1}, \boldsymbol{w}_t - \eta(1-\beta)\nabla f(\boldsymbol{w}_t) - \eta\beta\boldsymbol{m}_{t-1}\}}\right]$$

$$\geq \mathbb{E}\left[|\nabla f(w_{t+1})| \mathbb{1}_{z_t \leq \min\{\frac{\boldsymbol{w}_t - \frac{1}{L_1}}{\eta(1-\beta)} - \nabla f(\boldsymbol{w}_t) - \frac{\beta}{1-\beta}\boldsymbol{m}_{t-1}, 0\}}\right]$$

$$\geq \frac{1}{2} \int_{-\infty}^{\min\{\frac{\boldsymbol{w}_t - \frac{1}{L_1}}{\eta(1-\beta)} - \nabla f(\boldsymbol{w}_t) - \frac{\beta}{1-\beta}\boldsymbol{m}_{t-1}, 0\}} \frac{L_0}{L_1} e^{L_1(\boldsymbol{w}_t - \eta(1-\beta)\nabla f(\boldsymbol{w}_t) - \eta\beta\boldsymbol{m}_{t-1} - \eta(1-\beta)z) - 1} e^{-\frac{\sqrt{-z}}{\sqrt[6]{\sigma_0^2/960}}} \mathrm{d}z.$$

Since $\lim_{z \to -\infty} e^{L_1(\boldsymbol{w}_t - \eta(1-\beta)\nabla f(\boldsymbol{w}_t) - \eta\beta\boldsymbol{m}_{t-1} - \eta(1-\beta)z) - 1} e^{-\frac{\sqrt{-z}}{\sqrt[6]{\sigma_0^2/960}}} = \infty$ regardless of $\eta$, $\beta$, and $\boldsymbol{m}_{t-1}$, we have $\mathbb{E}[\|\nabla f(w_{t+1})\| | \boldsymbol{w}_t] = \infty$ based on the above inequalities. This means that an update from any point over this example will always lead to the divergence on expected gradient norm, thus we have $\forall t > 1$,

$$\min_{t \in [T]} \mathbb{E}|\nabla f(w_t)| = |\nabla f(w_1)|.$$

The proof is completed. $\qquad\square$

# D PROOFS FOR SECTION 5

## D.1 PROOF FOR THEOREM 5

**Theorem 11** (Theorem 5, restated). *Let Assumption 1 hold. Then, $\forall \beta_1 \geq 0$, if $\varepsilon \leq \frac{1}{\mathrm{Poly}(L_0, L_1, \sigma_0, \sigma_1, \frac{1}{1-\beta_1}, f(\boldsymbol{w}_1) - f^*)}$, with $\eta = (1-\beta_1)\frac{\sqrt{L_0(f(\boldsymbol{w}_1) - f^*)}}{\sqrt{T}}$ and momentum hyperparameter $\beta_2 = 1 - \eta^2 \frac{(256\sigma_1^2 L_1)^2}{1-\beta_1}$, we have that if $T \geq \Theta(\frac{L_0 \sigma_0^2 (f(\boldsymbol{w}_1) - f^*)}{\varepsilon^4})$*

$$\mathbb{E} \min_{t \in [1, T]} \|\nabla f(\boldsymbol{w}_t)\| \leq \varepsilon.$$

*Proof.* Recall that $\boldsymbol{u}_t \triangleq \frac{\boldsymbol{w}_t - \frac{\beta_1}{\sqrt{\beta_2}}\boldsymbol{w}_{t-1}}{1 - \frac{\beta_1}{\sqrt{\beta_2}}}$. and the surrogate second-order momentum be defined as

$\widetilde{\boldsymbol{\nu}}_t \triangleq \beta_2 \boldsymbol{\nu}_{t-1} + (1-\beta_2)\sigma_0^2$. Due to $\frac{\eta}{\sqrt{1-\beta_2}} \leq \frac{\sqrt{1-\beta_1}}{8L_1}$ and following the similar routine as Theorem 3, one can easily verify that

$$\|\boldsymbol{u}_t - \boldsymbol{w}_t\| \leq \frac{1}{4L_1}, \|\boldsymbol{u}_{t+1} - \boldsymbol{w}_t\| \leq \frac{1}{4L_1}.$$

Therefore, if Lemma 2 can be applied with $\boldsymbol{w}^1 = \boldsymbol{w}_t$, $\boldsymbol{w}^2 = \boldsymbol{u}_{t+1}$, and $\boldsymbol{w}^3 = \boldsymbol{u}_t$, we see the conditions of Lemma 2 is satisfied, which after taking expectation gives

$\mathbb{E}^{|\mathcal{F}_t} f(\boldsymbol{u}_{t+1})$

$\leq f(\boldsymbol{u}_t) + \mathbb{E}^{|\mathcal{F}_t}\langle \nabla f(\boldsymbol{w}_t), \boldsymbol{u}_{t+1} - \boldsymbol{u}_t\rangle + \frac{1}{2}(L_0 + L_1\|\nabla f(\boldsymbol{w}_t)\|)\mathbb{E}^{|\mathcal{F}_t}(\|\boldsymbol{u}_{t+1} - \boldsymbol{w}_t\| + \|\boldsymbol{u}_t - \boldsymbol{w}_t\|)\|\boldsymbol{u}_{t+1} - \boldsymbol{u}_t\|.$

We call $\langle \nabla f(\boldsymbol{w}_t), \boldsymbol{u}_{t+1} - \boldsymbol{u}_t\rangle$ the first-order term and $\frac{1}{2}(L_0 + L_1\|\nabla f(\boldsymbol{w}_t)\|)(\|\boldsymbol{u}_{t+1} - \boldsymbol{w}_t\| + \|\boldsymbol{u}_t - \boldsymbol{w}_t\|)\|\boldsymbol{u}_{t+1} - \boldsymbol{u}_t\|$ the second-order term, as they respectively correspond to the first-order and second-order Taylor's expansion. We then respectively bound these two terms as follows.

**Analysis for the first-order term.** Similar to bounding the first-order term in the proof of Theorem 3, we have the following decomposition :

$$\boldsymbol{u}_{t+1} - \boldsymbol{u}_t = \frac{\boldsymbol{w}_{t+1} - \boldsymbol{w}_t}{1 - \frac{\beta_1}{\sqrt{\beta_2}}} - \frac{\beta_1}{\sqrt{\beta_2}}\frac{\boldsymbol{w}_t - \boldsymbol{w}_{t-1}}{1 - \frac{\beta_1}{\sqrt{\beta_2}}}$$

$$= -\eta\frac{1-\beta_1}{1 - \frac{\beta_1}{\sqrt{\beta_2}}}\frac{1}{\sqrt{\beta_2 \boldsymbol{\nu}_{t-1}}}\boldsymbol{g}_t - \frac{\eta}{1 - \frac{\beta_1}{\sqrt{\beta_2}}}\left(\frac{1}{\sqrt{\boldsymbol{\nu}_t}} - \frac{1}{\sqrt{\beta_2 \boldsymbol{\nu}_{t-1}}}\right)\boldsymbol{m}_t.$$

According to the above decomposition, we have the first-order term can also be decomposed into

$$\mathbb{E}^{|\mathcal{F}_t}\left[\langle \nabla f(\boldsymbol{w}_t), \boldsymbol{u}_{t+1} - \boldsymbol{u}_t\rangle\right]$$

$$= \frac{1-\beta_1}{1 - \frac{\beta_1}{\sqrt{\beta_2}}}\mathbb{E}^{|\mathcal{F}_t}\left[\left\langle \boldsymbol{G}_t, -\eta\frac{1}{\sqrt{\beta_2 \boldsymbol{\nu}_{t-1}}}\boldsymbol{g}_t\right\rangle\right] + \mathbb{E}^{|\mathcal{F}_t}\left[\left\langle \boldsymbol{G}_t, -\frac{\eta}{1 - \frac{\beta_1}{\sqrt{\beta_2}}}\left(\frac{1}{\sqrt{\boldsymbol{\nu}_t}} - \frac{1}{\sqrt{\beta_2 \boldsymbol{\nu}_{t-1}}}\right)\boldsymbol{m}_t\right\rangle\right].$$

As $\mathbb{E}^{|\mathcal{F}_t}\left[\left\langle \boldsymbol{G}_t, -\eta\frac{1}{\sqrt{\beta_2 \boldsymbol{\nu}_{t-1}}}\boldsymbol{g}_t\right\rangle\right] = -\eta\frac{\|\boldsymbol{G}_t\|^2}{\sqrt{\beta_2 \boldsymbol{\nu}_{t-1}}}$, we have

$$\frac{1-\beta_1}{1 - \frac{\beta_1}{\sqrt{\beta_2}}}\mathbb{E}^{|\mathcal{F}_t}\left[\left\langle \boldsymbol{G}_t, -\eta\frac{1}{\sqrt{\beta_2 \boldsymbol{\nu}_{t-1}}}\boldsymbol{g}_t\right\rangle\right] \leq -\frac{\|\boldsymbol{G}_t\|^2}{\sqrt{\beta_2 \boldsymbol{\nu}_{t-1}}}.$$

We then bound $\mathbb{E}^{|\mathcal{F}_t}\left[\left\langle \boldsymbol{G}_t, -\frac{\eta}{1 - \frac{\beta_1}{\sqrt{\beta_2}}}\left(\frac{1}{\sqrt{\boldsymbol{\nu}_t}} - \frac{1}{\sqrt{\beta_2 \boldsymbol{\nu}_{t-1}}}\right)\boldsymbol{m}_t\right\rangle\right]$ as follows

$$\mathbb{E}^{|\mathcal{F}_t}\left[\left\langle \boldsymbol{G}_t, -\frac{\eta}{1 - \frac{\beta_1}{\sqrt{\beta_2}}}\left(\frac{1}{\sqrt{\boldsymbol{\nu}_t}} - \frac{1}{\sqrt{\beta_2 \boldsymbol{\nu}_{t-1}}}\right)\boldsymbol{m}_t\right\rangle\right]$$

$$= \mathbb{E}^{|\mathcal{F}_t}\left[\left\langle \boldsymbol{G}_t, -\frac{\eta}{1 - \frac{\beta_1}{\sqrt{\beta_2}}}\left(\frac{(1-\beta_2)\|\boldsymbol{g}_t\|^2}{\sqrt{\boldsymbol{\nu}_t}\sqrt{\beta_2 \boldsymbol{\nu}_{t-1}}(\sqrt{\boldsymbol{\nu}_t} + \sqrt{\beta_2 \boldsymbol{\nu}_{t-1}})}\right)\boldsymbol{m}_t\right\rangle\right]$$

$$\leq \frac{\eta}{1 - \frac{\beta_1}{\sqrt{\beta_2}}}\mathbb{E}^{|\mathcal{F}_t}\left[\|\boldsymbol{G}_t\|\left(\frac{(1-\beta_2)\|\boldsymbol{g}_t\|^2}{\sqrt{\boldsymbol{\nu}_t}\sqrt{\beta_2 \boldsymbol{\nu}_{t-1}}(\sqrt{\boldsymbol{\nu}_t} + \sqrt{\beta_2 \boldsymbol{\nu}_{t-1}})}\right)\|\boldsymbol{m}_t\|\right]$$

$$= \frac{\eta}{1 - \frac{\beta_1}{\sqrt{\beta_2}}}\mathbb{E}^{|\mathcal{F}_t}\left[\|\boldsymbol{G}_t\|\left(\frac{(1-\beta_2)\|\boldsymbol{g}_t\|^2}{\sqrt{\boldsymbol{\nu}_t}\sqrt{\beta_2 \boldsymbol{\nu}_{t-1}}(\sqrt{\boldsymbol{\nu}_t} + \sqrt{\beta_2 \boldsymbol{\nu}_{t-1}})}\right)\|\boldsymbol{m}_t\|\right].$$

Due to Lemma 1, the right-hand-side of the above inequality can be further bounded as

$$\frac{\eta}{1 - \frac{\beta_1}{\sqrt{\beta_2}}} \mathbb{E}^{|\mathcal{F}_t} \left[ \|\boldsymbol{G}_t\| \left( \frac{(1-\beta_2)\|\boldsymbol{g}_t\|^2}{\sqrt{\boldsymbol{\nu}_t}\sqrt{\beta_2\boldsymbol{\nu}_{t-1}}(\sqrt{\boldsymbol{\nu}_t} + \sqrt{\beta_2\boldsymbol{\nu}_{t-1}})} \right) \|\boldsymbol{m}_t\| \right] \leq \frac{\eta(1-\beta_1)}{\left(\sqrt{1 - \frac{\beta_1}{\sqrt{\beta_2}}}\right)^3} \mathbb{E}^{|\mathcal{F}_t} \left[ \|\boldsymbol{G}_t\| \left( \frac{\sqrt{1-\beta_2}\|\boldsymbol{g}_t\|^2}{\sqrt{\beta_2\boldsymbol{\nu}_{t-1}}(\sqrt{\boldsymbol{\nu}_t} + \sqrt{\beta_2\boldsymbol{\nu}_{t-1}})} \right) \right]$$

$$\overset{(\circ)}{\leq} \frac{\eta(1-\beta_1)}{\left(\sqrt{1 - \frac{\beta_1}{\sqrt{\beta_2}}}\right)^3} \frac{\|\boldsymbol{G}_t\|}{\sqrt{\beta_2\boldsymbol{\nu}_{t-1}}} \sqrt{\mathbb{E}^{|\mathcal{F}_t}\|\boldsymbol{g}_t\|^2} \sqrt{\mathbb{E}^{|\mathcal{F}_t} \frac{\|\boldsymbol{g}_t\|^2}{(\sqrt{\boldsymbol{\nu}_t} + \sqrt{\beta_2\boldsymbol{\nu}_{t-1}})^2}} \overset{(\bullet)}{\leq} \frac{\eta(1-\beta_1)\sqrt{1-\beta_2}}{\left(\sqrt{1 - \frac{\beta_1}{\sqrt{\beta_2}}}\right)^3} \frac{\|\boldsymbol{G}_t\|}{\sqrt{\beta_2\boldsymbol{\nu}_{t-1}}} \sqrt{\sigma_0^2 + \sigma_1^2\|\boldsymbol{G}_t\|^2} \sqrt{\mathbb{E}^{|\mathcal{F}_t} \frac{\|\boldsymbol{g}_t\|^2}{(\sqrt{\boldsymbol{\nu}_t} + \sqrt{\beta_2\boldsymbol{\nu}_{t-1}})^2}}$$

$$\leq \frac{\eta(1-\beta_1)\sqrt{1-\beta_2}}{\left(\sqrt{1 - \frac{\beta_1}{\sqrt{\beta_2}}}\right)^3} \frac{\|\boldsymbol{G}_t\|}{\sqrt{\beta_2\boldsymbol{\nu}_{t-1}}} (\sigma_0 + \sigma_1\|\boldsymbol{G}_t\|) \sqrt{\mathbb{E}^{|\mathcal{F}_t} \frac{\|\boldsymbol{g}_t\|^2}{(\sqrt{\boldsymbol{\nu}_t} + \sqrt{\beta_2\boldsymbol{\nu}_{t-1}})^2}},$$

where inequality $(\circ)$ is due to Holder's inequality, and inequality $(\bullet)$ is due to Assumption 2. Applying mean-value inequality respectively to $\frac{\eta(1-\beta_1)\sqrt{1-\beta_2}}{\left(\sqrt{1 - \frac{\beta_1}{\sqrt{\beta_2}}}\right)^3} \mathbb{E}^{|\mathcal{F}_t} \frac{\|\boldsymbol{G}_t\|}{\sqrt{\beta_2\boldsymbol{\nu}_{t-1}}} \sigma_0 \sqrt{\mathbb{E}^{|\mathcal{F}_t} \frac{\|\boldsymbol{g}_t\|^2}{(\sqrt{\boldsymbol{\nu}_t} + \sqrt{\beta_2\boldsymbol{\nu}_{t-1}})^2}}$ and

$\frac{\eta(1-\beta_1)\sqrt{1-\beta_2}}{\left(\sqrt{1 - \frac{\beta_1}{\sqrt{\beta_2}}}\right)^3} \mathbb{E}^{|\mathcal{F}_t} \frac{\|\boldsymbol{G}_t\|}{\sqrt{\beta_2\boldsymbol{\nu}_{t-1}}} \sigma_1 \|\boldsymbol{G}_t\| \sqrt{\mathbb{E}^{|\mathcal{F}_t} \frac{\|\boldsymbol{g}_t\|^2}{(\sqrt{\boldsymbol{\nu}_t} + \sqrt{\beta_2\boldsymbol{\nu}_{t-1}})^2}}$ and due to $\beta_1 \leq \beta_2$, we obtain that the

right-hand-side of the above inequality can be bounded by

$$\frac{1}{16}\eta \frac{1-\beta_1}{1 - \frac{\beta_1}{\sqrt{\beta_2}}} \frac{\|\boldsymbol{G}_t\|^2}{\sqrt{\beta_2\boldsymbol{\nu}_{t-1}}} + \frac{4\eta(1-\beta_2)\sigma_0^2}{\left(1 - \frac{\beta_1}{\sqrt{\beta_2}}\right)^2 \sqrt{\beta_2\boldsymbol{\nu}_{t-1}}} \mathbb{E}^{|\mathcal{F}_t} \frac{\|\boldsymbol{g}_t\|^2}{(\sqrt{\boldsymbol{\nu}_t} + \sqrt{\beta_2\boldsymbol{\nu}_{t-1}})^2}$$

$$+ \frac{1}{16}\eta \frac{1-\beta_1}{1 - \frac{\beta_1}{\sqrt{\beta_2}}} \frac{\|\boldsymbol{G}_t\|^2}{\sqrt{\beta_2\boldsymbol{\nu}_{t-1}}} + 4\eta \frac{(1-\beta_2)(1-\beta_1)}{(1 - \frac{\beta_1}{\sqrt{\beta_2}})^2} \sigma_1^2 \frac{\|\boldsymbol{G}_t\|^2}{\sqrt{\beta_2\boldsymbol{\nu}_{t-1}}} \mathbb{E}^{|\mathcal{F}_t} \frac{\|\boldsymbol{g}_t\|^2}{(\sqrt{\boldsymbol{\nu}_t} + \sqrt{\beta_2\boldsymbol{\nu}_{t-1}})^2}$$

$$\leq \frac{1}{8}\eta \frac{\|\boldsymbol{G}_t\|^2}{\sqrt{\beta_2\boldsymbol{\nu}_{t-1}}} + \frac{8\eta(1-\beta_2)\sigma_0^2}{(1-\beta_1)^2} \mathbb{E}^{|\mathcal{F}_t} \frac{\|\boldsymbol{g}_t\|^2}{\sqrt{\beta_2\boldsymbol{\nu}_{t-1}}(\sqrt{\boldsymbol{\nu}_t} + \sqrt{\beta_2\boldsymbol{\nu}_{t-1}})^2}$$

$$+ \frac{1}{8}\eta \frac{\|\boldsymbol{G}_t\|^2}{\sqrt{\beta_2\boldsymbol{\nu}_{t-1}}} + 16\eta \frac{(1-\beta_2)}{(1-\beta_1)} \sigma_1^2 \frac{\|\boldsymbol{G}_t\|^2}{\sqrt{\beta_2\boldsymbol{\nu}_{t-1}}} \mathbb{E}^{|\mathcal{F}_t} \frac{\|\boldsymbol{g}_t\|^2}{(\sqrt{\boldsymbol{\nu}_t} + \sqrt{\beta_2\boldsymbol{\nu}_{t-1}})^2}.$$

Meanwhile, we have

$$\left( \frac{1}{\sqrt{\beta_2\boldsymbol{\nu}_{t-1}}} - \frac{1}{\sqrt{\boldsymbol{\nu}_t}} \right) \|\boldsymbol{G}_t\|^2$$

$$= \frac{\|\boldsymbol{G}_t\|^2((1-\beta_2)\|\boldsymbol{g}_t\|^2)}{\sqrt{\beta_2\boldsymbol{\nu}_{t-1}}\sqrt{\boldsymbol{\nu}_t}(\sqrt{\beta_2\boldsymbol{\nu}_{t-1}} + \sqrt{\boldsymbol{\nu}_t})} \geq \frac{\|\boldsymbol{G}_t\|^2((1-\beta_2)\|\boldsymbol{g}_t\|^2)}{\sqrt{\beta_2\boldsymbol{\nu}_{t-1}}(\sqrt{\beta_2\boldsymbol{\nu}_{t-1}} + \sqrt{\boldsymbol{\nu}_t})^2},$$

and

$$\frac{1}{\sqrt{\beta_2\boldsymbol{\nu}_{t-1}}} - \frac{1}{\sqrt{\boldsymbol{\nu}_t}} = \frac{(1-\beta_2)\|\boldsymbol{g}_t\|^2}{\sqrt{\beta_2\boldsymbol{\nu}_{t-1}}\sqrt{\boldsymbol{\nu}_t}(\sqrt{\beta_2\boldsymbol{\nu}_{t-1}} + \sqrt{\boldsymbol{\nu}_t})} \geq \frac{(1-\beta_2)\|\boldsymbol{g}_t\|^2}{\sqrt{\beta_2\boldsymbol{\nu}_{t-1}}(\sqrt{\beta_2\boldsymbol{\nu}_{t-1}} + \sqrt{\boldsymbol{\nu}_t})^2}.$$

Combing the above two inequalities, we further obtain

$$\frac{\eta}{1 - \beta_1} \mathbb{E}^{|\mathcal{F}_t} \left[ \|\boldsymbol{G}_t\| \left( \frac{(1-\beta_2)\boldsymbol{g}_t^2}{\sqrt{\boldsymbol{\nu}_t}\sqrt{\beta_2\boldsymbol{\nu}_{t-1}}(\sqrt{\boldsymbol{\nu}_t} + \sqrt{\beta_2\boldsymbol{\nu}_{t-1}})} \right) \|\boldsymbol{m}_t\| \right]$$

$$\leq \frac{1}{4}\eta \frac{\|\boldsymbol{G}_t\|^2}{\sqrt{\beta_2\boldsymbol{\nu}_{t-1}}} + \frac{8\eta\sigma_0^2}{(1-\beta_1)^2} \mathbb{E}^{|\mathcal{F}_t} \left( \frac{1}{\sqrt{\beta_2\boldsymbol{\nu}_{t-1}}} - \frac{1}{\sqrt{\boldsymbol{\nu}_t}} \right) + \eta \frac{16}{(1-\beta_1)} \sigma_1^2 \mathbb{E}^{|\mathcal{F}_t} \left( \frac{1}{\sqrt{\beta_2\boldsymbol{\nu}_{t-1}}} - \frac{1}{\sqrt{\boldsymbol{\nu}_t}} \right) \|\boldsymbol{G}_t\|^2.$$

Furthermore, due to Assumption 1, we have (we define $G_0 \triangleq G_1$)

$$\|\boldsymbol{G}_{t+1}\|^2 \leq \|\boldsymbol{G}_t\|^2 + 2\|\boldsymbol{G}_t\|\|\boldsymbol{G}_{t+1} - \boldsymbol{G}_t\| + \|\boldsymbol{G}_{t+1} - \boldsymbol{G}_t\|^2$$

$$\leq \|\boldsymbol{G}_t\|^2 + 2(L_0 + L_1\|\boldsymbol{G}_t\|)\|\boldsymbol{G}_t\|\|\boldsymbol{w}_{t+1} - \boldsymbol{w}_t\| + 2(L_0^2 + L_1^2\|\boldsymbol{G}_t\|^2)\|\boldsymbol{w}_{t+1} - \boldsymbol{w}_t\|^2,$$

which further leads to

$$\frac{1}{\sqrt{\boldsymbol{\nu}_t}}\|\boldsymbol{G}_t\|^2$$

$$\geq \frac{1}{\sqrt{\boldsymbol{\nu}_t}}\left(\|\boldsymbol{G}_{t+1}\|^2 - 2(L_0 + L_1\|\boldsymbol{G}_t\|)\|\boldsymbol{G}_t\|\|\boldsymbol{w}_{t+1} - \boldsymbol{w}_t\| - 2(L_0^2 + L_1^2\|\boldsymbol{G}_t\|^2)\|\boldsymbol{w}_{t+1} - \boldsymbol{w}_t\|^2\right)$$

$$\geq \frac{1}{\sqrt{\boldsymbol{\nu}_t}}\|\boldsymbol{G}_{t+1}\|^2 - \frac{1-\beta_1}{128\sigma_1^2}\frac{\|\boldsymbol{G}_t\|^2}{\sqrt{\boldsymbol{\nu}_t}} - \frac{128L_0^2\sigma_1^2}{(1-\beta_1)\sqrt{\boldsymbol{\nu}_t}}\|\boldsymbol{w}_{t+1} - \boldsymbol{w}_t\|^2 - 2L_1\frac{\|\boldsymbol{G}_t\|^2}{\sqrt{\boldsymbol{\nu}_t}}\|\boldsymbol{w}_{t+1} - \boldsymbol{w}_t\| - 2\frac{L_0^2}{\sqrt{\boldsymbol{\nu}_t}}\|\boldsymbol{w}_{t+1} - \boldsymbol{w}_t\|^2$$

$$- \frac{2L_1^2\|\boldsymbol{G}_t\|^2\|\boldsymbol{w}_{t+1} - \boldsymbol{w}_t\|^2}{\sqrt{\boldsymbol{\nu}_t}}$$

$$\geq \frac{1}{\sqrt{\boldsymbol{\nu}_t}}\|\boldsymbol{G}_{t+1}\|^2 - \frac{1-\beta_1}{128\sigma_1^2}\frac{\|\boldsymbol{G}_t\|^2}{\sqrt{\boldsymbol{\nu}_t}} - \frac{128L_0^2\sigma_1^2}{(1-\beta_1)\sqrt{\boldsymbol{\nu}_t}}\|\boldsymbol{w}_{t+1} - \boldsymbol{w}_t\|^2 - \frac{1-\beta_1}{128\sigma_1^2}\frac{\|\boldsymbol{G}_t\|^2}{\sqrt{\boldsymbol{\nu}_t}} - 2\frac{L_0^2}{\sqrt{\boldsymbol{\nu}_t}}\|\boldsymbol{w}_{t+1} - \boldsymbol{w}_t\|^2$$

$$- \frac{1-\beta_1}{128\sigma_1^2}\frac{\|\boldsymbol{G}_t\|^2}{\sqrt{\boldsymbol{\nu}_t}}, \tag{25}$$

where the second inequality is due to Young's inequality, and the last inequality is due to $\|\boldsymbol{w}_{t+1} - \boldsymbol{w}_t\| \leq \frac{\eta(1-\beta_1)}{\sqrt{1-\beta_2}\sqrt{1-\frac{\beta_1^2}{\beta_2}}} \leq \frac{\eta\sqrt{1-\beta_1}}{\sqrt{1-\beta_2}} \leq \frac{1-\beta_1}{256\sigma_1^2 L_1}$.

Applying the above inequality back to the estimation of $\frac{\eta}{1-\beta_1}\mathbb{E}^{|\mathcal{F}_t}\left[\|\boldsymbol{G}_t\|\left(\frac{(1-\beta_2)\boldsymbol{g}_t^2}{\sqrt{\boldsymbol{\nu}_t}\sqrt{\beta_2\boldsymbol{\nu}_{t-1}}(\sqrt{\boldsymbol{\nu}_t}+\sqrt{\beta_2\boldsymbol{\nu}_{t-1}})}\right)\|\boldsymbol{m}_t\|\right]$ leads to that

$$\frac{\eta}{1-\beta_1}\mathbb{E}^{|\mathcal{F}_t}\left[\|\boldsymbol{G}_t\|\left(\frac{(1-\beta_2)\boldsymbol{g}_t^2}{\sqrt{\boldsymbol{\nu}_t}\sqrt{\beta_2\boldsymbol{\nu}_{t-1}}(\sqrt{\boldsymbol{\nu}_t}+\sqrt{\beta_2\boldsymbol{\nu}_{t-1}})}\right)\|\boldsymbol{m}_t\|\right]$$

$$\leq \frac{5}{8}\eta\frac{\|\boldsymbol{G}_t\|^2}{\sqrt{\beta_2\boldsymbol{\nu}_{t-1}}} + \frac{8\eta\sigma_0^2}{(1-\beta_1)^2}\mathbb{E}^{|\mathcal{F}_t}\left(\frac{1}{\sqrt{\beta_2\boldsymbol{\nu}_{t-1}}} - \frac{1}{\sqrt{\boldsymbol{\nu}_t}}\right) + \eta\frac{16}{(1-\beta_1)}\sigma_1^2\mathbb{E}^{|\mathcal{F}_t}\left(\frac{\|\boldsymbol{G}_t\|^2}{\sqrt{\beta_2\boldsymbol{\nu}_{t-1}}} - \frac{\|\boldsymbol{G}_{t+1}\|^2}{\sqrt{\boldsymbol{\nu}_t}}\right)$$

$$+ \frac{32768L_0^2\sigma_1^4\eta^3}{(1-\beta_1)(1-\beta_2)}\mathbb{E}^{|\mathcal{F}_t}\left(\sum_{s=1}^t\frac{\beta_1^{t-s}}{\sqrt[4]{\beta_2^{3(t-s)}}}\left(\frac{1}{\sqrt{\beta_2\boldsymbol{\nu}_{s-1}}} - \frac{1}{\sqrt{\boldsymbol{\nu}_s}}\right)\right).$$

All in all, we conclude that the first-order term can be bounded by

$$\mathbb{E}^{|\mathcal{F}_t}\left[\langle\nabla f(\boldsymbol{w}_t), \boldsymbol{u}_{t+1} - \boldsymbol{u}_t\rangle\right] \leq -\frac{3}{8}\eta\frac{\|\boldsymbol{G}_t\|^2}{\sqrt{\beta_2\boldsymbol{\nu}_{t-1}}} + \frac{8\eta\sigma_0^2}{(1-\beta_1)^2}\mathbb{E}^{|\mathcal{F}_t}\left(\frac{1}{\sqrt{\beta_2\boldsymbol{\nu}_{t-1}}} - \frac{1}{\sqrt{\boldsymbol{\nu}_t}}\right) + \eta\frac{16}{(1-\beta_1)}\sigma_1^2\mathbb{E}^{|\mathcal{F}_t}\left(\frac{\|\boldsymbol{G}_t\|^2}{\sqrt{\beta_2\boldsymbol{\nu}_{t-1}}} - \frac{\|\boldsymbol{G}_{t+1}\|^2}{\sqrt{\boldsymbol{\nu}_t}}\right)$$

$$+ \frac{32768L_0^2\sigma_1^4\eta^3}{(1-\beta_1)(1-\beta_2)}\mathbb{E}^{|\mathcal{F}_t}\left(\sum_{s=1}^t\frac{\beta_1^{t-s}}{\sqrt[4]{\beta_2^{3(t-s)}}}\left(\frac{1}{\sqrt{\beta_2\boldsymbol{\nu}_{s-1}}} - \frac{1}{\sqrt{\boldsymbol{\nu}_s}}\right)\right).$$

**Analysis for the second-order term.** To recall, the second order term is $\frac{1}{2}(L_0 + L_1\|\nabla f(\boldsymbol{w}_t)\|)(\|\boldsymbol{u}_{t+1} - \boldsymbol{w}_t\| + \|\boldsymbol{u}_t - \boldsymbol{w}_t\|)\|\boldsymbol{u}_{t+1} - \boldsymbol{u}_t\|$. Before we start, we have the following expansion for $\boldsymbol{u}_{t+1} - \boldsymbol{u}_t$: (here the operations are all coordinate-wisely)

$$\boldsymbol{u}_{t+1} - \boldsymbol{u}_t = \frac{\boldsymbol{w}_{t+1} - \boldsymbol{w}_t - \frac{\beta_1}{\sqrt{\beta_2}}(\boldsymbol{w}_t - \boldsymbol{w}_{t-1})}{1 - \frac{\beta_1}{\sqrt{\beta_2}}}$$

$$= \frac{-\eta\frac{\boldsymbol{m}_t}{\sqrt{\boldsymbol{\nu}_t}} + \eta\frac{\beta_1}{\sqrt{\beta_2}}\frac{\boldsymbol{m}_{t-1}}{\sqrt{\boldsymbol{\nu}_{t-1}}}}{1 - \frac{\beta_1}{\sqrt{\beta_2}}} = \frac{-\eta\frac{\boldsymbol{m}_t}{\sqrt{\boldsymbol{\nu}_t}} + \eta\beta_1\frac{\boldsymbol{m}_{t-1}}{\sqrt{\boldsymbol{\nu}_t}} - \eta\beta_1\frac{\boldsymbol{m}_{t-1}}{\sqrt{\boldsymbol{\nu}_t}} + \eta\frac{\beta_1}{\sqrt{\beta_2}}\frac{\boldsymbol{m}_{t-1}}{\sqrt{\boldsymbol{\nu}_{t-1}}}}{1 - \frac{\beta_1}{\sqrt{\beta_2}}}$$

$$= \frac{-\eta\frac{(1-\beta_1)\boldsymbol{g}_t}{\sqrt{\boldsymbol{\nu}_t}} + \eta\frac{\beta_1(1-\beta_2)\|\boldsymbol{g}_t\|^2}{\sqrt{\beta_2}}\frac{\boldsymbol{m}_{t-1}}{\sqrt{\boldsymbol{\nu}_{t-1}}\sqrt{\boldsymbol{\nu}_t}(\sqrt{\boldsymbol{\nu}_t}+\sqrt{\beta_2\boldsymbol{\nu}_{t-1}})}}{1 - \frac{\beta_1}{\sqrt{\beta_2}}}$$

Then firstly, we have

$$\frac{1}{2}L_0(\|\boldsymbol{u}_{t+1} - \boldsymbol{w}_t\| + \|\boldsymbol{u}_t - \boldsymbol{w}_t\|)\|\boldsymbol{u}_{t+1} - \boldsymbol{u}_t\|$$

$$\leq \frac{1}{2}L_0\left(\|\boldsymbol{u}_{t+1} - \boldsymbol{u}_t\|^2 + \frac{1}{2}\|\boldsymbol{u}_{t+1} - \boldsymbol{w}_t\|^2 + \frac{1}{2}\|\boldsymbol{u}_t - \boldsymbol{w}_t\|^2\right)$$

$$= \frac{1}{2}L_0\left(\left\|\frac{-\eta\frac{(1-\beta_1)\boldsymbol{g}_t}{\sqrt{\boldsymbol{\nu}_t}} + \eta\frac{\beta_1(1-\beta_2)\|\boldsymbol{g}_t\|^2}{\sqrt{\beta_2}}\frac{\boldsymbol{m}_{t-1}}{\sqrt{\boldsymbol{\nu}_{t-1}}\sqrt{\boldsymbol{\nu}_t}(\sqrt{\boldsymbol{\nu}_t}+\sqrt{\beta_2\boldsymbol{\nu}_{t-1}})}}{1 - \frac{\beta_1}{\sqrt{\beta_2}}}\right\|^2 + \frac{1}{2}\left\|\frac{\frac{\beta_1}{\sqrt{\beta_2}}}{1 - \frac{\beta_1}{\sqrt{\beta_2}}}(\boldsymbol{w}_t - \boldsymbol{w}_{t-1})\right\|^2 + \frac{1}{2}\left\|\frac{1}{1 - \frac{\beta_1}{\sqrt{\beta_2}}}(\boldsymbol{w}_{t+1} - \boldsymbol{w}_t)\right\|^2\right)$$

$$\leq \frac{L_0\eta^2}{2}\left(\left(\frac{1-\beta_1}{1 - \frac{\beta_1}{\sqrt{\beta_2}}} + \frac{\beta_1(1-\beta_1)}{(\sqrt{\beta_2}-\beta_1)\sqrt{1-\frac{\beta_1^2}{\beta_2}}}\right)^2\left\|\frac{\boldsymbol{g}_t}{\sqrt{\boldsymbol{\nu}_t}}\right\|^2 + \frac{1}{2}\left(\frac{\frac{\beta_1}{\sqrt{\beta_2}}}{1 - \frac{\beta_1}{\sqrt{\beta_2}}}\right)^2\left\|\frac{\boldsymbol{m}_{t-1}}{\sqrt{\boldsymbol{\nu}_{t-1}}}\right\|^2 + \frac{1}{2}\left(\frac{1}{1 - \frac{\beta_1}{\sqrt{\beta_2}}}\right)^2\left\|\frac{\boldsymbol{m}_t}{\sqrt{\boldsymbol{\nu}_t}}\right\|^2\right)$$

$$\leq \frac{L_0\eta^2}{2}\left(2\left(\frac{1-\beta_1}{1 - \frac{\beta_1}{\sqrt{\beta_2}}} + \frac{\beta_1(1-\beta_1)}{(\sqrt{\beta_2}-\beta_1)\sqrt{1-\frac{\beta_1^2}{\beta_2}}}\right)^2\left\|\frac{\boldsymbol{g}_t}{\sqrt{\boldsymbol{\nu}_t}}\right\|^2 + \left(\frac{\frac{\beta_1}{\sqrt{\beta_2}}}{1 - \frac{\beta_1}{\sqrt{\beta_2}}}\right)^2\left\|\frac{\boldsymbol{m}_{t-1}}{\sqrt{\boldsymbol{\nu}_{t-1}}}\right\|^2\right)$$

$$\leq \frac{L_0\eta^2}{2}\left(\frac{32}{(1-\beta_1)^2}\left\|\frac{\boldsymbol{g}_t}{\sqrt{\boldsymbol{\nu}_t}}\right\|^2 + \frac{4}{(1-\beta_1)^2}\left\|\frac{\boldsymbol{m}_{t-1}}{\sqrt{\boldsymbol{\nu}_{t-1}}}\right\|^2\right).$$

Secondly, we have

$$\frac{1}{2}L_1\|\nabla f(\boldsymbol{w}_t)\|(\|\boldsymbol{u}_{t+1} - \boldsymbol{w}_t\| + \|\boldsymbol{u}_t - \boldsymbol{w}_t\|)\|\boldsymbol{u}_{t+1} - \boldsymbol{u}_t\|$$

$$\leq \frac{1}{2}L_1\|\nabla f(\boldsymbol{w}_t)\|(2\|\boldsymbol{u}_{t+1} - \boldsymbol{w}_t\| + \|\boldsymbol{u}_{t+1} - \boldsymbol{u}_t\|)\left(\frac{\left\|\eta\frac{(1-\beta_1)\boldsymbol{g}_t}{\sqrt{\boldsymbol{\nu}_t}}\right\|}{1 - \frac{\beta_1}{\sqrt{\beta_2}}} + \frac{\eta\frac{\beta_1(1-\beta_2)\|\boldsymbol{g}_t\|^2}{\sqrt{\beta_2}}\frac{\|\boldsymbol{m}_{t-1}\|}{\sqrt{\boldsymbol{\nu}_{t-1}}\sqrt{\boldsymbol{\nu}_t}(\sqrt{\boldsymbol{\nu}_t}+\sqrt{\beta_2\boldsymbol{\nu}_{t-1}})}}{1 - \frac{\beta_1}{\sqrt{\beta_2}}}\right)$$

$$\overset{(*)}{\leq} \frac{1}{2}L_1\|\nabla f(\boldsymbol{w}_t)\|(2\|\boldsymbol{u}_{t+1} - \boldsymbol{w}_t\| + \|\boldsymbol{u}_{t+1} - \boldsymbol{u}_t\|)\left(\frac{\left\|\eta\frac{(1-\beta_1)\boldsymbol{g}_t}{\sqrt{\boldsymbol{\nu}_t}}\right\|}{1 - \frac{\beta_1}{\sqrt{\beta_2}}} + \frac{\eta\frac{\beta_1(1-\beta_1)}{\sqrt{\beta_2}}\frac{\|\boldsymbol{g}_t\|}{\sqrt{\boldsymbol{\nu}_t}}}{(1 - \frac{\beta_1}{\sqrt{\beta_2}})\sqrt{1-\frac{\beta_1^2}{\beta_2}}}\right)$$

$$= \frac{L_1}{2}\eta\left(\frac{1-\beta_1}{1 - \frac{\beta_1}{\sqrt{\beta_2}}} + \frac{\beta_1(1-\beta_1)}{(\sqrt{\beta_2}-\beta_1)\sqrt{1-\frac{\beta_1^2}{\beta_2}}}\right)\|\nabla f(\boldsymbol{w}_t)\|(2\|\boldsymbol{u}_{t+1} - \boldsymbol{w}_t\| + \|\boldsymbol{u}_t - \boldsymbol{u}_{t+1}\|)\frac{\|\boldsymbol{g}_t\|}{\sqrt{\boldsymbol{\nu}_t}}$$

$$\overset{(\circ)}{=} \frac{L_1}{2}\eta\left(\frac{1-\beta_1}{1 - \frac{\beta_1}{\sqrt{\beta_2}}} + \frac{\beta_1(1-\beta_1)}{(\sqrt{\beta_2}-\beta_1)\sqrt{1-\frac{\beta_1^2}{\beta_2}}}\right)\|\boldsymbol{G}_t\|\left(\|\boldsymbol{u}_{t+1} - \boldsymbol{u}_t\| + 2\frac{1}{1 - \frac{\beta_1}{\sqrt{\beta_2}}}\eta\left\|\frac{\boldsymbol{m}_t}{\sqrt{\boldsymbol{\nu}_t}}\right\|\right)\frac{\|\boldsymbol{g}_t\|}{\sqrt{\boldsymbol{\nu}_t}}$$

$$\leq \frac{2L_1\eta}{\sqrt{1-\beta_1}}\|\boldsymbol{G}_t\|\left(\|\boldsymbol{u}_{t+1} - \boldsymbol{u}_t\| + 4\frac{1}{1-\beta_1}\eta\left\|\frac{\boldsymbol{m}_t}{\sqrt{\boldsymbol{\nu}_t}}\right\|\right)\frac{\|\boldsymbol{g}_t\|}{\sqrt{\boldsymbol{\nu}_t}}.$$

where inequality $(*)$ is due to that $\frac{\|\boldsymbol{m}_{t-1}\|}{\sqrt{\boldsymbol{\nu}_{t-1}}} \leq \frac{1-\beta_1}{\sqrt{1-\beta_2}\sqrt{1-\frac{\beta_1^2}{\beta_2}}}$, $\frac{\|\boldsymbol{g}_t\|}{\sqrt{\boldsymbol{\nu}_t}} \leq \frac{1}{\sqrt{1-\beta_2}}$, equation $(\circ)$ is due to

$\boldsymbol{u}_t - \boldsymbol{w}_t = \frac{\frac{\beta_1}{\sqrt{\beta_2}}}{1 - \frac{\beta_1}{\sqrt{\beta_2}}}(\boldsymbol{w}_t - \boldsymbol{w}_{t-1})$ and $\boldsymbol{u}_{t+1} - \boldsymbol{w}_t = \frac{1}{1 - \frac{\beta_1}{\sqrt{\beta_2}}}(\boldsymbol{w}_{t+1} - \boldsymbol{w}_t)$, and the last inequality is

due to $\beta_1 \leq \beta_2$. As for the term $\|\boldsymbol{G}_t\|\frac{\|\boldsymbol{m}_t\|}{\sqrt{\boldsymbol{\nu}_t}}\frac{\|\boldsymbol{g}_t\|}{\sqrt{\boldsymbol{\nu}_t}}$, we have

$$\mathbb{E}^{|\mathcal{F}_t}\frac{\|\boldsymbol{G}_t\|\|\boldsymbol{m}_t\|\|\boldsymbol{g}_t\|}{\boldsymbol{\nu}_t} \leq \mathbb{E}^{|\mathcal{F}_t}\frac{\|\boldsymbol{G}_t\|\|\boldsymbol{m}_t\|\|\boldsymbol{g}_t\|}{\sqrt{\boldsymbol{\nu}_t}\sqrt{\beta_2\boldsymbol{\nu}_{t-1}}} \leq \frac{\|\boldsymbol{G}_t\|}{\sqrt{\beta_2\boldsymbol{\nu}_{t-1}}}\sqrt{\mathbb{E}^{|\mathcal{F}_t}\|\boldsymbol{g}_t\|^2}\sqrt{\mathbb{E}^{|\mathcal{F}_t}\frac{\|\boldsymbol{m}_t\|^2}{\boldsymbol{\nu}_t}}$$

$$\leq \frac{\|\boldsymbol{G}_t\|}{\sqrt{\beta_2\boldsymbol{\nu}_{t-1}}}(\sigma_0 + \sigma_1\|\boldsymbol{G}_t\|)\sqrt{\mathbb{E}^{|\mathcal{F}_t}\frac{\|\boldsymbol{m}_t\|^2}{\boldsymbol{\nu}_t}}$$

$$\leq \frac{\sqrt{(1-\beta_1)^3}}{256\eta L_1}\frac{\|\boldsymbol{G}_t\|^2}{\sqrt{\beta_2\boldsymbol{\nu}_{t-1}}} + \frac{64\eta\sigma_0^2 L_1}{\sqrt{(1-\beta_1)^3}\sqrt{\beta_2\boldsymbol{\nu}_{t-1}}}\mathbb{E}^{|\mathcal{F}_t}\frac{\|\boldsymbol{m}_t\|^2}{\boldsymbol{\nu}_t} + \frac{\sqrt{(1-\beta_1)^3}}{256\eta L_1}\frac{\|\boldsymbol{G}_t\|^2}{\sqrt{\beta_2\boldsymbol{\nu}_{t-1}}}$$

$$+ \frac{64\eta\sigma_1^2 L_1\|\boldsymbol{G}_t\|^2}{\sqrt{(1-\beta_1)^3}\sqrt{\beta_2\boldsymbol{\nu}_{t-1}}}\mathbb{E}^{|\mathcal{F}_t}\frac{\|\boldsymbol{m}_t\|^2}{\boldsymbol{\nu}_t}.$$

By applying Lemma 5, we further obtain

$$\mathbb{E}^{|\mathcal{F}_t} \frac{\|\boldsymbol{G}_t\|\|\boldsymbol{m}_t\|\|\boldsymbol{g}_t\|}{\boldsymbol{\nu}_t}$$

$$\leq \frac{\sqrt{(1-\beta_1)^3}}{256\eta L_1} \frac{\|\boldsymbol{G}_t\|^2}{\sqrt{\beta_2 \boldsymbol{\nu}_{t-1}}} + \frac{64\eta\sigma_0^2 L_1}{\sqrt{(1-\beta_1)^3}\sqrt{\beta_2 \boldsymbol{\nu}_{t-1}}} \mathbb{E}^{|\mathcal{F}_t} \frac{\|\boldsymbol{m}_t\|^2}{\boldsymbol{\nu}_t} + \frac{\sqrt{(1-\beta_1)^3}}{256\eta L_1} \frac{\|\boldsymbol{G}_t\|^2}{\sqrt{\beta_2 \boldsymbol{\nu}_{t-1}}}$$

$$+ \frac{64\eta\sigma_1^2 L_1}{\sqrt{(1-\beta_1)^3}} \mathbb{E}^{|\mathcal{F}_t} \left( 4(1-\beta_1) \left( \sum_{s=1}^{t} \frac{\sqrt[8]{\beta_1^{t-s}} \|\boldsymbol{g}_s\|^2 \|\boldsymbol{G}_s\|^2}{\sqrt{\beta_2 \boldsymbol{\nu}_{s-1}}\sqrt{\boldsymbol{\nu}_s^2}} \right) + 8\frac{1-\beta_1}{1-\beta_2}\frac{L_1^2}{L_0^2} \left( \sum_{s=1}^{t} \sqrt[8]{\beta_1^{t-s}} \left( \frac{1}{\sqrt{\beta_2 \boldsymbol{\nu}_{s-1}}} - \frac{1}{\sqrt{\boldsymbol{\nu}_s}} \right) \right) \right) \right),$$

which further indicates that

$$\frac{8L_1\eta^2}{(1-\beta_1)^{\frac{3}{2}}} \mathbb{E}^{|\mathcal{F}_t} \|\boldsymbol{G}_t\| \frac{\|\boldsymbol{g}_t\|}{\sqrt{\boldsymbol{\nu}_t}} \frac{\|\boldsymbol{m}_t\|}{\sqrt{\boldsymbol{\nu}_t}}$$

$$\leq \frac{1}{16}\eta \frac{\|\boldsymbol{G}_t\|^2}{\sqrt{\beta_2 \boldsymbol{\nu}_{t-1}}} + \frac{8L_1\eta^2}{(1-\beta_1)^{\frac{3}{2}}} \frac{64\eta\sigma_0^2 L_1}{\sqrt{(1-\beta_1)^3}\sqrt{\beta_2 \boldsymbol{\nu}_{t-1}}} \mathbb{E}^{|\mathcal{F}_t} \frac{\|\boldsymbol{m}_t\|^2}{\boldsymbol{\nu}_t}$$

$$+ \frac{64\eta\sigma_1^2 L_1}{\sqrt{(1-\beta_1)^3}} \mathbb{E}^{|\mathcal{F}_t} \frac{32L_1\eta^2}{(1-\beta_1)^{\frac{1}{2}}} \left( \sum_{s=1}^{t} \frac{\sqrt[8]{\beta_1^{t-s}} \|\boldsymbol{g}_s\|^2 \|\boldsymbol{G}_s\|^2}{\sqrt{\beta_2 \boldsymbol{\nu}_{s-1}}\sqrt{\boldsymbol{\nu}_s^2}} \right)$$

$$+ \frac{64L_1\eta^2}{(1-\beta_1)^{\frac{3}{2}}} \frac{64\eta\sigma_1^2 L_1}{\sqrt{(1-\beta_1)^3}} \frac{1-\beta_1}{1-\beta_2}\frac{L_1^2}{L_0^2} \left( \sum_{s=1}^{t} \sqrt[8]{\beta_1^{t-s}} \left( \frac{1}{\sqrt{\beta_2 \boldsymbol{\nu}_{s-1}}} - \frac{1}{\sqrt{\boldsymbol{\nu}_s}} \right) \right)$$

$$\leq \frac{1}{16}\eta \frac{\|\boldsymbol{G}_t\|^2}{\sqrt{\beta_2 \boldsymbol{\nu}_{t-1}}} + \frac{8L_1\eta^2}{(1-\beta_1)^{\frac{3}{2}}} \frac{64\eta\sigma_0^2 L_1}{\sqrt{(1-\beta_1)^3}} \mathbb{E}^{|\mathcal{F}_t} 4(1-\beta_1) \left( \sum_{s=1}^{t} \sqrt[4]{\beta_1^{t-s}} \frac{2}{1-\beta_2} \left( \frac{1}{\sqrt{\beta_2 \boldsymbol{\nu}_{s-1}}} - \frac{1}{\sqrt{\boldsymbol{\nu}_s}} \right) \right)$$

$$+ \frac{64\eta\sigma_1^2 L_1}{\sqrt{(1-\beta_1)^3}} \mathbb{E}^{|\mathcal{F}_t} \frac{32L_1\eta^2}{(1-\beta_1)^{\frac{1}{2}}} \left( \sum_{s=1}^{t} \frac{\sqrt[8]{\beta_1^{t-s}} \|\boldsymbol{g}_s\|^2 \|\boldsymbol{G}_s\|^2}{\sqrt{\beta_2 \boldsymbol{\nu}_{s-1}}\sqrt{\boldsymbol{\nu}_s^2}} \right)$$

$$+ \frac{64L_1\eta^2}{(1-\beta_1)^{\frac{3}{2}}} \frac{64\eta\sigma_1^2 L_1}{\sqrt{(1-\beta_1)^3}} \frac{1-\beta_1}{1-\beta_2}\frac{L_1^2}{L_0^2} \left( \sum_{s=1}^{t} \sqrt[8]{\beta_1^{t-s}} \left( \frac{1}{\sqrt{\beta_2 \boldsymbol{\nu}_{s-1}}} - \frac{1}{\sqrt{\boldsymbol{\nu}_s}} \right) \right).$$

Here the last inequality is due to Lemma 4.

Following similar reasoning, we have $\|\boldsymbol{u}_{t+1} - \boldsymbol{u}_t\| \leq \frac{4\eta}{\sqrt{1-\beta_2}} \frac{\|\boldsymbol{g}_t\|}{\sqrt{\boldsymbol{\nu}_t}}$, and

$$\mathbb{E}^{|\mathcal{F}_t} \frac{\|\boldsymbol{G}_t\|\|\boldsymbol{g}_t\|\|\boldsymbol{g}_t\|}{\boldsymbol{\nu}_t}$$

$$\leq \frac{\sqrt{(1-\beta_1)^3}}{256\eta L_1} \frac{\|\boldsymbol{G}_t\|^2}{\sqrt{\beta_2 \boldsymbol{\nu}_{t-1}}} + \frac{64\eta\sigma_0^2 L_1}{\sqrt{(1-\beta_1)^3}\sqrt{\beta_2 \boldsymbol{\nu}_{t-1}}} \mathbb{E}^{|\mathcal{F}_t} \frac{\|\boldsymbol{g}_t\|^2}{\boldsymbol{\nu}_t} + \frac{\sqrt{(1-\beta_1)^3}}{256\eta L_1} \frac{\|\boldsymbol{G}_t\|^2}{\sqrt{\beta_2 \boldsymbol{\nu}_{t-1}}}$$

$$+ \frac{64\eta\sigma_1^2 L_1 \|\boldsymbol{G}_t\|^2}{\sqrt{(1-\beta_1)^3}\sqrt{\beta_2 \boldsymbol{\nu}_{t-1}}} \mathbb{E}^{|\mathcal{F}_t} \frac{\|\boldsymbol{g}_t\|^2}{\boldsymbol{\nu}_t}$$

$$\leq \frac{\sqrt{(1-\beta_1)^3}}{256\eta L_1} \frac{\|\boldsymbol{G}_t\|^2}{\sqrt{\beta_2 \boldsymbol{\nu}_{t-1}}} + \frac{128\eta\sigma_1^2 L_1}{\sqrt{(1-\beta_1)^3}(1-\beta_2)} \mathbb{E}^{|\mathcal{F}_t} \left( \frac{1}{\sqrt{\beta_2 \boldsymbol{\nu}_{t-1}}} - \frac{1}{\sqrt{\boldsymbol{\nu}_t}} \right) + \frac{\sqrt{(1-\beta_1)^3}}{256\eta L_1} \frac{\|\boldsymbol{G}_t\|^2}{\sqrt{\beta_2 \boldsymbol{\nu}_{t-1}}}$$

$$+ \frac{128\eta\sigma_1^2 L_1 \|\boldsymbol{G}_t\|^2}{\sqrt{(1-\beta_1)^3}(1-\beta_2)} \mathbb{E}^{|\mathcal{F}_t} \left( \frac{1}{\sqrt{\beta_2 \boldsymbol{\nu}_{t-1}}} - \frac{1}{\sqrt{\boldsymbol{\nu}_t}} \right).$$

Then, following the similar routine as Eq. (25) and due to $\frac{\eta}{\sqrt{1-\beta_2}} \leq \frac{1-\beta_1}{128L_1\sigma_1}$, we have

$$\frac{2L_1\eta}{(1-\beta_1)^{\frac{1}{2}}}\mathbb{E}^{|\mathcal{F}_t}\|\boldsymbol{G}_t\|\|\boldsymbol{u}_{t+1}-\boldsymbol{u}_t\|\frac{\|\boldsymbol{g}_t\|}{\sqrt{\boldsymbol{\nu}_t}} \leq \frac{8L_1\eta^2}{(1-\beta_1)}\mathbb{E}^{|\mathcal{F}_t}\|\boldsymbol{G}_t\|\frac{\|\boldsymbol{g}_t\|^2}{\boldsymbol{\nu}_t}$$

$$\leq \frac{1}{16}\frac{\|\boldsymbol{G}_t\|^2}{\sqrt{\beta_2\boldsymbol{\nu}_{t-1}}} + \frac{8L_1\eta^2}{(1-\beta_1)^{\frac{3}{2}}}\frac{128\eta\sigma_1^2L_1}{\sqrt{(1-\beta_1)^3(1-\beta_2)}}\mathbb{E}^{|\mathcal{F}_t}\left(\frac{1}{\sqrt{\beta_2\boldsymbol{\nu}_{t-1}}}-\frac{1}{\sqrt{\boldsymbol{\nu}_t}}\right)$$

$$+ \frac{8L_1\eta^2}{(1-\beta_1)^{\frac{3}{2}}}\frac{128\eta\sigma_1^2L_1}{\sqrt{(1-\beta_1)^3(1-\beta_2)}}\mathbb{E}^{|\mathcal{F}_t}\left(\frac{\|\boldsymbol{G}_t\|^2}{\sqrt{\beta_2\boldsymbol{\nu}_{t-1}}}-\frac{\|\boldsymbol{G}_{t+1}\|^2}{\sqrt{\boldsymbol{\nu}_t}}\right)$$

$$+ \frac{1}{16}\eta\frac{\|\boldsymbol{G}_t\|^2}{\sqrt{\beta_2\boldsymbol{\nu}_{t-1}}} + \eta^3\frac{64L_0^2}{(1-\beta_1)^2}\mathbb{E}^{|\mathcal{F}_t}4(1-\beta_1)\left(\sum_{s=1}^t \sqrt[4]{\beta_1^{t-s}}\frac{2}{1-\beta_2}\left(\frac{1}{\sqrt{\beta_2\boldsymbol{\nu}_{s-1}}}-\frac{1}{\sqrt{\boldsymbol{\nu}_s}}\right)\right).$$

Putting all the estimations together, we have that the second-order term can be bounded by (note here due to the complexity of coefficients, we use $\mathrm{Poly}(L_0, L_1, \sigma_0, \sigma_1, \frac{1}{1-\beta_1}, f(\boldsymbol{w}_1) - f^*)(L_0, L_1, \sigma_0, \sigma_1, \frac{1}{1-\beta_1})$ to denote the polynomial function of $L_0, L_1, \sigma_0, \sigma_1, \frac{1}{1-\beta_1}$)

$$\mathbb{E}^{|\mathcal{F}_t}\frac{1}{2}(L_0 + L_1\|\nabla f(\boldsymbol{w}_t)\|)(\|\boldsymbol{u}_{t+1}-\boldsymbol{w}_t\|+\|\boldsymbol{u}_t-\boldsymbol{w}_t\|)\|\boldsymbol{u}_{t+1}-\boldsymbol{u}_t\|$$

$$\leq \frac{L_0\eta^2}{2}\left(\frac{32}{(1-\beta_1)^2}\left\|\frac{\boldsymbol{g}_t}{\sqrt{\boldsymbol{\nu}_t}}\right\|^2 + \frac{4}{(1-\beta_1)^2}\left\|\frac{\boldsymbol{m}_{t-1}}{\sqrt{\boldsymbol{\nu}_{t-1}}}\right\|^2\right) + \frac{3}{16}\eta\frac{\|\boldsymbol{G}_t\|^2}{\sqrt{\beta_2\boldsymbol{\nu}_{t-1}}}$$

$$+ \frac{\eta^3}{1-\beta_2}\mathrm{Poly}(L_0, L_1, \sigma_0, \sigma_1, \frac{1}{1-\beta_1}, f(\boldsymbol{w}_1) - f^*)\mathbb{E}^{|\mathcal{F}_t}\left(\sum_{s=1}^t \sqrt[8]{\beta_1^{t-s}}\left(\frac{1}{\sqrt{\beta_2\boldsymbol{\nu}_{s-1}}}-\frac{1}{\sqrt{\boldsymbol{\nu}_s}}\right)\right)$$

$$+ \frac{\eta^3}{1-\beta_2}\mathrm{Poly}(L_0, L_1, \sigma_0, \sigma_1, \frac{1}{1-\beta_1}, f(\boldsymbol{w}_1) - f^*)\mathbb{E}^{|\mathcal{F}_t}\left(\frac{\|\boldsymbol{G}_t\|^2}{\sqrt{\beta_2\boldsymbol{\nu}_{t-1}}}-\frac{\|\boldsymbol{G}_{t+1}\|^2}{\sqrt{\boldsymbol{\nu}_t}}\right)$$

$$+ \frac{64\eta\sigma_1^2L_1}{\sqrt{(1-\beta_1)^3}}\mathbb{E}^{|\mathcal{F}_t}\frac{32L_1\eta^2}{(1-\beta_1)^{\frac{1}{2}}}\left(\sum_{s=1}^t \frac{\sqrt[8]{\beta_1^{t-s}}\|\boldsymbol{g}_s\|^2\|\boldsymbol{G}_s\|^2}{\sqrt{\beta_2\boldsymbol{\nu}_{s-1}}\sqrt{\boldsymbol{\nu}_s^2}}\right)$$

Here in the second inequality we use $\beta_2 \geq \beta_1$, and in the last inequality we use $\frac{\eta}{\sqrt{1-\beta_2}} = \frac{\sqrt{1-\frac{\beta_1^2}{\beta_2}(1-\frac{\beta_1}{\sqrt{\beta_2}})^2}}{1024\sigma_1^2(L_1+L_0)(1-\beta_1)}$.

Applying the estimations of the first-order term and the second-order term back into the descent lemma, we derive that

$$\mathbb{E}^{|\mathcal{F}_t}f(\boldsymbol{u}_{t+1}) \leq f(\boldsymbol{u}_t) - \frac{3}{16}\eta\mathbb{E}^{|\mathcal{F}_t}\frac{\|\boldsymbol{G}_t\|^2}{\sqrt{\beta_2\boldsymbol{\nu}_{t-1}}} + \frac{L_0\eta^2}{2}\left(\frac{32}{(1-\beta_1)^2}\left\|\frac{\boldsymbol{g}_t}{\sqrt{\boldsymbol{\nu}_t}}\right\|^2 + \frac{4}{(1-\beta_1)^2}\left\|\frac{\boldsymbol{m}_{t-1}}{\sqrt{\boldsymbol{\nu}_{t-1}}}\right\|^2\right)$$

$$+ \frac{\eta^3}{1-\beta_2}\mathrm{Poly}(L_0, L_1, \sigma_0, \sigma_1, \frac{1}{1-\beta_1}, f(\boldsymbol{w}_1) - f^*)\mathbb{E}^{|\mathcal{F}_t}\left(\sum_{s=1}^t \sqrt[8]{\beta_1^{t-s}}\left(\frac{1}{\sqrt{\beta_2\boldsymbol{\nu}_{s-1}}}-\frac{1}{\sqrt{\boldsymbol{\nu}_s}}\right)\right)$$

$$+ \left(\frac{\eta^3}{1-\beta_2} + \eta\right)\mathrm{Poly}(L_0, L_1, \sigma_0, \sigma_1, \frac{1}{1-\beta_1}, f(\boldsymbol{w}_1) - f^*)\mathbb{E}^{|\mathcal{F}_t}\left(\frac{\|\boldsymbol{G}_t\|^2}{\sqrt{\beta_2\boldsymbol{\nu}_{t-1}}}-\frac{\|\boldsymbol{G}_{t+1}\|^2}{\sqrt{\boldsymbol{\nu}_t}}\right)$$

$$+ \eta\,\mathrm{Poly}(L_0, L_1, \sigma_0, \sigma_1, \frac{1}{1-\beta_1}, f(\boldsymbol{w}_1) - f^*)\mathbb{E}^{|\mathcal{F}_t}\left(\frac{\|\boldsymbol{G}_t\|^2}{\sqrt{\beta_2\boldsymbol{\nu}_{t-1}}}-\frac{\|\boldsymbol{G}_{t+1}\|^2}{\sqrt{\boldsymbol{\nu}_t}}\right)$$

$$+ \frac{64\eta\sigma_1^2L_1}{\sqrt{(1-\beta_1)^3}}\mathbb{E}^{|\mathcal{F}_t}\frac{32L_1\eta^2}{(1-\beta_1)^{\frac{1}{2}}}\left(\sum_{s=1}^t \frac{\sqrt[8]{\beta_1^{t-s}}\|\boldsymbol{g}_s\|^2\|\boldsymbol{G}_s\|^2}{\sqrt{\beta_2\boldsymbol{\nu}_{s-1}}\sqrt{\boldsymbol{\nu}_s^2}}\right). \tag{26}$$

Constructing stopping time as $\tau \triangleq \min\{t : \|\nabla f(\boldsymbol{w}_{t+1})\| \leq \frac{\sqrt[4]{L_0 \sigma_0^2 (f(\boldsymbol{w}_1) - f^*)}}{\sqrt[4]{T}}\} \wedge T$. Then, denote

$$
\begin{aligned}
x_t \triangleq &f(\boldsymbol{u}_t) - f(\boldsymbol{u}_{t+1}) - \frac{3}{16}\eta\mathbb{E}\frac{\|\boldsymbol{G}_t\|^2}{\sqrt{\beta_2 \boldsymbol{\nu}_{t-1}}} + \frac{L_0 \eta^2}{2}\left(\frac{32}{(1-\beta_1)^2}\left\|\frac{\boldsymbol{g}_t}{\sqrt{\boldsymbol{\nu}_t}}\right\|^2 + \frac{4}{(1-\beta_1)^2}\left\|\frac{\boldsymbol{m}_{t-1}}{\sqrt{\boldsymbol{\nu}_{t-1}}}\right\|^2\right) \\
&+ \frac{\eta^3}{1-\beta_2}\operatorname{Poly}(L_0, L_1, \sigma_0, \sigma_1, \frac{1}{1-\beta_1}, f(\boldsymbol{w}_1) - f^*)\left(\sum_{s=1}^t \sqrt[8]{\beta_1^{t-s}}\left(\frac{1}{\sqrt{\beta_2 \boldsymbol{\nu}_{s-1}}} - \frac{1}{\sqrt{\boldsymbol{\nu}_s}}\right)\right) \\
&+ \left(\frac{\eta^3}{1-\beta_2} + \eta\right)\operatorname{Poly}(L_0, L_1, \sigma_0, \sigma_1, \frac{1}{1-\beta_1}, f(\boldsymbol{w}_1) - f^*)\left(\frac{\|\boldsymbol{G}_t\|^2}{\sqrt{\beta_2 \boldsymbol{\nu}_{t-1}}} - \frac{\|\boldsymbol{G}_{t+1}\|^2}{\sqrt{\boldsymbol{\nu}_t}}\right) \\
&+ \frac{64\eta\sigma_1^2 L_1}{\sqrt{(1-\beta_1)^3}}\frac{32L_1\eta^2}{(1-\beta_1)^{\frac{1}{2}}}\left(\sum_{s=1}^t \frac{\sqrt[8]{\beta_1^{t-s}}\|\boldsymbol{g}_s\|^2\|\boldsymbol{G}_s\|^2}{\sqrt{\beta_2 \boldsymbol{\nu}_{s-1}}\sqrt{\boldsymbol{\nu}_s^2}}\right),
\end{aligned}
$$

and due to Eq. (26), we have $\mathbb{E}^{|\mathcal{F}_t} x_t \geq 0$, and thus $S_t \triangleq \sum_{s=1}^t x_s$ ($S_0 = 0$) is a submartingale with respect to $\{\mathcal{F}_t\}_t$. Also, as $\tau$ is a bounding stopping theorem, by optional stopping time, we obtain that $\mathbb{E}S_\tau \geq 0$, which gives

$$
\begin{aligned}
\frac{3}{16}\eta\mathbb{E}\sum_{t=1}^\tau \frac{\|\boldsymbol{G}_t\|^2}{\sqrt{\beta_2 \boldsymbol{\nu}_{t-1}}} \leq &f(\boldsymbol{u}_1) - \mathbb{E}f(\boldsymbol{u}_{\tau+1}) + \frac{L_0 \eta^2}{2}\mathbb{E}\sum_{t=1}^\tau\left(\frac{32}{(1-\beta_1)^2}\left\|\frac{\boldsymbol{g}_t}{\sqrt{\boldsymbol{\nu}_t}}\right\|^2 + \frac{4}{(1-\beta_1)^2}\left\|\frac{\boldsymbol{m}_{t-1}}{\sqrt{\boldsymbol{\nu}_{t-1}}}\right\|^2\right) \\
&+ \frac{\eta^3}{1-\beta_2}\operatorname{Poly}(L_0, L_1, \sigma_0, \sigma_1, \frac{1}{1-\beta_1}, f(\boldsymbol{w}_1) - f^*)\mathbb{E}\sum_{t=1}^\tau\left(\sum_{s=1}^t \sqrt[8]{\beta_1^{t-s}}\left(\frac{1}{\sqrt{\beta_2 \boldsymbol{\nu}_{s-1}}} - \frac{1}{\sqrt{\boldsymbol{\nu}_s}}\right)\right) \\
&+ \mathbb{E}\sum_{t=1}^\tau\left(\frac{\eta^3}{1-\beta_2} + \eta\right)\operatorname{Poly}(L_0, L_1, \sigma_0, \sigma_1, \frac{1}{1-\beta_1}, f(\boldsymbol{w}_1) - f^*)\left(\frac{\|\boldsymbol{G}_t\|^2}{\sqrt{\beta_2 \boldsymbol{\nu}_{t-1}}} - \frac{\|\boldsymbol{G}_{t+1}\|^2}{\sqrt{\boldsymbol{\nu}_t}}\right) \\
&+ \mathbb{E}\sum_{t=1}^\tau \frac{64\eta\sigma_1^2 L_1}{\sqrt{(1-\beta_1)^3}}\frac{32L_1\eta^2}{(1-\beta_1)^{\frac{1}{2}}}\left(\sum_{s=1}^t \frac{\sqrt[8]{\beta_1^{t-s}}\|\boldsymbol{g}_s\|^2\|\boldsymbol{G}_s\|^2}{\sqrt{\beta_2 \boldsymbol{\nu}_{s-1}}\sqrt{\boldsymbol{\nu}_s^2}}\right) \\
\leq &f(\boldsymbol{u}_1) - \mathbb{E}f(\boldsymbol{u}_{\tau+1}) + \frac{L_0 \eta^2}{2}\mathbb{E}\sum_{t=1}^\tau\left(\frac{32}{(1-\beta_1)^2}\left\|\frac{\boldsymbol{g}_t}{\sqrt{\boldsymbol{\nu}_t}}\right\|^2 + \frac{4}{(1-\beta_1)^2}\left\|\frac{\boldsymbol{m}_{t-1}}{\sqrt{\boldsymbol{\nu}_{t-1}}}\right\|^2\right) \\
&+ \frac{\eta^3}{1-\beta_2}\operatorname{Poly}(L_0, L_1, \sigma_0, \sigma_1, \frac{1}{1-\beta_1}, f(\boldsymbol{w}_1) - f^*)\mathbb{E}\sum_{t=1}^\tau\left(\frac{1}{\sqrt{\beta_2 \boldsymbol{\nu}_{t-1}}} - \frac{1}{\sqrt{\boldsymbol{\nu}_t}}\right) \\
&+ \mathbb{E}\sum_{t=1}^\tau\left(\frac{\eta^3}{1-\beta_2} + \eta\right)\operatorname{Poly}(L_0, L_1, \sigma_0, \sigma_1, \frac{1}{1-\beta_1}, f(\boldsymbol{w}_1) - f^*)\left(\frac{\|\boldsymbol{G}_t\|^2}{\sqrt{\beta_2 \boldsymbol{\nu}_{t-1}}} - \frac{\|\boldsymbol{G}_{t+1}\|^2}{\sqrt{\boldsymbol{\nu}_t}}\right) \\
&+ \mathbb{E}\sum_{t=1}^\tau \frac{64\eta\sigma_1^2 L_1}{\sqrt{(1-\beta_1)^3}}\frac{256L_1\eta^2}{(1-\beta_1)^{\frac{3}{2}}}\left(\frac{\|\boldsymbol{g}_t\|^2\|\boldsymbol{G}_t\|^2}{\sqrt{\beta_2 \boldsymbol{\nu}_{t-1}}\sqrt{\boldsymbol{\nu}_t^2}}\right) \\
\leq &f(\boldsymbol{u}_1) - \mathbb{E}f(\boldsymbol{u}_{\tau+1}) + \frac{L_0 \eta^2}{2}\mathbb{E}\sum_{t=1}^\tau\left(\frac{32}{(1-\beta_1)^2}\left\|\frac{\boldsymbol{g}_t}{\sqrt{\boldsymbol{\nu}_t}}\right\|^2 + \frac{4}{(1-\beta_1)^2}\left\|\frac{\boldsymbol{m}_{t-1}}{\sqrt{\boldsymbol{\nu}_{t-1}}}\right\|^2\right) \\
&+ \frac{\eta^3}{1-\beta_2}\operatorname{Poly}(L_0, L_1, \sigma_0, \sigma_1, \frac{1}{1-\beta_1}, f(\boldsymbol{w}_1) - f^*)\mathbb{E}\sum_{t=1}^\tau\left(\frac{1}{\sqrt{\beta_2 \boldsymbol{\nu}_{t-1}}} - \frac{1}{\sqrt{\boldsymbol{\nu}_t}}\right) \\
&+ \mathbb{E}\sum_{t=1}^\tau\left(\frac{\eta^3}{1-\beta_2} + \eta\right)\operatorname{Poly}(L_0, L_1, \sigma_0, \sigma_1, \frac{1}{1-\beta_1}, f(\boldsymbol{w}_1) - f^*)\left(\frac{\|\boldsymbol{G}_t\|^2}{\sqrt{\beta_2 \boldsymbol{\nu}_{t-1}}} - \frac{\|\boldsymbol{G}_{t+1}\|^2}{\sqrt{\boldsymbol{\nu}_t}}\right) \\
&+ \frac{1}{16}\eta\mathbb{E}\sum_{t=1}^\tau \frac{\|\boldsymbol{G}_t\|^2}{\sqrt{\beta_2 \boldsymbol{\nu}_{t-1}}},
\end{aligned}
$$

where the last inequality is because due to $\frac{\eta}{\sqrt{1-\beta_2}} \leq \frac{1-\beta_1}{128 L_1 \sigma_1}$, following the similar reasoning of Eq. (25), we have

$$\frac{64\eta\sigma_1^2 L_1}{\sqrt{(1-\beta_1)^3}} \frac{256 L_1 \eta^2}{(1-\beta_1)^{\frac{3}{2}}} \left( \frac{\|\boldsymbol{g}_t\|^2 \|\boldsymbol{G}_t\|^2}{\sqrt{\beta_2 \boldsymbol{\nu}_{t-1}} \sqrt{\boldsymbol{\nu}_t^2}} \right)$$

$$\leq \frac{128\eta\sigma_1^2 L_1}{\sqrt{(1-\beta_1)^3}} \frac{256 L_1 \eta^2}{(1-\beta_2)(1-\beta_1)^{\frac{3}{2}}} \left( \frac{\|\boldsymbol{G}_t\|^2}{\sqrt{\beta_2 \boldsymbol{\nu}_{t-1}}} - \frac{\|\boldsymbol{G}_{t+1}\|^2}{\sqrt{\boldsymbol{\nu}_t}} \right)$$

$$+ \frac{\eta^3}{1-\beta_2} \operatorname{Poly}\left(L_0, L_1, \sigma_0, \sigma_1, \frac{1}{1-\beta_1}, f(\boldsymbol{w}_1) - f^*\right) \sum_{s=1}^{t} \sqrt[8]{\beta_1^{t-s}} \left( \frac{1}{\sqrt{\beta_2 \boldsymbol{\nu}_{t-1}}} - \frac{1}{\sqrt{\boldsymbol{\nu}_t}} \right)$$

$$+ \frac{1}{16} \eta \mathbb{E} \sum_{t=1}^{\tau} \frac{\|\boldsymbol{G}_t\|^2}{\sqrt{\beta_2 \boldsymbol{\nu}_{t-1}}}.$$

By rearranging the inequality and due to Lemma 3, we obtain

$$\frac{1}{8} \eta \mathbb{E} \sum_{t=1}^{\tau} \frac{\|\boldsymbol{G}_t\|^2}{\sqrt{\beta_2 \boldsymbol{\nu}_{t-1}}}$$

$$\leq f(\boldsymbol{u}_1) - \mathbb{E} f(\boldsymbol{u}_{\tau+1}) + \frac{64 L_0 \eta^2}{(1-\beta_2)(1-\beta_1)^2} \mathbb{E} \left( \ln \frac{\boldsymbol{\nu}_\tau}{\boldsymbol{\nu}_0} - \tau \ln \beta_2 \right)$$

$$+ \frac{\eta^3}{1-\beta_2} \operatorname{Poly}\left(L_0, L_1, \sigma_0, \sigma_1, \frac{1}{1-\beta_1}, f(\boldsymbol{w}_1) - f^*\right) \frac{1}{\sqrt{\beta_2 \boldsymbol{\nu}_0}} + \eta^3 \operatorname{Poly}\left(L_0, L_1, \sigma_0, \sigma_1, \frac{1}{1-\beta_1}, f(\boldsymbol{w}_1) - f^*\right) \mathbb{E} \sum_{t=1}^{\tau-1} \frac{1}{\sqrt{\beta_2 \boldsymbol{\nu}_t}}$$

$$+ \left( \frac{\eta^3}{1-\beta_2} + \eta \right) \operatorname{Poly}\left(L_0, L_1, \sigma_0, \sigma_1, \frac{1}{1-\beta_1}, f(\boldsymbol{w}_1) - f^*\right) \left( \frac{\|\boldsymbol{G}_1\|^2}{\sqrt{\beta_2 \boldsymbol{\nu}_0}} \right)$$

$$+ \mathbb{E} \sum_{t=1}^{\tau} \left( \eta^3 + \eta(1-\beta_2) \right) \operatorname{Poly}\left(L_0, L_1, \sigma_0, \sigma_1, \frac{1}{1-\beta_1}, f(\boldsymbol{w}_1) - f^*\right) \left( \frac{\|\boldsymbol{G}_t\|^2}{\sqrt{\beta_2 \boldsymbol{\nu}_{t-1}}} \right),$$

Furthermore, as $T \geq \operatorname{Poly}(L_0, L_1, \sigma_0, \sigma_1, \frac{1}{1-\beta_1}, f(\boldsymbol{w}_1) - f^*)$, $\eta^3 = \frac{\eta}{T} \operatorname{Poly}(L_0, L_1, \sigma_0, \sigma_1, \frac{1}{1-\beta_1}, f(\boldsymbol{w}_1) - f^*)$, $\eta(1-\beta_2) = \frac{\eta}{T} \operatorname{Poly}(L_0, L_1, \sigma_0, \sigma_1, \frac{1}{1-\beta_1}, f(\boldsymbol{w}_1) - f^*)$, and $\|\boldsymbol{G}_t\| \geq \frac{\operatorname{Poly}(L_0, L_1, \sigma_0, \sigma_1, \frac{1}{1-\beta_1}, f(\boldsymbol{w}_1) - f^*)}{\sqrt[4]{T}}$, we have

$$\eta^3 \operatorname{Poly}\left(L_0, L_1, \sigma_0, \sigma_1, \frac{1}{1-\beta_1}, f(\boldsymbol{w}_1) - f^*\right) \mathbb{E} \sum_{t=1}^{\tau-1} \frac{1}{\sqrt{\beta_2 \boldsymbol{\nu}_t}} \leq \frac{1}{32} \eta \mathbb{E} \sum_{t=1}^{\tau} \frac{\|\boldsymbol{G}_t\|^2}{\sqrt{\beta_2 \boldsymbol{\nu}_{t-1}}},$$

$$\mathbb{E} \sum_{t=1}^{\tau} \left( \eta^3 + \eta(1-\beta_2) \right) \operatorname{Poly}\left(L_0, L_1, \sigma_0, \sigma_1, \frac{1}{1-\beta_1}, f(\boldsymbol{w}_1) - f^*\right) \left( \frac{\|\boldsymbol{G}_t\|^2}{\sqrt{\beta_2 \boldsymbol{\nu}_{t-1}}} \right) \leq \frac{1}{32} \eta \mathbb{E} \sum_{t=1}^{\tau} \frac{\|\boldsymbol{G}_t\|^2}{\sqrt{\beta_2 \boldsymbol{\nu}_{t-1}}},$$

which thus leads to

$$\frac{1}{16} \eta \mathbb{E} \sum_{t=1}^{\tau} \frac{\|\boldsymbol{G}_t\|^2}{\sqrt{\beta_2 \boldsymbol{\nu}_{t-1}}} \leq f(\boldsymbol{u}_1) - \mathbb{E} f(\boldsymbol{u}_{\tau+1}) + \frac{64 L_0 \eta^2}{(1-\beta_2)(1-\beta_1)^2} \mathbb{E} \left( \ln \frac{\boldsymbol{\nu}_\tau}{\boldsymbol{\nu}_0} - \tau \ln \beta_2 \right)$$

$$+ \frac{\eta^3}{1-\beta_2} \operatorname{Poly}\left(L_0, L_1, \sigma_0, \sigma_1, \frac{1}{1-\beta_1}, f(\boldsymbol{w}_1) - f^*\right) \frac{1}{\sqrt{\beta_2 \boldsymbol{\nu}_0}} + \left( \frac{\eta^3}{1-\beta_2} + \eta \right) \operatorname{Poly}\left(L_0, L_1, \sigma_0, \sigma_1, \frac{1}{1-\beta_1}, f(\boldsymbol{w}_1) - f^*\right) \left( \frac{\|\boldsymbol{G}_1\|^2}{\sqrt{\beta_2 \boldsymbol{\nu}_0}} \right). \tag{27}$$

Similar to the proof of Theorem 3, we then transfer the above bound to the bound of $\sum_{t=1}^{\tau} \|\boldsymbol{G}_t\|$ by two rounds of divide-and-conquer. In the first round, we will bound $\mathbb{E} \ln \boldsymbol{\nu}_\tau$. To start with, we have that

$$\frac{\|\boldsymbol{G}_t\|^2}{\sqrt{\beta_2 \boldsymbol{\nu}_{t-1}}} \mathbb{1}_{\|G_t\| \geq \frac{\sigma_0}{\sigma_1}} \geq \frac{\frac{1}{2\sigma_1^2} \mathbb{E}^{|\mathcal{F}_t} \|\boldsymbol{g}_t\|^2}{\sqrt{\beta_2 \boldsymbol{\nu}_{t-1}}} \mathbb{1}_{\|G_t\| \geq \frac{\sigma_0}{\sigma_1}}$$

Furthermore, we have

$$\mathbb{E}\frac{\frac{\boldsymbol{\nu}_0}{1-\beta_2}}{\sqrt{\sum_{t=1}^{\tau}\|\boldsymbol{g}_t\|^2 + \frac{\boldsymbol{\nu}_0}{1-\beta_2}}} + \mathbb{E}\sum_{t=1}^{\tau}\frac{\|\boldsymbol{g}_t\|^2}{\sqrt{\sum_{t=1}^{\tau}\|\boldsymbol{g}_t\|^2 + \frac{\boldsymbol{\nu}_0}{1-\beta_2}}}\mathbb{1}_{\|\boldsymbol{G}_t\|<\frac{\sigma_0}{\sigma_1}}$$

$$\leq\mathbb{E}\frac{\frac{\boldsymbol{\nu}_0}{1-\beta_2}}{\sqrt{\sum_{t=1}^{\tau}\|\boldsymbol{g}_t\|^2\mathbb{1}_{\|\boldsymbol{G}_t\|<\frac{\sigma_0}{\sigma_1}} + \frac{\boldsymbol{\nu}_0}{1-\beta_2}}} + \mathbb{E}\sum_{t=1}^{\tau}\frac{\|\boldsymbol{g}_t\|^2}{\sqrt{\sum_{t=1}^{\tau}\|\boldsymbol{g}_t\|^2\mathbb{1}_{\|\boldsymbol{G}_t\|<\frac{\sigma_0}{\sigma_1}} + \frac{\boldsymbol{\nu}_0}{1-\beta_2}}}\mathbb{1}_{\|\boldsymbol{G}_t\|<\frac{\sigma_0}{\sigma_1}}$$

$$=\mathbb{E}\sqrt{\sum_{t=1}^{\tau}\|\boldsymbol{g}_t\|^2\mathbb{1}_{\|\boldsymbol{G}_t\|<\frac{\sigma_0}{\sigma_1}} + \frac{\boldsymbol{\nu}_0}{1-\beta_2}} \leq \sqrt{\mathbb{E}\sum_{t=1}^{\tau}\|\boldsymbol{g}_t\|^2\mathbb{1}_{\|\boldsymbol{G}_t\|<\frac{\sigma_0}{\sigma_1}} + \frac{\boldsymbol{\nu}_0}{1-\beta_2}}$$

$$\overset{(\star)}{\leq}\sqrt{2\sigma_0^2\mathbb{E}\tau + \frac{\boldsymbol{\nu}_0}{1-\beta_2}} \leq \sqrt{2\sigma_0^2 T + \frac{\boldsymbol{\nu}_0}{1-\beta_2}},$$

where inequality $(\star)$ is due to $E^{|\mathcal{F}_t}\|\boldsymbol{g}_t\|^2\mathbb{1}_{\|\boldsymbol{G}_t\|<\frac{\sigma_0}{\sigma_1}} \leq 2\sigma_0^2$ and optimal stopping theorem.

Conclusively, we obtain

$$\mathbb{E}\sqrt{\sum_{t=1}^{\tau}\|\boldsymbol{g}_t\|^2 + \frac{\boldsymbol{\nu}_0}{1-\beta_2}}$$

$$=\left(\mathbb{E}\frac{\frac{\boldsymbol{\nu}_0}{1-\beta_2}}{\sqrt{\sum_{t=1}^{\tau}\|\boldsymbol{g}_t\|^2 + \frac{\boldsymbol{\nu}_0}{1-\beta_2}}} + \mathbb{E}\sum_{t=1}^{\tau}\frac{\|\boldsymbol{g}_t\|^2}{\sqrt{\sum_{t=1}^{\tau}\|\boldsymbol{g}_t\|^2 + \frac{\boldsymbol{\nu}_0}{1-\beta_2}}}\mathbb{1}_{\|\boldsymbol{G}_t\|<\frac{\sigma_0}{\sigma_1}}\right.$$

$$\left.+ \mathbb{E}\sum_{t=1}^{\tau}\frac{\|\boldsymbol{g}_t\|^2}{\sqrt{\sum_{t=1}^{\tau}\|\boldsymbol{g}_t\|^2 + \frac{\boldsymbol{\nu}_0}{1-\beta_2}}}\mathbb{1}_{\|\boldsymbol{G}_t\|\geq\frac{\sigma_0}{\sigma_1}}\right)$$

$$\leq\sqrt{\frac{\boldsymbol{\nu}_0}{1-\beta_2} + 2\sigma_0^2 T} + 2\sqrt{1-\beta_2}\mathbb{E}\sum_{t=1}^{\tau}\frac{\|\boldsymbol{g}_t\|^2}{\sqrt{\beta_2\boldsymbol{\nu}_{t-1}}}\mathbb{1}_{\|\boldsymbol{G}_t\|\geq\frac{\sigma_0}{\sigma_1}}$$

$$\overset{(\circ)}{\leq}\sqrt{\frac{\boldsymbol{\nu}_0}{1-\beta_2} + 2\sigma_0^2 T} + 2\sigma_1^2\sqrt{1-\beta_2}\mathbb{E}\sum_{t=1}^{\tau}\frac{\|\boldsymbol{G}_t\|^2}{\sqrt{\beta_2\boldsymbol{\nu}_{t-1}}},$$

where inequality $(\circ)$ is due to optimal stopping theorem.

Then by substituting $\mathbb{E}\sum_{t=1}^{\tau}\frac{\|\boldsymbol{G}_t\|^2}{\sqrt{\beta_2\boldsymbol{\nu}_{t-1}}}$ we obtain that

$$\mathbb{E}\sqrt{\sum_{t=1}^{\tau}\|\boldsymbol{g}_t\|^2 + \frac{\boldsymbol{\nu}_0}{1-\beta_2}}$$

$$\leq\sqrt{\frac{\boldsymbol{\nu}_0}{1-\beta_2} + 2\sigma_0^2 T} + \frac{32\sigma_1^2\sqrt{1-\beta_2}}{\eta}\frac{1}{16}\eta\mathbb{E}\sum_{t=1}^{\tau}\frac{\|\boldsymbol{G}_t\|^2}{\sqrt{\beta_2\boldsymbol{\nu}_{t-1}}}$$

$$\leq\sqrt{\frac{\boldsymbol{\nu}_0}{1-\beta_2} + 2\sigma_0^2 T} + \frac{32\sqrt{1-\beta_2}\sigma_1^2}{\eta}\left(f(\boldsymbol{u}_1) - f^* + \frac{64L_0\eta^2}{(1-\beta_2)(1-\beta_1)^2}\mathbb{E}\left(\ln\frac{\boldsymbol{\nu}_\tau}{\boldsymbol{\nu}_0} - T\ln\beta_2\right)\right.$$

$$\left.+ +\frac{\eta^3}{1-\beta_2}\text{Poly}\left(L_0, L_1, \sigma_0, \sigma_1, \frac{1}{1-\beta_1}, f(\boldsymbol{w}_1) - f^*\right)\frac{1}{\sqrt{\beta_2\boldsymbol{\nu}_0}}\left(\frac{\eta^3}{1-\beta_2} + \eta\right)\text{Poly}\left(L_0, L_1, \sigma_0, \sigma_1, \frac{1}{1-\beta_1}, f(\boldsymbol{w}_1) - f^*\right)\left(\frac{\|\boldsymbol{G}_1\|^2}{\sqrt{\beta_2\boldsymbol{\nu}_0}}\right)\right)$$

$$\leq\sqrt{\frac{\boldsymbol{\nu}_0}{1-\beta_2} + 2\sigma_0^2 T} + \frac{32\sqrt{1-\beta_2}\sigma_1^2}{\eta}\left(f(\boldsymbol{w}_1) - f^* + \frac{128L_0\eta^2}{(1-\beta_2)(1-\beta_1)^2}\left(\ln\frac{\mathbb{E}\sqrt{1-\beta_2}\sqrt{\sum_{t=1}^{\tau}\|\boldsymbol{g}_t\|^2 + \frac{\boldsymbol{\nu}_0}{1-\beta_2}}}{\sqrt{\boldsymbol{\nu}_0}} - T\ln\beta_2\right)\right.$$

$$\left.+ \frac{\eta^3}{1-\beta_2}\text{Poly}\left(L_0, L_1, \sigma_0, \sigma_1, \frac{1}{1-\beta_1}, f(\boldsymbol{w}_1) - f^*\right)\frac{1}{\sqrt{\beta_2\boldsymbol{\nu}_0}} + \left(\frac{\eta^3}{1-\beta_2} + \eta\right)\text{Poly}\left(L_0, L_1, \sigma_0, \sigma_1, \frac{1}{1-\beta_1}, f(\boldsymbol{w}_1) - f^*\right)\left(\frac{\|\boldsymbol{G}_1\|^2}{\sqrt{\beta_2\boldsymbol{\nu}_0}}\right)\right).$$

Multiplying both sides of the above inequality by $\sqrt{1-\beta_2}$ then gives

$$\sqrt{1-\beta_2}\mathbb{E}\sqrt{\sum_{t=1}^{\tau}\|\boldsymbol{g}_t\|^2 + \frac{\boldsymbol{\nu}_0}{1-\beta_2}} \leq 3\sqrt{\boldsymbol{\nu}_0 + 2\sigma_0^2 T(1-\beta_2)} + \frac{1}{4}\ln\mathbb{E}\sqrt{1-\beta_2}\sqrt{\sum_{t=1}^{\tau}\|\boldsymbol{g}_t\|^2 + \frac{\boldsymbol{\nu}_0}{1-\beta_2}},$$

where we use $\eta = \frac{\text{Poly}(L_0,L_1,\sigma_0,\sigma_1,\frac{1}{1-\beta_1},f(\boldsymbol{w}_1)-f^*)}{\sqrt{T}}$, $1 - \beta_2 = \frac{\text{Poly}(L_0,L_1,\sigma_0,\sigma_1,\frac{1}{1-\beta_1},f(\boldsymbol{w}_1)-f^*)}{T}$, and $T \geq \text{Poly}(L_0, L_1, \sigma_0, \sigma_1, \frac{1}{1-\beta_1}, f(\boldsymbol{w}_1) - f^*)$. Therefore, we obtain $\sqrt{1-\beta_2}\mathbb{E}\sqrt{\sum_{t=1}^{\tau} \|\boldsymbol{g}_t\|^2 + \frac{\boldsymbol{\nu}_0}{1-\beta_2}} \leq 6\sqrt{\boldsymbol{\nu}_0 + 2\sigma_0^2 T(1-\beta_2)}$.

Therefore, Eq. (27) can be rewritten as

$$
\frac{1}{16}\eta\mathbb{E}\sum_{t=1}^{\tau} \frac{\|\boldsymbol{G}_t\|^2}{\sqrt{\beta_2\boldsymbol{\nu}_{t-1}}} \leq f(\boldsymbol{u}_1) - \mathbb{E}f(\boldsymbol{u}_{\tau+1}) + \frac{128L_0\eta^2}{(1-\beta_2)(1-\beta_1)^2}\left(\ln\frac{6\sqrt{\boldsymbol{\nu}_0 + 2\sigma_0^2 T(1-\beta_2)}}{\sqrt{\boldsymbol{\nu}_0}} - T\ln\beta_2\right)
$$

$$
+ \frac{\eta^3}{1-\beta_2}\text{Poly}(L_0, L_1, \sigma_0, \sigma_1, \frac{1}{1-\beta_1}, f(\boldsymbol{w}_1) - f^*)\frac{1}{\sqrt{\beta_2\boldsymbol{\nu}_0}}
$$

$$
+ \left(\frac{\eta^3}{1-\beta_2} + \eta\right)\text{Poly}(L_0, L_1, \sigma_0, \sigma_1, \frac{1}{1-\beta_1}, f(\boldsymbol{w}_1) - f^*)\left(\frac{\|\boldsymbol{G}_1\|^2}{\sqrt{\beta_2\boldsymbol{\nu}_0}}\right).
$$

We then execute the second round of divide-and-conquer. To begin with, we have that

$$
\mathbb{E}\sum_{t=1}^{\tau}\left[\frac{\|\boldsymbol{G}_t\|^2}{\sqrt{\beta_2\boldsymbol{\nu}_{t-1}}}\mathbb{1}_{\|G_t\|\geq\frac{\sigma_0}{\sigma_1}}\right] \leq \mathbb{E}\sum_{t=1}^{\tau}\left[\frac{\|\boldsymbol{G}_t\|^2}{\sqrt{\beta_2\boldsymbol{\nu}_{t-1}}}\right].
$$

On the other hand, we have that

$$
\frac{\|\boldsymbol{G}_t\|^2}{\sqrt{\beta_2\boldsymbol{\nu}_{t-1}}}\mathbb{1}_{\|G_t\|\geq\frac{\sigma_0}{\sigma_1}} \geq \frac{1}{2\sigma_1^2}\mathbb{E}^{|\mathcal{F}_t}\frac{\|\boldsymbol{g}_t\|^2}{\sqrt{\boldsymbol{\nu}_t} + \sqrt{\beta_2\boldsymbol{\nu}_{t-1}}}\mathbb{1}_{\|G_t\|\geq\frac{\sigma_0}{\sigma_1}} \geq \frac{1}{2(1-\beta_2)\sigma_1^2}\mathbb{E}^{|\mathcal{F}_t}\left[\left(\sqrt{\boldsymbol{\nu}_t} - \sqrt{\beta_2\boldsymbol{\nu}_{t-1}}\right)\mathbb{1}_{\|G_t\|\geq\frac{\sigma_0}{\sigma_1}}\right].
$$

Meanwhile, recall $\{\bar{\boldsymbol{\nu}}_t\}_{t=0}^{\infty}$ as $\bar{\boldsymbol{\nu}}_0 = \boldsymbol{\nu}_0$, $\bar{\boldsymbol{\nu}}_t = \beta_2\bar{\boldsymbol{\nu}}_{t-1} + (1-\beta_2)|g_t|^2\mathbb{1}_{\|\boldsymbol{G}_t\|<\frac{\sigma_0^2}{\sigma_1^2}}$ and $\bar{\boldsymbol{\nu}}_t \leq \boldsymbol{\nu}_t$.

Therefore

$$
\mathbb{E}\sum_{t=1}^{\tau}\left[\left(\sqrt{\boldsymbol{\nu}_t} - \sqrt{\beta_2\boldsymbol{\nu}_{t-1}}\right)\mathbb{1}_{\|\boldsymbol{G}_t\|<\frac{\sigma_0^2}{\sigma_1^2}}\right]
$$

$$
= \mathbb{E}\sum_{t=1}^{\tau}\left(\sqrt{\beta_2\boldsymbol{\nu}_{t-1} + (1-\beta_2)\|g_t\|^2} - \sqrt{\beta_2\boldsymbol{\nu}_{t-1}}\right)\mathbb{1}_{\|\boldsymbol{G}_t\|<\frac{\sigma_0^2}{\sigma_1^2}}
$$

$$
\leq \mathbb{E}\sum_{t=1}^{\tau}\left(\sqrt{\beta_2^2\bar{\boldsymbol{\nu}}_{t-1} + \beta_2(1-\beta_2)\|g_t\|^2 + (1-\beta_2)\sigma_0^2} - \sqrt{\beta_2(\beta_2\bar{\boldsymbol{\nu}}_{t-1} + (1-\beta_2)\sigma_0^2)}\right)\mathbb{1}_{\|\boldsymbol{G}_t\|<\frac{\sigma_0^2}{\sigma_1^2}}
$$

$$
\leq \mathbb{E}\sum_{t=1}^{\tau}\left(\sqrt{\beta_2\bar{\boldsymbol{\nu}}_{t-1} + (1-\beta_2)\|g_t\|^2\mathbb{1}_{\|\boldsymbol{G}_t\|<\frac{\sigma_0^2}{\sigma_1^2}}} - \sqrt{\beta_2\bar{\boldsymbol{\nu}}_{t-1}}\right)
$$

$$
= \mathbb{E}\sum_{t=1}^{\tau}\left(\sqrt{\bar{\boldsymbol{\nu}}_t} - \sqrt{\beta_2\bar{\boldsymbol{\nu}}_{t-1}}\right)
$$

$$
= \mathbb{E}\sqrt{\bar{\boldsymbol{\nu}}_\tau} + (1-\sqrt{\beta_2})\sum_{t=1}^{\tau-1}\mathbb{E}\sqrt{\bar{\boldsymbol{\nu}}_t} - \mathbb{E}\sqrt{\beta_2\boldsymbol{\nu}_0}
$$

$$
\leq \mathbb{E}\sqrt{\bar{\boldsymbol{\nu}}_\tau} + (1-\sqrt{\beta_2})\sum_{t=1}^{T}\mathbb{E}\sqrt{\bar{\boldsymbol{\nu}}_t} - \mathbb{E}\sqrt{\beta_2\boldsymbol{\nu}_0}
$$

$$
\leq \mathbb{E}\sqrt{\bar{\boldsymbol{\nu}}_\tau} + (1-\sqrt{\beta_2})T\sigma_0 - \mathbb{E}\sqrt{\beta_2\boldsymbol{\nu}_0}.
$$

All in all, summing the above two inequalities together, we obtain that

$$
\mathbb{E}\sqrt{\boldsymbol{\nu}_\tau} + (1 - \sqrt{\beta_2})\mathbb{E}\sum_{t=1}^{\tau-1}\sqrt{\boldsymbol{\nu}_t} - \sqrt{\beta_2\boldsymbol{\nu}_0}
$$

$$
=\mathbb{E}\sum_{t=1}^{\tau}\left(\sqrt{\boldsymbol{\nu}_t} - \sqrt{\beta_2\boldsymbol{\nu}_{t-1}}\right)
$$

$$
\leq\mathbb{E}\sum_{t=1}^{\tau}\left(\sqrt{\boldsymbol{\nu}_t} - \sqrt{\beta_2\boldsymbol{\nu}_{t-1}}\right)\mathbb{1}_{\|\boldsymbol{G}_t\|\geq\frac{\sigma_0}{\sigma_1}} + \mathbb{E}\sum_{t=1}^{\tau}\left(\sqrt{\boldsymbol{\nu}_t} - \sqrt{\beta_2\boldsymbol{\nu}_{t-1}}\right)\mathbb{1}_{\|\boldsymbol{G}_t\|<\frac{\sigma_0^2}{\sigma_1^2}}
$$

$$
\leq 2(1-\beta_2)\sigma_1^2\sum_{t=1}^{\tau}\mathbb{E}\left[\frac{\|\boldsymbol{G}_t\|^2}{\sqrt{\beta_2\boldsymbol{\nu}_{t-1}}}\right] + \mathbb{E}\sqrt{\bar{\boldsymbol{\nu}}_\tau} + (1-\sqrt{\beta_2})T\sigma_0 - \mathbb{E}\sqrt{\beta_2\boldsymbol{\nu}_0}.
$$

Since $\boldsymbol{\nu}_0 = \bar{\boldsymbol{\nu}}_0$ and $\mathbb{E}\sqrt{\boldsymbol{\nu}_\tau} \geq \mathbb{E}\sqrt{\bar{\boldsymbol{\nu}}_\tau}$, we obtain

$$
(1-\sqrt{\beta_2})\mathbb{E}\sum_{t=0}^{\tau-1}\sqrt{\widetilde{\boldsymbol{\nu}}_t} \leq 2(1-\beta_2)\sigma_1^2\mathbb{E}\sum_{t=1}^{\tau}\left[\frac{\|\boldsymbol{G}_t\|^2}{\sqrt{\beta_2\boldsymbol{\nu}_{t-1}}}\right] + (1-\sqrt{\beta_2})T\sigma_0.
$$

Dividing both sides of the above equation by $1 - \sqrt{\beta_2}$ then gives

$$
\mathbb{E}\sum_{t=0}^{\tau-1}\sqrt{\widetilde{\boldsymbol{\nu}}_t} \leq 4\sigma_1^2\mathbb{E}\sum_{t=1}^{\tau}\left[\frac{\|\boldsymbol{G}_t\|^2}{\sqrt{\beta_2\boldsymbol{\nu}_{t-1}}}\right] + T\sigma_0.
$$

By applying Eq. (21) and the constraint of $\tau$, we obtain that

$$
\begin{aligned}
\mathbb{E}\sum_{t=0}^{\tau-1}\sqrt{\boldsymbol{\nu}_t} \leq{}& T\sigma_0 + 64\frac{\sigma_1^2}{\eta}\left(f(\boldsymbol{u}_1) - \mathbb{E}f(\boldsymbol{u}_{\tau+1}) + \frac{64L_0\eta^2}{1-\beta_2}\left(\ln\frac{6\sqrt{\boldsymbol{\nu}_0 + 2\sigma_0^2 T(1-\beta_2)}}{\sqrt{\boldsymbol{\nu}_0}} - T\ln\beta_2\right)\right. \\
&+ \frac{\eta^3}{1-\beta_2}\operatorname{Poly}\left(L_0, L_1, \sigma_0, \sigma_1, \frac{1}{1-\beta_1}, f(\boldsymbol{w}_1) - f^*\right)\frac{1}{\sqrt{\beta_2\boldsymbol{\nu}_0}} \\
&\left.+ \left(\frac{\eta^3}{1-\beta_2} + \eta\right)\operatorname{Poly}\left(L_0, L_1, \sigma_0, \sigma_1, \frac{1}{1-\beta_1}, f(\boldsymbol{w}_1) - f^*\right)\left(\frac{\|\boldsymbol{G}_1\|^2}{\sqrt{\beta_2\boldsymbol{\nu}_0}}\right)\right) \\
\leq{}& 4T\sigma_0.
\end{aligned}
$$

Here the last inequality is due to $\eta = \frac{\operatorname{Poly}(L_0, L_1, \sigma_0, \sigma_1, \frac{1}{1-\beta_1}, f(\boldsymbol{w}_1)-f^*)}{\sqrt{T}}$, $1 - \beta_2 = \frac{\operatorname{Poly}(L_0, L_1, \sigma_0, \sigma_1, \frac{1}{1-\beta_1}, f(\boldsymbol{w}_1)-f^*)}{T}$, and $T \geq \operatorname{Poly}(L_0, L_1, \sigma_0, \sigma_1, \frac{1}{1-\beta_1}, f(\boldsymbol{w}_1) - f^*)$. Therefore, by Cauchy's inequality, we obtain that

$$
\begin{aligned}
\left(\mathbb{E}\sum_{t=1}^{\tau}\|\nabla f(\boldsymbol{w}_t)\|\right)^2 \leq{}& \left(\mathbb{E}\sum_{t=0}^{\tau-1}\sqrt{\boldsymbol{\nu}_t}\right)\left(\mathbb{E}\sum_{t=1}^{\tau}\left[\frac{\|\boldsymbol{G}_t\|^2}{\sqrt{\beta_2\boldsymbol{\nu}_{t-1}}}\right]\right) \\
\leq{}& 4T\sigma_0 \times \frac{16}{\eta}\left(f(\boldsymbol{w}_1) - f^* + \frac{64L_0\eta^2}{1-\beta_2}\left(\ln\frac{6\sqrt{\boldsymbol{\nu}_0 + 2\sigma_0^2 T(1-\beta_2)}}{\sqrt{\boldsymbol{\nu}_0}} - T\ln\beta_2\right)\right. \\
&+ \frac{\eta^3}{1-\beta_2}\operatorname{Poly}\left(L_0, L_1, \sigma_0, \sigma_1, \frac{1}{1-\beta_1}, f(\boldsymbol{w}_1) - f^*\right)\frac{1}{\sqrt{\beta_2\boldsymbol{\nu}_0}} \\
&\left.+ \left(\frac{\eta^3}{1-\beta_2} + \eta\right)\operatorname{Poly}\left(L_0, L_1, \sigma_0, \sigma_1, \frac{1}{1-\beta_1}, f(\boldsymbol{w}_1) - f^*\right)\left(\frac{\|\boldsymbol{G}_1\|^2}{\sqrt{\beta_2\boldsymbol{\nu}_0}}\right)\right) \\
\overset{(*)}{\leq}{}& 4T\sigma_0 \times \frac{16}{\eta}\left(3(f(\boldsymbol{w}_1) - f^*) + \frac{128L_0\eta^2}{(1-\beta_2)(1-\beta_1)^2}\left(\ln\frac{6\sqrt{\boldsymbol{\nu}_0 + 2\sigma_0^2 T(1-\beta_2)}}{\sqrt{\boldsymbol{\nu}_0}} - T\ln\beta_2\right)\right) \\
\overset{(\bullet)}{\leq}{}& 64T\sigma_0\left(3\sqrt{TL_0(f(\boldsymbol{w}_1) - f^*)} + \frac{128L_0\eta}{(1-\beta_2)(1-\beta_1)^2 T}T\left(\ln\frac{6\sqrt{\boldsymbol{\nu}_0 + 2\sigma_0^2 T(1-\beta_2)}}{\sqrt{\boldsymbol{\nu}_0}}\right)\right) \\
\leq{}& 64T\sigma_0\left(387(1-\beta_1)\sqrt{TL_0(f(\boldsymbol{w}_1) - f^*)}\right),
\end{aligned}
$$

where inequality $(*)$ is due to $\eta = \frac{\text{Poly}(L_0, L_1, \sigma_0, \sigma_1, \frac{1}{1-\beta_1}, f(\boldsymbol{w}_1) - f^*)}{\sqrt{T}}$, $1 - \beta_2 = \frac{\text{Poly}(L_0, L_1, \sigma_0, \sigma_1, \frac{1}{1-\beta_1}, f(\boldsymbol{w}_1) - f^*)}{T}$, and $T \geq \text{Poly}(L_0, L_1, \sigma_0, \sigma_1, \frac{1}{1-\beta_1}, f(\boldsymbol{w}_1) - f^*)$, inequality $(\bullet)$ is due to that $\eta = \frac{\sqrt{f(\boldsymbol{w}_1) - f^*}}{\sqrt{L_0}}$, and last inequality is due to $\frac{1}{(1-\beta_2)T} \ln \frac{6\sqrt{\boldsymbol{\nu}_0 + 2\sigma_0^2 T(1-\beta_2)}}{\sqrt{\boldsymbol{\nu}_0}} \leq 6$.

We then consider two cases: $\tau < T$ and $\tau = T$: for the first case, we have that according to the definition of $\tau$

$$\mathbb{E} \min_{t \in [1,T]} \|\boldsymbol{G}_t\| \mathbb{1}_{\tau < T} \leq \frac{\sqrt[4]{\sigma_0^2 L_0 (f(\boldsymbol{w}_1) - f^*)}}{\sqrt[4]{T}}.$$

For the latter case, we have

$$\mathbb{E} \min_{t \in [1,T]} \|\boldsymbol{G}_t\| \mathbb{1}_{\tau = T} \leq \frac{1}{T} \left( \mathbb{E} \sum_{t=1}^{T} \|\nabla f(\boldsymbol{w}_t)\| \mathbb{1}_{\tau = T} \right) \leq \frac{1}{T} \left( \mathbb{E} \sum_{t=1}^{\tau} \|\nabla f(\boldsymbol{w}_t)\| \right) \leq \frac{256 \sqrt[4]{\sigma_0^2 L_0 (f(\boldsymbol{w}_1) - f^*)}}{\sqrt[4]{T}}.$$

Summing the two inequalities above complete the proof. $\qquad\square$

### D.2 PROOF OF PARAMETER-AGNOSTIC ADAM

#### D.2.1 RELATED WORKS ON PARAMETER AGNOSTIC OPTIMIZATION

**Parameter-agnostic optimization.** The term "parameter-agnostic" implies that the optimizer is capable of converging without the need for extensive hyperparameter tuning or detailed knowledge of the task characteristics. Designing parameter-agnostic or parameter-free optimizers is a significant challenge, as it can help avoid the extensive cost associated with hyperparameter search. Existing works on parameter-agnostic optimization can be categorized into several streams based on the settings they are predicated upon. In the deterministic offline setting, it is widely acknowledged that GD is not parameter-agnostic, even under an $L$-smooth condition Nesterov et al. (2018). However, this can be rectified by combining the GD with the Backtracking Line Search technique Armijo (1966). In the stochastic offline setting, under the $L$-smooth condition, multiple algorithms have been shown to be parameter-agnostic Yang et al. (2023); Ward et al. (2020); Faw et al. (2022); Wang et al. (2023b); Cutkosky & Mehta (2020). More recently, Hübler et al. (2023) demonstrated that Normalized-SGDM can be parameter-agnostic even under an $(L_0, L_1)$-smooth condition. In the realm of online convex optimization, Orabona & Pál (2016); Orabona & Tommasi (2017) have shown there exist parameter-free algorithms achieving optimal dependence regarding not only the final error but also other problem specifics.

#### D.2.2 OUR RESULTS

As we select $\eta = 1/\sqrt{T}$, choosing $1 - \beta_2 = \Omega(1/T)$ has the advantage that the update norm decreases with respect to $T$. This makes Adam parameter-agnostic under the $(L_0, L_1)$-smooth condition, as the update norm will eventually become smaller than $\frac{1}{L_1}$ as $T$ increases.

**Theorem 12.** *Let Assumptions 1 and 2 hold. Then, at the $t$-th iteration, setting $\eta = \frac{1}{\sqrt{t}}$, $\beta_2 = 1 - \frac{1}{\sqrt[4]{t^3}}$, we have that Algorithm 1 satisfies*

$$\mathbb{E} \min_{t \in [1,T]} \|\nabla f(\boldsymbol{w}_t)\| \leq \tilde{\mathcal{O}} \left( \frac{1}{\sqrt[4]{T}} \right).$$

It is shown in Hübler et al. (2023) that Normed-SGDM is parameter-agnostic. Here we show that Adam with a specific scheduler can achieve the same goal.

*Proof.* As described in Section 5, the proof immediately follows by several modifications of the proof of Theorem 11:

- We start our analysis for $t \geq \frac{L_1^4 128^4 \sigma_1^4}{(1-\beta_1)^4}$. For $t \leq \frac{L_1^4 128^4 \sigma_1^4}{(1-\beta_1)^4}$, the function value can be bounded by constant as $\frac{L_1^4 128^4 \sigma_1^4}{(1-\beta_1)^4}$ is independent of $t$;

- For $t \geq \frac{L_1^4 128^4 \sigma_1^4}{(1-\beta_1)^4}$, we have $\max\{\|\boldsymbol{u}_t - \boldsymbol{w}_t\|, \|\boldsymbol{u}_{t+1} - \boldsymbol{w}_t\|\} \leq \frac{1}{L_1}$ since $\|\boldsymbol{w}_{t+1} - \boldsymbol{w}_t\| \leq \frac{1}{\sqrt[4]{t}}$ which also can be used to prove Eq. (25).

The proof is completed. $\qquad\square$

