# OpenReview forum: "On the Convergence of Adam under Non-uniform Smoothness: Separability from SGDM and Beyond"
_ICLR.cc/2025/Conference — Submitted to ICLR 2025_

### Official Review · Reviewer_ubht · 2024-10-24

**Soundness:** 2
**Presentation:** 1
**Contribution:** 1
**Rating:** 1
**Confidence:** 4

**Summary:**

ADAM and momentum based stochastic gradient algorithms are popular optimization methods used in deep learning. Classical analyses of such methods rely on a globally Lipschitz gradient assumption (L-smoothness). Yet   this condition is not satisfied in practical situations. This paper studies stochastic gradient method with momentum and ADAM algorithm under "$L_0, L_1$ smoothness", where compared L-smoothness, the Lipschitz constant of the gradient depends on the gradient norm (hence allowing for non Lipschitz gradients). This follows many works dealing with relaxed smoothness conditions.

The authors provide convergence rates in (expected) best iterate, or averaged gradient norms. They also provide lower bounds by exhibiting one dimensional functions attaining the guarantees.

**Strengths:**

L0,L1 smoothness assumptions are interesting to consider as they better coincide with deep learning situations.

**Weaknesses:**

Overall, the paper brings no substantial contribution compared to the literature.


- The convergence rates use classical and elementary tools (smoothness inequalities, stopping time in the stochastic setting) that already exist in the literature. For instance, similar convergence rates were established in Li et al., 2023a. The authors highlight some differences (section 4.1) with the related works but the arguments seem wrong or irrelevant.

- Some mathematical parts are wrong. For instance, the authors consider the regularization parameter $\lambda$ to be zero but this leads to divisions by zero in the proofs (see for instance line 857) with no further assumption.

- The lower bounds are based on trivial and degenerate situations, especially in the stochastic setting. The lower bound function (Theorem 10 in Appendix) is based on a gradient noise which image by the gradient is not integrable, thus leading to a vacuous bound (all expectations become infinite). For instance, it does not depend on the smoothness parameters, which was the motivation stated in the introduction. Exhibiting the lower bounding function in the main text would make the presentation more honest and transparent. Lower bounds found in the literature are based on constructions that are much more sophisticated because they exhibit more restrictive properties (stability, gradient boundedness, etc.); see, for instance, Carmon et al., Arjevani et al.

**Questions:**

- Remove the blue color in theorems.
- Define the sequence $\( w_t \)$ in Theorems 3, 4, 5. Also, add a proper name for each theorem to understand which algorithm each result deals with.
- The appendix needs a thorough re-reading; for instance, look at line 2015: "Also, as  $\tau$  is a bounding stopping theorem, by optional stopping time, we obtain ..."
- Writing literature reviews in the appendix in the middle of the proof makes the reading inefficient.
- Line 465: the quantifier $\forall$ looks weird this way. I suggest writing "for all."
- Throughout the paper, rewrite the left quotation mark in the correct direction.
- Avoid cumbersome expressions such as "to the best of our knowledge," "for the first time" (used many times on page 7).

---

> ### Author Response · Authors · 2024-12-01
> **Thanks for your review**
>
> We thank the reviewer for the review.
>
> > The convergence rates use classical and elementary tools (smoothness inequalities, stopping time in the stochastic setting) that already exist in the literature. For instance, similar convergence rates were established in Li et al., 2023a. The authors highlight some differences (section 4.1) with the related works but the arguments seem wrong or irrelevant.
>
> We kindly request the reviewer to illustrate more on why "The authors highlight some differences (section 4.1) with the related works but the arguments seem wrong or irrelevant" as we found no supporting evidence for this claim in the review.
>
> >  Some mathematical parts are wrong. For instance, the authors consider the regularization parameter
>  to be zero but this leads to divisions by zero in the proofs (see for instance line 857) with no further assumption.
>
> The claim "this leads to divisions by zero in the proofs" is **wrong**. $\nu_t$ can't be zero, as in Algorithm 1 we have described that $\nu_0 \in \mathbb{R}^+$ and since $\nu_t = \beta_2 \nu_{t-1} + (1-\beta_2) g_t^2$, one can easily prove $\nu_t > 0$ by induction.
>
> > The lower bounds are based on trivial and degenerate situations, especially in the stochastic setting. The lower bound function (Theorem 10 in Appendix) is based on a gradient noise which image by the gradient is not integrable, thus leading to a vacuous bound (all expectations become infinite). For instance, it does not depend on the smoothness parameters, which was the motivation stated in the introduction. Exhibiting the lower bounding function in the main text would make the presentation more honest and transparent. Lower bounds found in the literature are based on constructions that are much more sophisticated because they exhibit more restrictive properties (stability, gradient boundedness, etc.); see, for instance, Carmon et al., Arjevani et al.
>
> **We believe the reviewer has completely misunderstood our results of lower bounds.** First of all, the result does depend on the smoothness parameter -- that the smoothness parameter is unbounded is why SGDM fails to converge (if assuming uniformly smoothness, SGDM is guaranteed to converge under proper hyperparameters, which is a folktale). Secondly, it is not our purpose to make the construction "much more sophisticated". Our purpose is to show the separation between SGDM and Adam.

---

> > ### Comment · Reviewer_ubht · 2024-12-01
> >
> > - For instance the square dependence on the function value does not appear in Wang et al. You mention the issue of having lambda = 0 in Li et Al. 2023a. but in your work this is traded against the dependence on  nu_0., this is the same.
> >
> > - Your results can basically be retrieved from Li et Al 2023a by applying Cauchy Schwarz inequality on their bound.
> >
> > - Indeed  the proof works with nu0 positive but it does not appear in the paper.
> >
> > - My point about the lower bound is that you consider a degenerate case which is not relevant because it falls outside the setting of your upperbound ( expectations are infinite).

---

> ### Author Response · Authors · 2024-12-02
> **Response**
>
> Dear reviewer,
>
> > For instance the square dependence on the function value does not appear in Wang et al. You mention the issue of having lambda = 0 in Li et Al. 2023a. but in your work this is traded against the dependence on nu_0., this is the same.
>
> For Wang et al.: the stated rate for Wang et al. is calculated from their Theorem 2 by minimizing their bound with respect to a, b, c (which is rather straightforward). Nevertheless, we will increase the calculation in the revised version.
>
> For Li et al., we never claim that having lambda \ne 0 is an issue in Li et al, but rather "presents a convergence rate that depends polynomially on 1/\lambda" (line 196 in our paper) is. Our bound does not have a dependence over $1/\nu_0$, as can be seen from lines 1609-1618.
>
> > Your results can basically be retrieved from Li et Al 2023a by applying Cauchy Schwarz inequality on their bound.
>
> This claim is **WRONG** and if the reviewer insists not, we kindly request the reviewer to illustrate in details. A basic fact is, that Theorem 4.1 in Li et al. uses the bounded noise assumption $|g_t-G_t| \le \sigma$ **almost surely**, while our assumption uses a much looser assumption of noise, allowing $\vert g_t - G_t \vert $ to be linear in $\vert G_t \vert$ **in expectation**. Such a gap is **impossible** to be bridged by simply "applying Cauchy Schwarz inequality on their bound".
>
> > Indeed the proof works with nu0 positive but it does not appear in the paper.
>
> **IT DOES APPEAR IN THE PAPER.** Please see line 149 in algorithm 1.
>
> > My point about the lower bound is that you consider a degenerate case which is not relevant because it falls outside the setting of your upperbound ( expectations are infinite).
>
> We are confused as to why it falls outside the setting of our upper bound. Kindly ask the reviewer to illustrate. As we described in Theorem 4, the lower bound result uses the same set of assumptions as theorem 3, and the results are also in the same form: Theorem 3 uses $1/T \sum E \Vert \nabla f(w_t) \Vert $, which naturally leads to an upper bound of $\min E \Vert \nabla f(w_t) \Vert $, while Theorem 4 presents a lower bound of $\min E \Vert \nabla f(w_t) \Vert $ and matches Theorem 3.

---

> > ### Comment · Reviewer_ubht · 2024-12-02
> >
> > - Sorry I did not see that it was written at line 149, but how come $\nu_0$ has to be chosen positive and at the same time, the bound does not depend on $\nu_0$? Something is wrong here.
> >
> > - Line 1609 suggests as you say that $\nu_0$ is not involved, but at line 1509 you take $\nu_0 = \sigma_0$, so dependence on $\nu_0$ comes from $\sigma_0$... Overall the proofs contain too many (not useful) layers and I doubt they significantly improve existing results. Such a theoretical paper should give more insights into the proof techniques and more transparency in order to be useful to the community.
> >
> > - About your lower bound, if the conditional expectation of the gradient is infinite then it cannot be upper bounded in expectation. You proof only require to build the lower bound function on the first iterate, it is then obvious that something is going wrong here.
> >
> >
> > Maybe you were able to negociate some boundedness assumptions compared to past works (such as the gradient noise) but this remains incremental to me. The contributions that you claimed to bring were rather on the lower bound, relaxing   the dependence on the conditioning parameter $\lambda $ or $\nu_0$ and on the  function value optimality gap (but the literature review is unclear). You answers did not address my concerns on these points. Besides the weaknesses of the contributions, the presentation and the redaction also have to be reworked (see my past remarks).

---

> ### Author Response · Authors · 2024-12-03
> **Comment**
>
> Dear Reviewer,
>
> > "negociate some boundedness assumptions compared to past works" is not incremental, as the main contribution of this COLT paper is doing so for AdaGrad "The Power of Adaptivity in SGD: Self-Tuning Step Sizes with Unbounded Gradients and Affine Variance".
>
> Nevertheless, we never claim "negociate some boundedness assumptions compared to past works" is a main contribution of our work. If the reviewer still remembers, it is just a response to the reviewer's claim "Your results can basically be retrieved from Li et Al 2023a by applying Cauchy Schwarz inequality on their bound.".
>
> > how come $\nu_0$ has to be chosen positive and at the same time, the bound does not depend on $\nu_0$
>
> We never claim that the bound does not depend on $\nu_0$ -- line 1609 shows there is a dependence over $\tilde{\nu}_1$, which in turn leads to the dependence over $\nu_0$. Also, in line 1509 the reviewer can see the bound **positively correlates with $\nu_0$**, which means if the reviewer chooses $\nu_0$ much smaller, the bound becomes even better (in coefficients).
>
> > About your lower bound, if the conditional expectation of the gradient is infinite then it cannot be upper bounded in expectation. You proof only require to build the lower bound function on the first iterate, it is then obvious that something is going wrong here.
>
> We don't understand the reviewer's point here. Does the reviewer agree that the lower bound result uses the same set of assumptions as the upper bound result? Also, there is a misunderstanding that our proof only applies to the first iterate, from line 1628 to line 1662, the proof applies to all iterate. We kindly request that if the reviewer believes that the proof is wrong, the reviewer needs to point out the specific line of proof where it is wrong, instead of saying that "it is then obvious that something is going wrong here."

---

> > ### Comment · Reviewer_ubht · 2024-12-03
> >
> > About your "lower bound" function: Yes, it uses the same assumptions, but your lower bound applies to minimal expected gradient norm while your upper bound applies to expected averaged gradients norms. Using the minimal gradient norm enables your shortcut by setting the expectations to infinity, which makes things much more simple.
> >
> > You can try it yourself: compute averaged expected gradients on your lower bounding function and you will see that it cannot be upper bounded in this sense.

---

> ### Author Response · Authors · 2024-12-03
> **Response**
>
> Dear reviewer,
>
> we don't understand your point here -- you seemingly refer to that the considered "convergence" is not the same for the upper bound and lower bound result, right?
>
> To be clear:
>
> * For the upper bound, we consider $1/T \sum_t E \Vert \nabla f(w_t) \Vert $.
>
> * For the lower bound, we consider $min E \Vert \nabla f(w_t) \Vert$.
>
> The reviewer seems to suggest that these two measures are not the same, right?  But this is not a problem -- we have $min E \Vert \nabla f(w_t) \Vert \le 1/T \sum_t E \Vert \nabla f(w_t) \Vert$ -- that means if we have a lower bound for $min E \Vert \nabla f(w_t) \Vert $, we will immediately have a lower bound for $ 1/T \sum_t E \Vert \nabla f(w_t) \Vert$. Similarly, if we have an upper bound for   $ 1/T \sum_t E \Vert \nabla f(w_t) \Vert$, we will immediately have an upper bound  $ min E \Vert \nabla f(w_t) \Vert$.
>
> The above reasoning shows, although our upper bound is in $1/T \sum_t E \Vert \nabla f(w_t) \Vert $, it trivially leads to an upper bound in $\min E \Vert \nabla f(w_t) \Vert$; the same for the lower bound -- the current lower bound also trivially leads to a lower bound in $\min E \Vert \nabla f(w_t) \Vert$.
>
> **Therefore, both comparisons in $\min E \Vert \nabla f(w_t) \Vert$ and $1/T \sum_t E \Vert \nabla f(w_t) \Vert$ lead to the separation of SGDM and Adam.**
>
> **This is not short cut, and we prove something stronger --  the stronger version of the upper bound, i.e., $1/T \sum_t E \Vert \nabla f(w_t) \Vert$, and the stronger version of lower bound, i.e., min E \Vert \nabla f(w_t) \Vert**.
>
> > You can try it yourself: compute averaged expected gradients on your lower bounding function and you will see that it cannot be upper bounded in this sense.
>
> We have to say we have been very confused by the reviewer's comment... Why do we need to upper bound the average expected gradient? Isn't it our goal to lower-bound it? If the reviewer is referring to the lower bound, the concerns should be resolved by the above reasoning.

---

### Official Review · Reviewer_ksq7 · 2024-11-02

**Soundness:** 2
**Presentation:** 3
**Contribution:** 3
**Rating:** 3
**Confidence:** 4

**Summary:**

This paper aims to theoretically compare the convergence rates of Adam and Stochastic Gradient Descent with Momentum (SGDM) to explore the superior empirical performance of Adam over SGDM. Specifically, it separates Adam and SGDM regarding the convergence rate under $(L_0,L_1)$-smooth condition in both deterministic and stochastic settings. One interesting point in this paper is that the authors used a novel stopping time technique to show Adam's convergence rate in terms of the minimum gradient norm matching the lower bound for first-order stochastic optimizers, considering all problem hyperparameters. With the aid of this technique, the authors show that Adam with a specific parameter scheduler is parameter-agnostic.

**Strengths:**

1. This work analyzes Adam's convergence rate under mild assumptions. Specifically, the convergence results are established under the $(L_0,L_1)$-smooth condition and affine noise variance assumption, relaxing the commonly used $L$-smooth condition and bounded variance assumption in prior studies.

2. The obtained upper bound on the convergence rate of Adam is tighter than those given in prior studies and matches the lower bound of first-order optimizers in both deterministic and stocahstic settings.

3. A novel stopping time technique is developed. Based on this technique, it is demonstrated in Theorem 12 that Adam with some **specific parameter scheduler** is parameter-agnostic.

**Weaknesses:**

1. The proof of Theorem 3 in Appendix C.1 appears to be **NOT complete**. Specifically, in deriving an upper bound for Adam in terms of the average gradient norm, the first-order term $\mathbb{E}^{\vert \mathcal{F}_t}[\langle\nabla f(w_t),u_{t+1}-u_t\rangle]$ is decomposed into three components as shown in equation (7). However, referring to equation (8), the combination of inequalities (11) and (12) provides an upper bound only for the second term on the RHS of equation (7). Then, the RHS of inequality (13) provides an upper bound only for the sum of the first two terms, neglecting the third term on the RHS of equation (7). Thus, the proof lacks justification for establishing an upper bound on the third term.

2.  Some statements in the paper are not consistent. The consistency of the statements is critical for a theoretical paper. For example, the proof of Theorem 3 provided in Appendix C.1 relies on assumptions that $\beta_1 \leq \beta_2$ (line 1390 in page 26), $0\leq \beta_1^2 < \beta_2 < 1$ (requirement of Lemma 1), $\beta_2 \geq \frac{1}{2}$ and $1-\beta_2\leq \frac{1-\beta_1}{1024\sigma_1^2}$ (line 1416 in page 27), which are not captured in the corresponding formal statement. Similarly, the formal statement of Theorem 5 omits assumptions that $\beta_1 \leq \beta_2$ and $0\leq \beta_1^2 < \beta_2 < 1$ (see Theorem 11 in Appendix D.1).

3. The convergence results for Adam presented in Theorem 3 and Theorem 5 stipulate "a proper choice of learning rate $\eta$ and momentum hyperparameter $\beta_2$". However, as shown in the corresponding formal statements in Appendix C.1 (Theorem 9) and Appendix D.1 (Theorem 11), deriving these results necessitates tuning $\eta$ and $\beta_2$ with prior knowledge of the non-uniformly bounded smoothness parameters $(L_0,L_1)$ and affine noise variance parameters $(\sigma_0,\sigma_1)$. Since these parameters may be unknown or difficult to estimate in realistic situations, the practical utility of the derived results is questionable as their conditions on hyperparameters would be challenging to satisfy.

4. There is a lack of clarity around the focused version of Adam. Two different versions are assumed in this paper: the coordinate-wise version and the norm version. While Algorithm 1 introduced in the main text is the coordinate-wise Adam, it is not until Appendix B that the authors first state that all the subsequent theorems apply to the norm version of Adam. Although a footnote in the appendix mentions that an extension to the coordinate-wise version from the norm version is not difficult, more detailed discussions are necessary and helpful.

5. There are some typos in this paper. Also, the citation format should be carefully checked.

Overall, I found this paper made some interesting theoretical comparisons between Adam and SGDM. But the rigorous proofs and statements are the key to a theoretical paper. **My current rating of this paper is under the assumption that the first point above can be fully addressed**.

**Questions:**

The main concerns from my perspective for this paper have been listed in the **Weakness** part. Below are some more related questions that need to be addressed:

1. How to ensure that the choice of $\beta_2$ provided in Theorem 9 satisfies all the assumptions ($\beta_1 \leq \beta_2$, $0\leq \beta_1^2 < \beta_2 < 1$, $\beta_2 \geq \frac{1}{2}$ and $1-\beta_2\leq \frac{1-\beta_1}{1024\sigma_1^2}$) required in the derivation of this result?

2.  How to ensure that the choice of $\beta_2$ provided in Theorem 11 satisfies all the assumptions ($\beta_1 \leq \beta_2$ and $0\leq \beta_1^2 < \beta_2 < 1$) required in the derivation of this result? If such a choice of $\beta_2$ exists, can we guarantee that inequality $\frac{\eta}{\sqrt{1-\beta_2}}\leq \frac{1-\beta_1}{128L_1\sigma_1}$ given on page 39 hold true?

3. Could you please give more details on the choice of $\beta_2$ and the derivation of the inequality $\Vert w_{t+1} - w_t\Vert \leq \frac{1}{\sqrt[4]{t}}$ in the case where $t\geq \left(\frac{128\sigma_1 L_1}{1-\beta_1}\right)^4$?

---

> ### Comment · Reviewer_ksq7 · 2024-12-01
>
> Since the authors haven't addressed my concerns about the proof of Theorem 3 yet, I modified my rating of this paper. I will re-evaluate the paper after receiving the updates from the authors.

---

> ### Author Response · Authors · 2024-12-01
> **Thanks for your careful review**
>
> We want to thank the reviewer for the careful review sincerely. Also, sorry for the late response -- the authors just got time to carefully read the reviews today.
>
> 1. For theorem 3: thanks for pointing it out and you are correct. We missed the third term in the proof and apologize for that. Nevertheless, the third term itself is easy to bound (a similar proof can be found in eq. (13) in "Closing the Gap Between the Upper Bound and the Lower Bound of Adam’s Iteration Complexity"). Specifically, this term can be bounded as
> $$\frac{\beta_1 \eta(1-\beta_2)\sigma_0^2}{1-\beta_1/\sqrt{\beta_2}} \Vert G_t\Vert \Vert m_{t-1}\Vert  \frac{1}{\sqrt{\tilde{\nu}_t {\nu}\_{t-1} }(\sqrt{\tilde{\nu}_t} + \sqrt{ {\nu}\_{t-1}})} \le O(\Vert G_t\Vert^2/\sqrt{\tilde{\nu}_t}) + O(\eta\sigma_0\sqrt{1-\beta_2}/(1-\beta_1)^2 \Vert m\_{t-1}\Vert^2 /\nu\_{t-1})$$. The two terms on the right side all appear at eq. (13), line 1269 (the latter one has a different footscript $t-1$, but this does not matter as the latter we take summation of t), and thus the rest of the proof is the same. We will include the missing proof in the latter version.

---

> ### Author Response · Authors · 2024-12-01
> **Other concerns**
>
> 2. On the restrictions: Maybe we got the reviewer wrong, but the restrictions of $\beta_2$ and $\beta_1$ has already been captured in line 1102 $\beta_2 = 1- \eta^2 (...)^2$. Similar for the reviewer's first two questions.
>
> 3. For the optimal solution of $\beta_2$: we only intend to indicate the "existence" of the optimal hyperparameters that make Adam's convergence meet the lower bound. In practice, there is more direct practice such as grid search to get the optimal hyperparams.
>
> 4. Extension to Adam: We thank the reviewer for the suggestion. We will include a detailed explanation on how to extend the result to coordinate-wise Adam in the revised version of this paper.

---

### Official Review · Reviewer_toeM · 2024-11-04

**Soundness:** 3
**Presentation:** 3
**Contribution:** 3
**Rating:** 8
**Confidence:** 2

**Summary:**

The paper presents a convergence analysis of Adam and SGD with Momentum (SGDM) under the non-uniform (L_0, L_1)-smoothness assumption (where the gradient smoothness depends on the gradient norm itself) and an affine noise assumption on the gradient in the stochastic setting. The authors show that under these assumptions Adam has an upper bound that is better than the lower bound of (S)GDM in both the deterministic and stochastic settings.

In the deterministic setting the authors provide a tighter upper bound for Adam than two cited previous papers and for GDM they re-iterate on a prior known lower bound in more realistic setting (with choosing learning rate after the objective function is known). Based on these two bounds the authors are then able to separate the convergence rates of Adam and GDM showing that Adam is superior in the deterministic setting.

The authors then move on to extend their analysis to the stochastic setting where they claim that their new Adam upper bound extends prior convergence results of Adam and clipped SGDM with their new Adam bound giving the best upper bound achieved under the assumptions made in the paper so far. For the lower bound of SGDM the authors provide a bound that is tighter than the prior results from the literature to be able to separate the Adam and SGDM bounds.

In the final sections the authors discuss what happens when the “regularizing” constant lambda is used and introduce a stopping-time analysis under which they discuss how Adam can actually achieve the optimal lower bound rate.

**Strengths:**

* Paper is clearly structured, moving from motivations to assumptions, to the simpler determinstic setting to the more complex stochastic section.
* Paper provides clear exposition and arguments for the practical relevance of the non-uniform smoothness assumptions they use.
* Paper provides the intuition behind the different convergence behaviors e.g. explaining the behavior of the adaptive conditioner used in AdaGrad or GDM’s inability to manage varying degrees of sharpness in the loss function.
* Authors explain the limitations of a previous derivation of the lower bound of determinstic GD in the literature and use this to motivate their lower bound formulation that allows adjusted hyperparams (learning rate) after the objective function has been selected but arrives at the same bound and is taking into account momentum.
* For their Adam upper bound in the stochastic case they provide a high level summary of the additional challenges addressed compare to an analysis of Adam under L-smooth condition from the literature.

**Weaknesses:**

* The paper could perhaps be improved with a more extensive overview/discussion of the attributes of optimizers that allow the type of analysis they provide for Adam (as an extension of e.g. their brief discussion on AdaGrad). The authors already provide the intuition behind why e.g. SGDM fails to converge under non-uniform smoothness etc. but perhaps these reasons could be summarized in a sort of "intuitions behind the results" paragraph earlier in the paper.

**Questions:**

How does this paper relate to the following prior work which also analyzes Adam and SGD(M) under non-uniform smoothness:
```@inproceedings{wang2024provable,
  title={Provable adaptivity of adam under non-uniform smoothness},
  author={Wang, Bohan and Zhang, Yushun and Zhang, Huishuai and Meng, Qi and Sun, Ruoyu and Ma, Zhi-Ming and Liu, Tie-Yan and Luo, Zhi-Quan and Chen, Wei},
  booktitle={Proceedings of the 30th ACM SIGKDD Conference on Knowledge Discovery and Data Mining},
  pages={2960--2969},
  year={2024}
}
```

---

### Official Review · Reviewer_r4j3 · 2024-11-07

**Soundness:** 2
**Presentation:** 3
**Contribution:** 2
**Rating:** 5
**Confidence:** 3

**Summary:**

This paper provides a comparative analysis of Stochastic Gradient Descent with Momentum (SGDM) and simplified Adam (considering the same step-size for each coordinate and ignoring the corrective terms), focusing on their convergence rates under varying conditions. The authors argue that the variant of Adam converges faster than SGDM when dealing with non-uniformly bounded smoothness.

In the deterministic environments, the authors claim that simplified Adam can reach the optimal convergence rate for deterministic first-order methods, while SGDM shows a higher dependence on the initial function value, suggesting a slower convergence rate.

In the stochastic setting: the authors claim that simplified Adam's convergence rate upper bound is shown to align with the theoretical lower bounds for stochastic first-order methods, considering both the initial function value and final error. The authors highlight that SGDM can sometimes fail to converge, regardless of the learning rate, under certain conditions.

The paper use a stopping-time-based technique, enabling analysis of the minimum gradient norm over iterations. This method demonstrates that simplified Adam’s convergence rate can match lower bounds.

**Strengths:**

The article establishes convergence theory for Stochastic Gradient Descent with Momentum (SGDM) and simplified Adam.
The comparsion of the convergence theory between SGDM with simplified Adam could be interesting from some certain points of view.

**Weaknesses:**

The paper establishes convergence results only for a simplified version of Adam, where uniform step sizes are applied across coordinates, and corrective terms are ignored. This limitation leaves the convergence behavior of the full coordinate-wise Adam algorithm unaddressed.

Notation issues: Notation is inconsistently applied, leading to potential confusion. For instance, in the algorithm’s statement, $v_t$
is a vector, but in the appendix proof,
$v_t$ is treated as a scalar.

It seems some of the cited references may already have derived optimal convergence rates for deterministic Adam (or provided results in the stochastic setting that could be applied to the deterministic setting). A thorough comparison with these existing results would strengthen the paper and clarify its contributions.

The stopping rule employed appears similar to those used in analyses of other stochastic gradient methods under generalized smoothness conditions. The paper may clarify what novelty this rule brings to the current analysis. Additionally, please clarify the dependence of
$\tau$ on random variables. Since
$\tau$ itself is a random variable, it’s unclear how the inequality in line 2019 follows directly from Equation (26). Similar clarifications may be needed for the statements in lines 2276 and 2280.

Is there any explicit expression for $\beta_2$ from Theorem 9?

The overall write-up would benefit from improvements in clarity. For example, several formulas in the appendix exceed the width and could be improved to improve readability.

**Questions:**

see weaknesses

---

### Meta-Review · Area_Chair_dFmv · 2024-12-10

**Metareview:**

The paper presents a convergence analysis comparing Adam and SGDM under non-uniform smoothness assumptions in both deterministic and stochastic settings. The study introduces stopping-time techniques to show that Adam achieves convergence rates that align with lower bounds for first-order methods, while SGDM struggles under specific conditions.

There was a strong divergence of opinion in the initial reviews but the authors unfortunately did not reply to all the reviewers. Some concerns are important and regard the correctness of the analysis. I'm therefore not in a position to accept the paper and ask the authors to address the comments from the reviewers in a resubmission.

**Additional Comments On Reviewer Discussion:**

The authors unfortunately did not reply to all the reviewers, they only provided a subset of answers to some reviewers. Many concerns were not addressed.

---

### Decision · Program_Chairs · 2025-01-22

Reject